# Sample Complexity of Distributionally Robust Off-Dynamics Reinforcement Learning with Online Interaction

**Yiting He** [* 1]  **Zhishuai Liu** [* 1]  **Weixin Wang** [1]  **Pan Xu** [1]

## Abstract

Off-dynamics reinforcement learning (RL), where training and deployment transition dynamics are different, can be formulated as learning in a robust Markov decision process (RMDP) where uncertainties in transition dynamics are imposed. Existing literature mostly assumes access to generative models allowing arbitrary state-action queries or pre-collected datasets with a good state coverage of the deployment environment, bypassing the challenge of exploration. In this work, we study a more realistic and challenging setting where the agent is limited to online interaction with the training environment. To capture the intrinsic difficulty of exploration in online RMDPs, we introduce the supremal visitation ratio, a novel quantity that measures the mismatch between the training dynamics and the deployment dynamics. We show that if this ratio is unbounded, online learning becomes exponentially hard. We propose the first computationally efficient algorithm that achieves sublinear regret in online RMDPs with $f$-divergence based transition uncertainties. We also establish matching regret lower bounds, demonstrating that our algorithm achieves optimal dependence on both the supremal visitation ratio and the number of interaction episodes. Finally, we validate our theoretical results through comprehensive numerical experiments.

## 1. Introduction

Off-dynamics reinforcement learning (RL) (Eysenbach et al., 2021; Lyu et al., 2024) has recently gained significant attention in scenarios where the transition dynamics of the deployment environment differs from that of the training environment. Such problems can be modeled as learning a robust Markov decision process (RMDP) (Satia & Lave Jr, 1973; Iyengar, 2005; Nilim & El Ghaoui, 2005), where the objective is to learn a policy that performs well when uncertainties are imposed into the transition dynamics. Two major frameworks have been proposed in the literature to incorporate the uncertainty of transition dynamics in RMDPs. The first, known as the Constrained Robust Markov Decision Process (CRMDP), formulates a max-min optimization problem that seeks the best policy under the worst-case transition dynamics within a predefined uncertainty set. The second, the Regularized Robust Markov Decision Process (RRMDP), replaces the hard constraint on uncertainty sets with a regularization term that quantifies the divergence between the training and deployment dynamics.

CRMDP was initially introduced for optimal control problems (Nilim & El Ghaoui, 2005; Iyengar, 2005; Xu & Mannor, 2006; Wiesemann et al., 2013), where the transition dynamics and reward functions of the nominal MDP are assumed to be fully known. More recently, CRMDPs have been studied from a learning perspective (Zhou et al., 2021; Yang et al., 2022), where an agent must gather data to estimate the environment rather than relying on perfect knowledge. Existing research on learning CRMDPs can be categorized into three settings: (1) *Learning with a generative model (simulator).* In this setting, the agent can query transitions at any state-action pair an arbitrary number of times (Panaganti & Kalathil, 2022; Yang et al., 2022; Xu et al., 2023b; Shi et al., 2024). (2) *Learning with an offline dataset.* Here, the agent learns from a pre-collected dataset, typically assumed to be generated by a behavior policy from the nominal MDP (Panaganti et al., 2022; Blanchet et al., 2023; Shi & Chi, 2024; Liu & Xu, 2024b; Tang et al., 2024). Effective robust policy learning relies on sufficient coverage of the dataset on states in the deployment environment. (3) *Learning through online interaction.* More recent works have considered online learning of CRMDPs through direct interaction with the training environment to collect data (Liu & Xu, 2024a; Lu et al., 2024; Liu et al., 2024), focusing on a specific case where the uncertainty set is defined via total

---

*Equal contribution [1]Duke University. Correspondence to: Pan Xu <pan.xu@duke.edu>.

*Proceedings of the 42$^{nd}$ International Conference on Machine Learning*, Vancouver, Canada. PMLR 267, 2025. Copyright 2025 by the author(s).

variation (TV) distance[1].

RRMDP was introduced to get rid of the constrained optimization in the formulation of CRMDPs for the tractability of robust policy learning (Yang et al., 2023; Zhang et al., 2024). In particular, Yang et al. (2023) studied RRMDPs with general $f$-divergence-based regularization under a generative model setting, while Zhang et al. (2024) analyzed RRMDPs with Kullback-Leibler divergence-based regularization in the offline setting, relying on similar data coverage assumptions as in offline CRMDPs. More recently, Panaganti et al. (2024) extended this work to general $f$-divergence-based regularization in the offline setting and explored RRMDPs with total variation regularization and fail-states in a hybrid online-offline setting.

Despite these advances, in more realistic applications where simulators or pre-collected datasets with strong state coverage are not available, the problem of efficient online exploration in RMDPs remains understudied. Unlike standard MDPs, where exploration aims to reduce uncertainty within a fixed transition model, in RMDPs, the agent can only gather experience from a nominal environment, yet it must generalize to potentially shifted dynamics at deployment. This presents a fundamental challenge of *information deficit* in online RMDPs, requiring the development of exploration strategies that proactively account for distributional shifts. More specifically, the *information deficit* in online RMDPs arises when states that are rarely visited in the nominal environment become critical in the deployment environment. For instance, consider a state $s$ in the nominal MDP that is extremely difficult to visit, e.g., with exponentially small visitation probability, resulting in limited data collection. If, in the deployment environment, the dynamics shift increases the visitation probability of $s$, the agent must make informed decisions at this state despite having little prior experience. In standard MDPs, such rare states typically have negligible effects on policy learning, but in RMDPs, they can critically impact performance, making online learning in RMDPs significantly more challenging than in standard MDPs.

To overcome this information deficit issue, existing research on online RMDPs adopt a fail-state type of assumption–there exist states with zero reward that only transit among themselves (Liu & Xu, 2024a; Lu et al., 2024; Liu et al., 2024). As we show in Proposition 5.4 and the discussion following it, these assumptions essentially ensure that worst-case distribution shifts occur in a deterministic direction, which eliminating the information deficit issue and makes provably efficient online learning possible. However,

such nice properties do not hold in RMDPs with general $f$-divergence based uncertainty sets or regularization, limiting all existing research on online RMDPs to CRMDPs with TV-distance based uncertainty sets.

In this work, we answer the following fundamental question:

*Under what conditions can provably efficient online learning of RMDPs be achieved?*

We investigate tabular RMDPs with finite states and actions. We show that if the nominal environment is sufficiently exploratory—i.e., the agent can collect enough information through interaction—then sample-efficient online learning should be achievable for broader classes of RMDPs, including those with general $f$-divergence based dynamics uncertainties and without restrictive structural assumptions like fail-states. We rigorously prove that the sample complexity of any online learning algorithm should be proportional to the difficulty of exploration.

Our contributions are summarized as follows.

- We introduce the *supremal visitation ratio* $C_{vr}$ (see Assumption 5.5) as a measure of exploration difficulty in RMDPs. We develop the first computationally efficient algorithm, Online Robust Bellman Iteration (ORBIT), for CRMDPs and RRMDPs based on the total variation (TV), Kullback-Leibler (KL), and $\chi^2$ divergences, and prove sample complexity bounds that explicitly depend on $C_{vr}$.
- We establish regret lower bounds, demonstrating that the supremal visitation ratio $C_{vr}$ is an unavoidable term in the sample complexity of online RMDP learning. This result confirms that $C_{vr}$ serves as a fundamental measure of exploration difficulty and a sufficient condition for provably efficient online learning in RMDPs. As a corollary, we construct hard instances to demonstrate that if $C_{vr}$ is unbounded, general online learning in CRMDPs can become exponentially difficult.
- We conduct comprehensive numerical experiments to validate our theoretical findings. In a simulated MDP, we show that the performance of learned policies degrades as $C_{vr}$ increases. We evaluate our algorithms in a simulated RMDP and the Frozen Lake environment, highlighting their effectiveness when distribution shifts are significant.

**Notations** For any positive integer $H \in \mathbb{Z}_+$, we denote $[H] = \{1, 2, \cdots, H\}$. For any set $\mathcal{S}$, define $\Delta(\mathcal{S})$ as the set of probability distributions over $\mathcal{S}$. Let $P, Q \in \Delta(\mathcal{S})$ and $P \ll Q$. For a convex function $f : [0, +\infty) \to (-\infty, +\infty]$ such that $f(x)$ is finite for all $x > 0$, $f(1) = 0$ and $f(0) = \lim_{t \to 0^+} f(t)$. The $f$-divergence of $P$ from $Q$, which measures their difference, is defined as $D_f(P\|Q) =$

---

[1]We note that Dong et al. (2024) also studied online CRMDPs, but we found essential flaws in proofs of their Lemmas A.2 and C.5, which invalidates their results.

$\int_\Omega f(\frac{dP}{dQ}) \, dQ$. In our paper, we consider three common $f$-divergences including total variation (TV) distance with $f(t) = \frac{1}{2}|t-1|$, Kullback-Leibler (KL) divergence with $f(t) = t \ln t$, and $\chi^2$-divergence with $f(t) = (t-1)^2$. We use $\mathcal{O}(\cdot)$ to hide absolute constant factors and $\widetilde{\mathcal{O}}(\cdot)$ to further hide logarithmic factors. For any two integers $a$ and $b$, we denote $a \vee b := \max\{a, b\}$.

## 2. Related Work

**CRMDPs and RRMDPs** The framework of CRMDPs was first introduced in the context of optimal control (Iyengar, 2005; Nilim & El Ghaoui, 2005; Xu & Mannor, 2006; Wiesemann et al., 2013; Mannor et al., 2016), where the nominal MDP is assumed to be exactly known, and robust policies are obtained by solving a constrained max-min optimization problem. Subsequent works extended CRMDPs to the learning setting with access to a generative model (Zhou et al., 2021; Yang et al., 2022; Panaganti & Kalathil, 2022; Shi et al., 2024). More recently, CRMDPs have been studied in the offline learning setting, where only a pre-collected dataset from the nominal MDP is available through a behavior policy (Shi & Chi, 2024; Panaganti et al., 2022; Blanchet et al., 2023; Wang et al., 2024a; Liu & Xu, 2024b; 2025). To ensure that a robust policy can be learned from a reasonably sized offline dataset, these works make assumptions about the behavior policy (and implicitly, the nominal MDP) to guarantee sufficient coverage. Such assumptions include the robust single-policy clipped concentrability (Shi & Chi, 2024), robust partial coverage (Blanchet et al., 2023), and uniformly well coverage assumptions (Liu & Xu, 2024b; Wang et al., 2024a). The framework of RRMDPs was more recently proposed by Yang et al. (2023) and Zhang et al. (2024), who studied it under the generative model setting and the offline setting, respectively. This line of work was extended to function approximation settings by Panaganti et al. (2024) and Tang et al. (2024), considering both hybrid offline-online and purely offline scenarios.

It is worth noting that CRMDPs are sometimes referred to in the literature as Robust MDPs (RMDPs) or Distributionally Robust MDPs (DRMDPs). To distinguish them from the regularized robust framework, we adopt the term CR-MDPs. Similarly, RRMDPs appear under various names, including penalized robust MDPs (Yang et al., 2023), soft robust MDPs (Zhang et al., 2024), and robust $\phi$-regularized MDPs (Panaganti et al., 2024). We use the term RRMDPs to clearly differentiate them from CRMDPs while remaining consistent with the literature.

**Online RMDPs** Wang & Zou (2021); Badrinath & Kalathil (2021) studied the online learning for infinite-horizon RMDPs with $R$-contamination and more general uncertainty

sets, respectively. Their algorithmic design and theoretical analysis rely on assuming access to exploratory policies, which implicitly assumes that the nominal MDP is sufficiently exploratory. In contrast, we revisit this challenge from a different angle, focusing on the information deficit issue induced by distributional shift. Our work differs from theirs in two key aspects. First, we introduce a novel quantity to characterize the hardness of exploration in the nominal MDP and analyze it thoroughly via both upper and lower bounds on the sample complexity. Second, instead of assuming access to exploratory policies, we design algorithms that explicitly incorporate exploration strategies tailored for finite-horizon tabular CRMDPs and RRMDPs with $(s, a)$-rectangular uncertainty sets defined by general $f$-divergences.

Liu & Xu (2024a); Lu et al. (2024); Liu et al. (2024) focused on online robust RL under the specific setting of CRMDPs with uncertainty sets defined by the TV-distance, coupled with assumptions such as the existence of fail-states or vanishing minimal values. From an information-theoretic perspective, we show that these assumptions effectively circumvent the information deficit by constraining the direction of the distribution shift. In contrast, our work seeks to identify a more general sufficient condition for provably efficient online learning in CRMDPs—one that applies to arbitrary divergence-based uncertainty sets and does not rely on the fail-state or vanishing minimal value assumption.

**Off-dynamics RL** A substantial body of empirical work addresses off-dynamics RL through the lens of domain adaptation and transfer learning (Eysenbach et al., 2021; Desai et al., 2020; Zhang et al., 2021; Xu et al., 2023a; Wen et al., 2024; Guo et al., 2024; Wang et al., 2024b; Lyu et al., 2024; Da et al., 2025), among others. In this paper, we focus on the robust MDP (RMDP) formulation of off-dynamics RL. We refer readers to the above works for complementary approaches along this orthogonal line of research.

## 3. Preliminaries

**Constrained Robust MDP (CRMDP)** We denote a finite horizon CRMDP as CRMDP $(\mathcal{S}, \mathcal{A}, P^o, r, \mathcal{U}^\rho(P^o), H)$, where $\mathcal{S}$ is the state space, $\mathcal{A}$ is the action space, $P^o = \{P_h^o\}_{h=1}^H$ is the nominal transition kernel, $r : \mathcal{S} \times \mathcal{A} \to [0, 1]$ is the reward function, $\mathcal{U}^\rho(P^o)$ is the uncertainty set centered around the nominal kernel, $\rho$ is the uncertainty level, $H$ is the horizon length. In this work, we specifically focus on general $f$-divergence defined $(s, a)$-rectangular uncertainty sets (Iyengar, 2005), $\mathcal{U}^\rho(P^o) = \otimes_{(s,a,h) \in \mathcal{S} \times \mathcal{A} \times [H]} \mathcal{U}_h^\rho(s, a)$, where $\mathcal{U}_h^\rho(s, a) = \{P \in \Delta(\mathcal{S}) | D_f(P || P_h^o(\cdot|s, a)) \leq \rho\}$. The robust value

function and $Q$-function are defined as

$$V_h^{\pi,\rho}(s) = \inf_{P\in\mathcal{U}^\rho(P^o)} \mathbb{E}_{\pi,P}\left[\sum_{t=h}^{H} r_t(s_t,a_t)\,\Big|\, s_h = s\right],$$

$$Q_h^{\pi,\rho}(s,a) = \inf_{P\in\mathcal{U}^\rho(P^o)} \mathbb{E}_{\pi,P}\left[\sum_{t=h}^{H} r_t(s_t,a_t)\,\Big|\, s_h = s, a_h = a\right].$$

The optimal robust value function and optimal robust $Q$-function are defined as: $V_h^{\star,\rho}(s) = \sup_{\pi\in\Pi} V_h^{\pi,\rho}(s)$, $Q_h^{\star,\rho}(s,a) = \sup_{\pi\in\Pi} Q_h^{\pi,\rho}(s,a)$, where $\Pi$ is the set of all policies. Correspondingly, the optimal robust policy is the policy that achieves the optimal robust value function $\pi_h^\star = \arg\sup_{\pi\in\Pi} V_h^{\pi,\rho}(s)$. For CRMDPs, Iyengar (2005) proved the robust Bellman optimality equations

$$Q_h^{*,\rho}(s,a) = r_h(s,a) + \inf_{P_h\in\mathcal{U}_h^\rho(P^o)} \mathbb{E}_{P_h}\left[V_{h+1}^{*,\rho}\right](s,a),$$
$$V_h^{*,\rho}(s) = \max_{a\in\mathcal{A}} Q_h^{*,\rho}(s,a), \tag{3.1}$$

where $\mathbb{E}_{P_h}\left[V_{h+1}^{*,\rho}\right](s,a) := \mathbb{E}_{s'\sim P_h(\cdot|s,a)}\left[V_{h+1}^{*,\rho}(s')\right]$.

**Regularized Robust MDP (RRMDP)** A finite horizon RRMDP can be denoted as RRMDP$(\mathcal{S},\mathcal{A},P^o,r,\beta,\mathrm{R},H)$, where $\beta$ is the regularizer parameter, $\mathrm{R}$ is a penalty on distribution shift, and we set $\mathrm{R}$ to be the probability divergence $\mathrm{D}$ throughout this paper. RRMDPs replace the uncertainty set constraint in CRMDPs with a regularization term. Specifically, the robust value function and $Q$-function under the regularized setting are defined as

$$V_h^{\pi,\beta}(s) = \inf_{P\in\Delta(S)} \mathbb{E}_{\pi,P}\Bigg[\sum_{t=h}^{H} r_t(s_t,a_t)$$
$$+ \beta\cdot\mathrm{D}(P_t(\cdot|s_t,a_t),P_t^o(\cdot|s_t,a_t))\,\Big|\, s_h = s\Bigg],$$

$$Q_h^{\pi,\beta}(s,a) = \inf_{P\in\Delta(S)} \mathbb{E}_{\pi,P}\Bigg[\sum_{t=h}^{H} r_t(s_t,a_t)$$
$$+ \beta\cdot\mathrm{D}(P_t(\cdot|s_t,a_t),P_t^o(\cdot|s_t,a_t))\,\Big|\, s_h = s, a_h = a\Bigg].$$

For RRMDPs, Yang et al. (2023) showed the robust Bellman optimality equations:

$$Q_h^{*,\beta}(s,a) = r_h(s,a) + \inf_{P_h\in\Delta(S)}\Big[\mathbb{E}_{P_h}\left[V_{h+1}^{*,\beta}\right](s,a)$$
$$+ \beta\cdot\mathrm{D}(P_h(\cdot|s,a),P_h^o(\cdot|s,a))\Big], \tag{3.2}$$
$$V_h^{*,\beta}(s) = \max_{a\in\mathcal{A}} Q_h^{*,\beta}(s,a).$$

**Learning Goal** We have an agent actively interacting with the nominal environment for $K$ episodes to learn the optimal robust policy. At the start of episode $k$ with initial state $s_1^k$, the agent chooses a policy $\pi^k$ based on the history information. Then it interacts with the nominal environment by executing $\pi^k$ until the end of episode $k$, and collects a new trajectory. The agent's goal is to minimize the cumulative

regret after $K$ episodes, defined as

$$\mathrm{Regret}(K) = \sum_{k=1}^{K}\left[V_1^{*,\rho}(s_1^k) - V_1^{\pi^k,\rho}(s_1^k)\right] \text{ for CRMDPs,}$$

$$\mathrm{Regret}(K) = \sum_{k=1}^{K}\left[V_1^{*,\beta}(s_1^k) - V_1^{\pi^k,\beta}(s_1^k)\right] \text{ for RRMDPs.}$$

# 4. Online Robust Bellman Iteration (ORBIT)

In this section, we first present a meta-algorithm for online tabular RMDPs with general $f$-divergence defined uncertainty sets or regularization terms. We then instantiate the algorithm for CRMDPs with TV, KL and $\chi^2$-divergences defined uncertainty sets and RRMDPs with TV, KL and $\chi^2$-divergences defined regularization terms, respectively.

---

**Algorithm 1** Online Robust Bellman Iteration (ORBIT)

---

**Require:** uncertainty level $\rho > 0$ (for CRMDPs), or regularizer $\beta > 0$ (for RRMDPs).
1: **for** $k = 1,\cdots,K$ **do**
2:    $\widehat{V}_{H+1}^k(\cdot) \leftarrow 0$.
3:    **for** $h = H,\cdots,1$ **do**
4:       **for** $\forall (s,a)\in\mathcal{S}\times\mathcal{A}$ **do**
5:          Update $Q$-function estimation $\widehat{Q}_h^k(s,a)$
           CRMDP: refer to Section 4.2;
           RRMDP: refer to Section 4.3.
6:       **end for**
7:       **for** $\forall s\in\mathcal{S}$ **do**
8:          $\pi_h^k(s) \leftarrow \arg\max_{a\in\mathcal{A}}\widehat{Q}_h^k(s,a)$,
         $\widehat{V}_h^k(s) \leftarrow \max_{a\in\mathcal{A}}\widehat{Q}_h^k(s,a)$.
9:       **end for**
10:   **end for**
11:   Collect trajectory $\tau^k$ by executing $\pi^k$.
12:   Update $n_h^k, \widehat{r}_h^{k+1}, \widehat{P}_h^{k+1}$ according to (4.1).
13: **end for**

---

## 4.1. Algorithm Interpretation

We present our meta-algorithm, Online Robust Bellman Iteration (ORBIT), in Algorithm 1. The algorithm follows a value iteration framework and integrates optimistic estimation and the robust Bellman optimality equation in (3.1) and (3.2) for estimating the robust $Q$-functions. In each episode $k\in[K]$, ORBIT consists of two stages. In the first stage (Lines 3 to 10), Algorithm 1 iteratively updates the value function and $Q$-function estimations in a backward manner. In the second stage (Lines 11 to 12), we collect trajectory $\tau^k = (s_1^k, a_1^k, r_1^k,\cdots, s_H^k, a_H^k, r_H^k)$ by executing $\pi^k$. After a new trajectory is collected, ORBIT updates the empirical reward function and transition kernel as follows

$$n_h^k(s,a) = \sum_{i=1}^{k}\mathbb{1}\left\{s_h^i = s, a_h^i = a\right\},$$

$$\widehat{r}_h^{k+1}(s,a) = \frac{\sum_{i=1}^{k} r_h^i(s,a) \cdot \mathbb{1}\{s_h^i = s, a_h^i = a\}}{n_h^k(s,a) \vee 1}, \quad (4.1)$$

$$\widehat{P}_h^{k+1}(s'|s,a) = \frac{\sum_{i=1}^{k} \mathbb{1}\{s_h^i = s, a_h^i = a, s_{h+1}^i = s'\}}{n_h^k(s,a) \vee 1}.$$

Algorithm 1 updates $Q$-functions at each $(s,a)$ according to different RMDP settings and choices of $f$-divergences. We differentiate these cases using specific labels: CRMDP-TV, CRMDP-KL, CRMDP-$\chi^2$, RRMDP-TV, RRMDP-KL, and RRMDP-$\chi^2$. Finally, Algorithm 1 adopts the greedy policy of the estimated $Q$-function as the estimated optimal policy at episode $k$.

For the robust $Q$-function estimation, we leverage the robust optimality Bellman equation (3.1) and (3.2). Incorporating the optimism principle in the face of uncertainty (Abbasi-Yadkori et al., 2011) in the $Q$-function update, we have

$$\widehat{Q}_h^k(s,a) = \text{RB}_h^k(s,a) + b_h^k(s,a). \quad (4.2)$$

There are two components in (4.2): a robust Bellman estimator $\text{RB}_h^k(s,a)$ and a bonus term $b_h^k(s,a)$. Next, we will instantiate this meta-algorithm for CRMDPs and RRMDPs with various $f$-divergences, and provide explicit formulation for robust Bellman estimation and bonus design.

### 4.2. ORBIT under Constrained Robust MDPs

We first focus on CRMDPs and detail the update of robust $Q$-functions (4.2) in various settings. To solve the optimization problem in the robust Bellman equation (3.1) and (3.2), we resort to strong duality results in the following.

**CRMDP-TV**   In CRMDPs with TV-distance defined uncertainty sets, estimators in (4.2) are defined as follows

$$\text{RB}_h^k(s,a) = \widehat{r}_h^k(s,a) - \inf_{\eta \in [0,H]} \Big( \mathbb{E}_{\widehat{P}_h^k} \big[ (\eta - \widehat{V}_{h+1}^{k,\rho})_+ \big](s,a)$$
$$+ \rho \Big( \eta - \min_{s \in \mathcal{S}} \widehat{V}_{h+1}^{k,\rho}(s) \Big)_+ - \eta \Big), \quad (4.3)$$

$$b_h^k(s,a) = 2H\sqrt{\frac{2S^2 \ln(12SAH^2K^2/\delta)}{n_h^{k-1}(s,a) \vee 1}} + \frac{1}{K}, \quad (4.4)$$

where (4.3) represents the empirical version of the robust Bellman operator and (4.4) is the bonus. The dual formulation for TV-distance and the optimism of the estimated $Q$-function are established in Appendix C.1.

**CRMDP-KL**   In CRMDPs with KL-divergence defined uncertainty sets, estimators in (4.2) are defined as follows

$$\text{RB}_h^k(s,a) = \widehat{r}_h^k(s,a)$$
$$- \inf_{\nu \in [0, \frac{H}{\rho}]} \big( \nu \ln \mathbb{E}_{\widehat{P}_h^k} \big[ \exp\big( -\nu^{-1} \widehat{V}_{h+1}^{k,\rho} \big) \big](s,a) + \nu\rho \big), \quad (4.5)$$

$$b_h^k(s,a) = \left( 1 + \frac{2H\sqrt{S}}{\rho C_{MP}} \right) \sqrt{\frac{2\ln(2SAHK/\delta)}{n_h^{k-1}(s,a) \vee 1}}, \quad (4.6)$$

where $C_{MP}$ is defined in Assumption 5.7. The dual formulation for KL-divergence and the optimism of the estimated $Q$-function are proved in Appendix C.2.

**CRMDP-$\chi^2$**   For ORBIT in CRMDPs with $\chi^2$-divergence defined uncertainty sets, we have

$$\text{RB}_h^k(s,a) = \widehat{r}_h^k(s,a) + \sup_{\boldsymbol{\lambda} \in [0,H]} \Big( \mathbb{E}_{\widehat{P}_h^k} \big[ \widehat{V}_{h+1}^{k,\rho} - \boldsymbol{\lambda} \big](s,a)$$
$$- \sqrt{\rho \text{Var}_{\widehat{P}_h^k} (\widehat{V}_{h+1}^{k,\rho} - \boldsymbol{\lambda})} \Big), \quad (4.7)$$

$$b_h^k(s,a) = 3H\sqrt{\frac{2S^2 \ln(192SAH^3K^3/\delta)}{n_h^{k-1}(s,a) \vee 1}} + \frac{2}{K}. \quad (4.8)$$

The dual formulation for $\chi^2$-divergence and the optimism of the estimated $Q$-function are proved in Appendix C.3.

### 4.3. ORBIT under Regularized Robust MDPs

We then focus on RRMDPs and detail the update of robust $Q$-functions (4.2) in various settings.

**RRMDP-TV**   In RRMDPs with TV-distance regularization terms, estimators in (4.2) are defined as follows

$$\text{RB}_h^k(s,a) = \widehat{r}_h^k(s,a) - \mathbb{E}_{\widehat{P}_h^k} \Big[ \Big( \min_{s \in \mathcal{S}} \widehat{V}_{h+1}^{k,\beta}(s) + \beta$$
$$- \widehat{V}_{h+1}^{k,\beta}(s) \Big)_+ \Big](s,a) + \Big( \min_{s \in \mathcal{S}} \widehat{V}_{h+1}^{k,\beta}(s) + \beta \Big), \quad (4.9)$$

$$b_h^k(s,a) = 2H\sqrt{\frac{2S \ln(2SAHK/\delta)}{n_h^{k-1}(s,a) \vee 1}}. \quad (4.10)$$

The dual formulation for TV-distance and the optimism of the estimated $Q$-function are proved in Appendix D.1.

**RRMDP-KL**   For ORBIT in RRMDPs with KL-divergence regularization terms, we have

$$\text{RB}_h^k(s,a) = \widehat{r}_h^k(s,a) - \beta \ln \mathbb{E}_{\widehat{P}_h^k} \big[ \exp\big( -\beta^{-1} \widehat{V}_{h+1}^{k,\beta} \big) \big](s,a), \quad (4.11)$$

$$b_h^k(s,a) = \big( 1 + \beta e^{\beta^{-1}H} \sqrt{S} \big) \sqrt{\frac{2\ln(2SAHK/\delta)}{n_h^{k-1}(s,a) \vee 1}}. \quad (4.12)$$

The dual formulation for KL-divergence and the optimism of the estimated $Q$-function are provided in Appendix D.2.

**RRMDP-$\chi^2$**   For ORBIT in RRMDPs with $\chi^2$-divergence regularization terms, we have

$$\text{RB}_h^k(s,a) = \widehat{r}_h^k(s,a) + \sup_{\boldsymbol{\lambda} \in [0,H]} \Big( \mathbb{E}_{\widehat{P}_h^k} \big[ \widehat{V}_{h+1}^{k,\beta} - \boldsymbol{\lambda} \big](s,a)$$

$$- \frac{1}{4\beta} \mathrm{Var}_{\widehat{P}_h^k} \big[ \widehat{V}_{h+1}^{k,\beta} - \boldsymbol{\lambda} \big](s,a) \bigg), \qquad (4.13)$$

$$b_h^k(s,a) = 5H^2 \sqrt{\frac{2S^2 \ln(48SAH^3K^2/\delta)}{n_h^{k-1}(s,a) \vee 1}} + \frac{2}{K}, \quad (4.14)$$

The dual formulation for $\chi^2$-divergence and the optimism of the estimated $Q$-function are provided in Appendix D.3.

# 5. Theoretical Results

In this section, we provide theoretical understandings on the online learning of RMDPs. We start with a new perspective–the information deficit issue–to understand existing conditions for provably efficient online learning. Motivated by existing solutions to address the information deficit issue, we propose a new metric, the supremal visitation ratio, to quantify the hardness in exploration under online RMDPs. Further, we provide upper and lower bounds, involving the supremal visitation ratio, on the regret of Algorithm 1 in all settings.

## 5.1. Learnability of Online RMDPs

Focusing on CRMDPs with TV-distance defined uncertainty sets, Liu & Xu (2024a) and Lu et al. (2024) identified that the following assumptions can admit provably efficient online learning. In particular, Liu & Xu (2024a) made the following fail-states assumption.

**Condition 5.1** (Fail-states). (Liu & Xu, 2024a, Condition 4.3) There exists a subset $\mathcal{S}_f \subset \mathcal{S}$ of fail-states such that $r_h(s,a) = 0$, $P_h^o(\mathcal{S}_f|s,a) = 1, \forall\, (s,a,h) \in \mathcal{S}_f \times \mathcal{A} \times [H]$.

Lu et al. (2024) extended Condition 5.1 to Condition 5.2, but both assumptions essentially serve the same purpose, eliminating $\min_{s \in \mathcal{S}} V(s)$ in the dual formulation of the optimization problem in the CRMDP-TV setting.

**Condition 5.2** (Vanishing minimal value). (Lu et al., 2024, Assumption 4.1) The RMDP satisfies that $\min_{s \in \mathcal{S}} V_1^{*,\rho}(s) = 0$.

To explain the rationale behind Condition 5.1, we first define the visitation measure as follows.

**Definition 5.3** (Visitation measure). Under both CRMDPs and RRMDPs, for any policy $\pi$, we define the worst-case transition corresponding to $\pi$ as

$$P_h^{w,\pi}(\cdot|s,a) = \operatorname*{argmin}_{P_h \in \mathcal{U}_h^\rho(P_h^o)} \mathbb{E}_{P_h}[V_{h+1}^{\pi,\rho}](s,a) \text{ for CRMDPs,}$$

$$P_h^{w,\pi}(\cdot|s,a) = \operatorname*{argmin}_{P_h \in \Delta(\mathcal{S})} \mathbb{E}_{P_h}[V_{h+1}^{\pi,\beta}](s,a)$$
$$+ \beta \cdot \mathrm{D}(P_h(\cdot|s,a), P_h^o(\cdot|s,a)) \text{ for RRMDPs.}$$

At timestep $h \in [H]$, we denote $\mathrm{d}_h^\pi(\cdot)$ as the visitation measure on $\mathcal{S}$ induced by the policy $\pi$ under $P^o$, and $\mathrm{q}_h^\pi(\cdot)$

as the visitation measure on $\mathcal{S}$ induced by the policy $\pi$ under $P^{w,\pi}$. For convenience, we also write $P^{w,k} := P^{w,\pi^k}$, $\mathrm{d}^k := \mathrm{d}^{\pi^k}$ and $\mathrm{q}^k := \mathrm{q}^{\pi^k}$.

We show that Condition 5.1 implies the following property on the CRMDP.

**Proposition 5.4.** For CRMDPs with TV-distance defined uncertainty set satisfying Condition 5.1, for any $s \in \mathcal{S}$, $a \in \mathcal{A}$, $s' \in \mathcal{S} \backslash \mathcal{S}_f$ and policy $\pi$, we have $P_h^{w,\pi}(s'|s,a) \leq P_h^o(s'|s,a)$.

Proposition 5.4 shows that, for any state-action pair, the transition probability to a non-fail state is smaller in the worst-case environment than in the nominal environment. Consequently, non-fail states that are rarely visited in the nominal environment remain rarely visited in the worst-case environment, hence it would not incur large regret when making decisions at these states. Meanwhile, by definition, states in $\mathcal{S}_f$ lead to precisely zero value no matter what action is taken and thus no regret could be incurred at these states. This implies that Condition 5.1 or Condition 5.2 ensures the information obtained from exploration in the nominal environment is sufficient for decision making in the worst-case environment, thus bypassing the information deficit issue.

Note that both Condition 5.1 and Condition 5.2 are specifically designed for CRMDPs with TV-distance defined uncertainty sets. In more general $f$-divergence contexts, such as RMDPs with KL-divergence or $\chi^2$-divergence defined uncertainty sets or regularization terms, the property described in Proposition 5.4 does not hold. Consequently, learning RMDPs through online interaction is in general a challenging open problem (Lu et al., 2024) without additional assumptions. To characterize the inherent difficulty in learning online RMDPs with general $f$-divergences, we propose a more intrinsic metric that captures the essential of the problem, based on visitation measures in both the nominal and worst-case environments. Specifically, we have the following assumption.

**Assumption 5.5** (Bounded visitation measure ratio). Under the definition of Definition 5.3, we define $C_{vr} := \sup_{\pi,h,s} \frac{\mathrm{q}_h^\pi(s)}{\mathrm{d}_h^\pi(s)}$ as the supremal ratio between the nominal visitation measure and the worst-case visitation measure. We assume that $C_{vr}$ is polynomial in $H$, $S$ and $A$.

**Remark 5.6** (Reduction to non-robust setting). In non-robust settings, Assumption 5.5 is always satisfied with $C_{vr} = 1$, since $\mathrm{q}_h^\pi(s) = \mathrm{d}_h^\pi(s)$. This indicates our results also apply to the non-robust setting as a special case.

Visitation measure is a common metric in the offline literature such as Li et al. (2024, Definition 3) and Shi & Chi

(2024, Assumption 1). Assumption 5.5 ensures that the information obtained in the nominal environment can be effectively used for estimation in the worst-case environment.

### 5.2. Regret Bound for Constrained Robust MDP

We first focus on CRMDPs with TV, KL and $\chi^2$ divergences defined uncertainty sets. Before we present our results, we introduce an extra assumption for the CRMDP-KL setting.

**Assumption 5.7.** We assume there exists a constant $C_{MP} > 0$, such that for any $(h, s, a, s') \in [H] \times \mathcal{S} \times \mathcal{A} \times \mathcal{S}$, if $P_h^o(s'|s, a) > 0$, then $P_h^o(s'|s, a) > C_{MP}$.

**Remark 5.8.** For CRMDPs with KL-divergence defined uncertainty sets, Assumption 5.7 guarantees the regularity of dual formulation for KL-divergence. We note that similar assumptions also appear in Yang et al. (2022, Theorem 3.2) and Shi & Chi (2024, Theorem 3), both study CRMDPs with KL-divergence defined uncertainty sets.

**Theorem 5.9** (CRMDP regret upper bounds). Assume Assumption 5.5 holds for CRMDPs with TV, KL and $\chi^2$ divergence defined uncertainty sets. Assume Assumption 5.7 holds for the CRMDP-KL setting. Then for any $\delta \in (0, 1)$, with probability at least $1 - \delta$, Algorithm 1 satisfies

Regret$(K) =$

$$
\begin{cases}
\widetilde{\mathcal{O}}\big(C_{vr}S^2AH^2 + C_{vr}^{\frac{1}{2}}S^{\frac{3}{2}}A^{\frac{1}{2}}H^2\sqrt{K}\big) & \text{(TV)} \\
\widetilde{\mathcal{O}}\Big(\big(1 + \frac{H\sqrt{S}}{\rho C_{MP}}\big)\big(C_{vr}SAH + C_{vr}^{\frac{1}{2}}S^{\frac{1}{2}}A^{\frac{1}{2}}H\sqrt{K}\big)\Big) & \text{(KL)} \\
\widetilde{\mathcal{O}}\big(C_{vr}S^2AH^2 + C_{vr}^{\frac{1}{2}}S^{\frac{3}{2}}A^{\frac{1}{2}}H^2\sqrt{K}\big) & (\chi^2)
\end{cases}
$$

**Remark 5.10.** Theorem 5.9 presents the first provably sublinear result in the online RMDP literature for KL and $\chi^2$ defined uncertainty sets. The differences in dominant terms stem from how value function errors are amplified in dual formulations through Bellman equations and induction. Although the radius $\rho$ is not explicitly part of the results for TV-distance and $\chi^2$-divergence, it implicitly affects the bounds through $C_{vr}$ in Assumption 5.5. A larger $\rho$ loosens constraints, increases distribution shifts, and consequently, requires a higher $C_{vr}$ and leads to increased regret bound.

It is worth noting that under the policy selection scheme in Algorithm 1, our regret upper bounds still hold if we relax the definition of $C_{vr}$ in Assumption 5.5 to be defined as the supremum visitation ratio over deterministic policies.

By the standard online-to-batch conversion (Cesa-Bianchi et al., 2004), the regret bounds in Theorem 5.9 immediately imply the following sample complexity results:

**Corollary 5.11.** Under the same setup in Theorem 5.9, when

$$
K = \begin{cases}
\widetilde{\mathcal{O}}\big(C_{vr}S^3AH^4/\epsilon^2\big) & \text{(TV)} \\
\widetilde{\mathcal{O}}\Big(\big(1 + \frac{H\sqrt{S}}{\rho C_{MP}}\big)^2\big(C_{vr}SAH^2/\epsilon^2\big)\Big) & \text{(KL)} \\
\widetilde{\mathcal{O}}\big(C_{vr}S^3AH^4/\epsilon^2\big) & (\chi^2)
\end{cases}
$$

with probability at least $1 - \delta$, the uniform mixture of the policies produced by Algorithm 1 is $\epsilon$-optimal.

To see how tight the upper bounds in Theorem 5.9 are, we provide the following results on lower bounds.

**Theorem 5.12** (CRMDP regret lower bound). For CRMDPs with TV, KL and $\chi^2$ divergence defined uncertainty sets, for any learning algorithm $\xi$, there exists a CRMDP $\mathcal{M}$ satisfying Assumption 5.5, such that $\mathbb{E}\big[\text{Regret}_{\mathcal{M}}(\xi, K)\big] = \Omega\big(C_{vr}^{\frac{1}{2}}\sqrt{K}\big)$.

**Remark 5.13.** Comparing Theorem 5.9 and Theorem 5.12, we observe the order of $C_{vr}$ in the dominant terms of the upper bounds matches that in the lower bounds. The upper bounds thus align with the lower bounds in the two most critical parameters governing sample complexity: $C_{vr}$ and $K$. This indicates the fundamental presence of the information deficit issue in the online learning of robust policies, which stems from the discrepancy between the nominal and worst-case transitions and can be characterized by $C_{vr}$.

Based on the proof of Theorem 5.12, we construct hard instances to illustrate the necessity of Assumption 5.5 in guaranteeing sample efficient online learning in CRMDPs.

**Lemma 5.14** (CRMDP hard instances). For CRMDPs with TV, KL and $\chi^2$ divergence defined uncertainty sets, for any learning algorithm $\xi$, there exists a CRMDP $\mathcal{M}$ with $C_{vr} = 2^{2A}$, such that $\mathbb{E}\big[\text{Regret}_{\mathcal{M}}(\xi, K)\big] = \Omega\big(2^A\sqrt{K}\big)$.

**Remark 5.15.** Lemma 5.14 shows that, in the absence of additional assumptions, any online learning algorithm may perform poorly in CRMDPs. The hard instances are constructed by selecting a critical state that has an exponentially small visitation measure in the nominal environment, and make it has a visitation measure of constant order in the worst-case environment. As a result, an agent requires an exponential number of episodes to explore sufficient information about this state, while suffering a constant regret per episode when taking a suboptimal action.

### 5.3. Regret Bound for Regularized Robust MDP

We then focus on RRMDPs with TV, KL and $\chi^2$ divergences defined regularization.

**Theorem 5.16** (RRMDP regret upper bound). Assume Assumption 5.5 holds for RRMDPs with TV, KL and $\chi^2$ divergence defined regularization terms. Then for any $\delta \in (0, 1)$ with probability at least $1 - \delta$, Algorithm 1 satisfies

Regret$(K) =$

$$
\begin{cases}
\widetilde{\mathcal{O}}\big(C_{vr}S^{\frac{3}{2}}AH^2 + C_{vr}^{\frac{1}{2}}SA^{\frac{1}{2}}H^2\sqrt{K}\big) & \text{(TV)} \\
\widetilde{\mathcal{O}}\big((1 + \beta e^{\beta^{-1}H}\sqrt{S})\big(C_{vr}SAH + C_{vr}^{\frac{1}{2}}S^{\frac{1}{2}}A^{\frac{1}{2}}H\sqrt{K}\big)\big) & \text{(KL)} \\
\widetilde{\mathcal{O}}\big(C_{vr}S^2AH^3 + C_{vr}^{\frac{1}{2}}S^{\frac{3}{2}}A^{\frac{1}{2}}H^3\sqrt{K}\big) & (\chi^2)
\end{cases}
$$

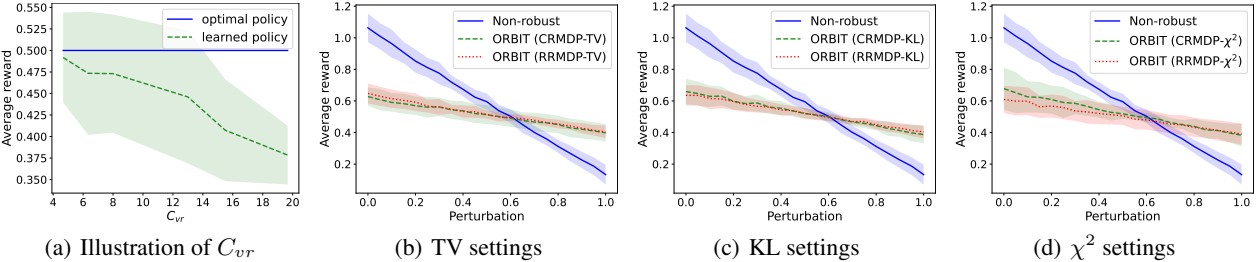

(a) Illustration of $C_{vr}$      (b) TV settings      (c) KL settings      (d) $\chi^2$ settings

*Figure 1.* Figure 1(a) shows the comparison of the learned policy and the optimal policy in Section 6.1 (Illustration of the Effect of $C_{vr}$ on Robustness), where the optimal policy represents the ground truth optimal policy, the learned policy is obtained by Algorithm 1. Figures 1(b) to 1(d) present the comparison between our algorithm ORBIT and the non-robust algorithm in Section 6.2 (Learning on Simulated RMDPs).

**Remark 5.17.** Theorem 5.16 presents the first provably sublinear results in the online RRMDP settings. The result in Theorem 5.16 corresponding to the TV-distance is smaller by a factor of $O(\sqrt{S})$ compared to Theorem 5.9. This efficiency gain arises because the dual formulation for RRMDPs eliminates the need for constructing an $\epsilon$-net, which also speeds up solving the inner optimization problem and demonstrates computational advantages. Notably, Assumption 5.7 is not required in the RRMDP-KL setting, as the dual formulation for KL has a closed-form solution, which also reduces the computational cost. In the $\chi^2$-divergence setting, the upper bound is larger than that in the constrained setting by a factor of $O(H)$. This gap derives from the differences in dual formulations in two RMDPs, where Lemma D.9 in the RRMDP-$\chi^2$ setting does not admit a square root compared to Lemma C.15 in the CRMDP-$\chi^2$ setting.

**Theorem 5.18** (RRMDP regret lower bound)**.** For RRMDPs with TV, KL and $\chi^2$ divergence defined regularization terms, for any learning algorithm $\xi$, there exists a RRMDP $\mathcal{M}$ satisfying Assumption 5.5, such that
$$\mathbb{E}\big[\text{Regret}_{\mathcal{M}}(\xi, K)\big] = \Omega\big(C_{vr}^{\frac{1}{2}}\sqrt{K}\big).$$

Comparing Theorem 5.16 and Theorem 5.18, we observe that the order of $C_{vr}$ in the dominant terms of upper bounds matches that in the lower bound. Together with the observation in Remark 5.13, we can conclude that $C_{vr}$ is a tight measure for evaluating exploration difficulty in RMDPs.

# 6. Experiments

In this section, we conduct numerical experiments to thoroughly verify the theoretical findings in previous sections. All numerical experiments were conducted on a server equipped with Intel(R) Xeon(R) Gold 5118 CPU @ 2.30GHz. The implementation of our ORBIT algorithm is available at https://github.com/panxulab/

`Online-Robust-Bellman-Iteration`.

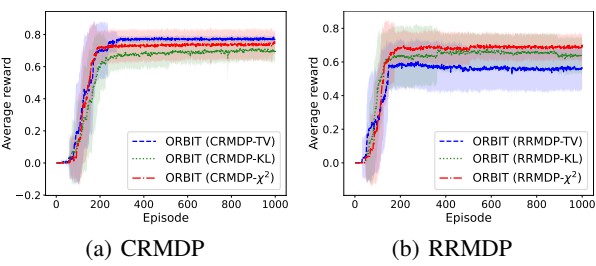

(a) CRMDP      (b) RRMDP

*Figure 2.* The convergence of ORBIT in Section 6.3 (Learning the Frozen Lake Problem). We use the policies obtained after each training episode to evaluate the convergence.

## 6.1. Illustration of the Effect of $C_{vr}$ on Robustness

The supremal visitation ratio $C_{vr}$ measures the difficulty in exploration. With a fixed number of episodes, our results Theorem 5.9 and Theorem 5.16 show that $C_{vr}$ would increase the sub-optimality gap of the learned policies. In this section, we construct a toy example (see Figure 4) with $H = 3$, $\mathcal{S} = \{s_0, \cdots, s_5\}$, and $\mathcal{A} = \{0, \cdots, 9\}$, focusing on the CRMDP-TV setting. More details about the environment can be found in Appendix A.1. The visitation measure of each states are influenced by a hyper-parameter $\beta$, and we can calculate that $C_{vr} = 3 + \frac{1}{\beta}$ in this case.

As we can see from Figure 1(a), when we increase $C_{vr}$ by decreasing $\beta$ in the nominal environment, it becomes harder for the agent to explore the nominal environment sufficiently to learn the optimal policy in the perturbed environment, and thus deteriorate the performance of the learned policy and enlarge the sub-optimality gap. This aligns well with our theoretical results.

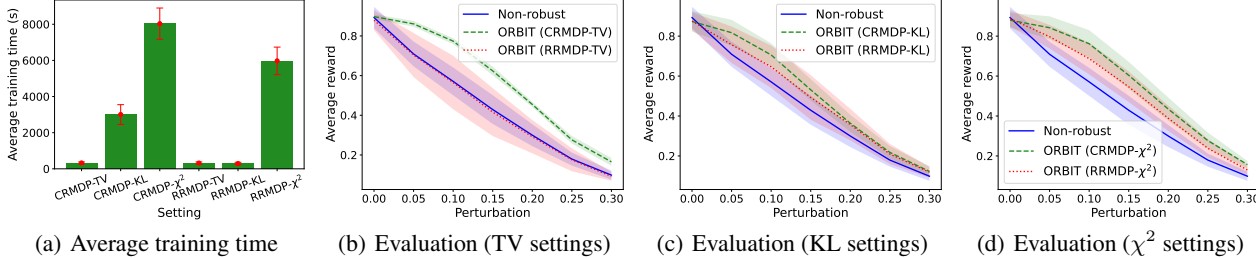

(a) Average training time     (b) Evaluation (TV settings)     (c) Evaluation (KL settings)     (d) Evaluation ($\chi^2$ settings)

*Figure 3.* Results in Section 6.3 (Learning the Frozen Lake Problem). Figure 3(a) presents the average time taken for training in various settings. Figures 3(b) to 3(d) present the comparison between our algorithm ORBIT and the non-robust algorithm. We use the last episode policy $\pi^K$ for the comparison.

## 6.2. Learning on Simulated RMDPs

Next, we design a simple MDP with learning horizon $H = 3$, the state space is $\mathcal{S} = \{s_0, \cdots, s_4\}$, and the action space is $\mathcal{A} = \{0, \cdots, 4\}$. The source environment and target environment are illustrated in Figures 5(a) and 5(b). More details about the environment can be found in Appendix A.2.

The experiment results are presented in Figures 1(b) to 1(d). We can see that the policies under CRMDPs perform similarly to their RRMDP counterpart. And compared to the non-robust algorithm, all robust policies are less sensitive to the environment perturbation. In particular, the performance of the non-robust algorithm drops drastically with respect to the perturbation. When the perturbation exceeds 0.6, all robust policies outperform the non-robust algorithm. This confirms the robustness of our proposed algorithm.

## 6.3. Learning the Frozen Lake Problem

Now we test our algorithm in a hard-to-explore setting, the Frozen Lake problem. In this scenario, the agent's objective is to traverse a frozen lake from the Start (S) to the Goal (G) without falling into any Holes (H), navigating over the Frozen (F) surface. A hyper-parameter $P_{\text{perturb}}$ is used to measure the perturbation in the test environment. More details about the environment can be found in Appendix A.3.

First, we evaluate the convergence of our algorithm by tracking the average reward throughout the training process in a single target environment with a fixed perturbation model, $P_{\text{perturb}}$. Specifically, we compute the average reward of the policy $\pi^k$ obtained after each episode $k$. As shown in Figures 2(a) and 2(b), our algorithm consistently converges by the end of training. The corresponding average training time is reported in Figure 3(a).

For RRMDPs with TV and KL divergence defined regularization terms, the dual formulations of the $Q$-functions ad-

mit closed-form solutions, simplifying the training process and resulting in lower computation complexity compared to CRMDPs. Lemma D.9 in the RRMDP-$\chi^2$ setting, on the other hand, requires solving optimization problems to get the dual formulations, leading to higher computational complexity than Lemmas C.1 and D.1 in both two RMDPs with TV-distance and Lemmas C.11 and D.5 in both two RMDPs with KL-divergence, though still lower than Lemma C.15 in the CRMDP-$\chi^2$ setting. Notably, the training time for the CRMDP-TV setting is not significantly higher than that of RRMDP-TV, as we incorporate an additional optimization algorithm (detailed in Algorithm 2) to accelerate computation. Also, due to our additional optimizations, increased exploration may result in longer training times.

We also evaluate the robustness of the policy $\pi^K$ after $K$ iterations of updates in various target environments with different $P_{\text{perturb}}$, by calculating the average reward obtained in each target environment. As shown in Figures 3(b) to 3(d), our robust algorithm outperforms the corresponding non-robust version in most cases.

## 7. Conclusion

We investigated online robust reinforcement learning within the context of tabular CRMDPs and RRMDPs, demonstrating that when the nominal MDP is sufficiently exploratory, sample-efficient online learning becomes feasible. We quantified the exploration efficiency of RMDPs through a novel quantity called the supremal visitation ratio. We constructed hard instances to show that a moderate supremal visitation ratio is necessary for ensuring sample-efficient online learning. We developed computationally efficient algorithms and provided regret analyses with both upper and lower bounds, which indicates our algorithm has an optimal dependency on the supremal visitation ratio and the number of episodes. We also conducted numerical experiments on diverse environments to validate our theory and show the robustness of our proposed algorithm.

## Impact Statement

This paper presents work whose goal is to advance the field of Machine Learning. There are many potential societal consequences of our work, none which we feel must be specifically highlighted here.

## Acknowledgements

W. Wang, Z. Liu and P. Xu are supported in part by the National Science Foundation (DMS-2323112) and the Whitehead Scholars Program at the Duke University School of Medicine.

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

# A. Additional Details on Experiments.

In this section, we provide more details about the experiments conducted in Section 6.

## A.1. More Details on Section 6.1 (Illustration of the Effect of $C_{vr}$ on Robustness)

**Construction**   We consider a simulated RMDP Figure 4(a) with horizon length $H = 3$, state space $\mathcal{S} = \{s_0, \cdots, s_5\}$, and action space $\mathcal{A} = \{0, \cdots, 9\}$. At each episode, the initial state is always $s_0$. The nominal transition at the first stage is independent of actions taken, $P^o(s_1|s_0) = 1 - P^o(s_2|s_0) = \beta$, where $\beta$ is a hyperparameter. At the second stage, if the current state is $s_2$, it will transit to $s_5$ with probability 1; if the current state is $s_1$, then we have $P^o(s_3|s_1, a = 0) = \frac{5}{6}$, $P^o(s_3|s_1, a) = \frac{1}{2}, \forall a \in \{1, \cdots, 9\}$ and $P^o(s_4|s_1, a) = 1 - P^o(s_3|s_1, a)$. Only $s_3$ and $s_5$ can generate reward, with $r(s_3, a) = 1, r(s_5, a) = \frac{1}{2}, \forall a \in \mathcal{A}$. By construction, the action taken at $s_1$ determines the final reward, and there are actually two kinds of actions: $\tilde{a} = 0$ if $a = 0$ and $\tilde{a} = 1$ if $a \in \mathcal{A}/0$. Though actions in $\mathcal{A}/0$ are equivalent, they are set to increase the harness in exploration. We construct a TV-distance defined uncertainty set with radius $\rho = \frac{1}{3}$.

It is easy to observe that, regardless of the policy $\pi$ chosen, $V^{\pi,\rho}(s_4) \leq V^{\pi,\rho}(s_3)$ and $V^{\pi,\rho}(s_1) \leq V^{\pi,\rho}(s_2)$, therefore the worst-case transition probability for any policy $\pi$ is $P^{w,\pi}(s_3|s_1, a = 0) = \frac{1}{2}$, $P^{w,\pi}(s_3|s_1, a) = \frac{1}{6}, \forall a \in \{1, \cdots, 9\}$ and $P^{w,\pi}(s_4|s_1, a) = 1 - P^{w,\pi}(s_3|s_1, a)$. Thus, $V^{\pi,\rho}(s_1) = \frac{1}{2} - \frac{\tilde{a}}{3} \leq \frac{1}{2} = V^{\pi,\rho}(s_2)$. Furthermore, the transition probability $P^{w,\pi}(s_1|s_0, a) = 1 - P^{w,\pi}(s_2|s_0, a) = \beta + \frac{1}{3}, \forall a$, where $\beta \in (0, \frac{2}{3})$ is a hyper-parameter. We can easily verify that all those transitions are within $[0, 1]$ and therefore well defined. With this analysis, we can calculate the visitation measure for each policy in Table 1 and derive $C_{vr} = 3 + \frac{1}{\beta}$.

**Implementation**   Under the optimal policy (taking $a = 0$ at $s = s_1$), the expected reward is $\mathbb{E}_{\pi^*}[r] = \frac{1}{2}$, regardless of $\beta$. We set the number of episodes $K = 1000$ to simulate a scenario with limited exploration and run Algorithm 1 in the CRMDP-TV setting. We test learned robust policies in the worst-case target environment and calculate the average reward among 2000 runs. The experimental results are based on 50 replications and plotted in Figure 1(a).

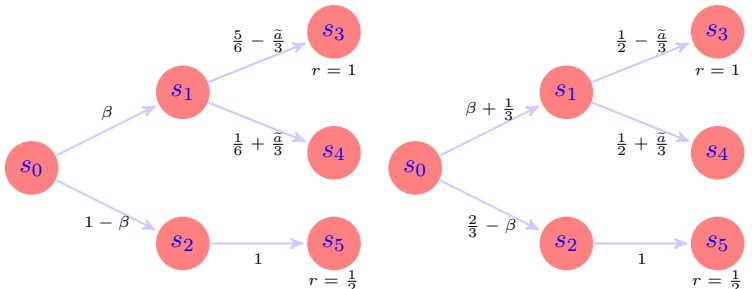

(a) The nominal MDP environment.   (b) The worst-case MDP environment.

*Figure 4.* The constructions of the nominal MDP and the worst-case MDP environments in Section 6.1.

*Table 1.* The visitation measure of each state in Section 6.1, the maximum of $\frac{q_h^\pi(s)}{d_h^\pi(s)}$ is achieved by taking $\tilde{a} = 0$ at $s_4$.

|  | $s_0$ | $s_1$ | $s_2$ | $s_3$ | $s_4$ | $s_5$ |
|---|---|---|---|---|---|---|
| $d_h^\pi(s)$ | 1 | $\beta$ | $1 - \beta$ | $(\frac{5}{6} - \frac{\tilde{a}}{3})\beta$ | $(\frac{1}{6} + \frac{\tilde{a}}{3})\beta$ | $1 - \beta$ |
| $q_h^\pi(s)$ | 1 | $\beta + \frac{1}{3}$ | $\frac{2}{3} - \beta$ | $(\frac{1}{2} - \frac{\tilde{a}}{3})(\beta + \frac{1}{3})$ | $(\frac{1}{2} + \frac{\tilde{a}}{3})(\beta + \frac{1}{3})$ | $\frac{2}{3} - \beta$ |

## A.2. More Details on Section 6.2 (Learning on Simulated RMDPs)

**Construction**   We consider a simple MDP Figure 5(a) as the source environment. The learning horizon $H = 3$, the state space is $\mathcal{S} = \{s_0, \cdots, s_4\}$, and the action space is $\mathcal{A} = \{0, \cdots, 4\}$. The initial state is always $s_0$, where it can transit to

$s_1$, $s_3$ and $s_4$ with probability $P^o(s_1|s_0, a) = 0.4 + \frac{a}{10}$, $P^o(s_3|s_0, a) = 0.1$ and $P^o(s_4|s_0, a) = 0.5 - \frac{a}{10}$ correspondingly. From $s_1$, it can transit to $s_2$ and $s_3$ with probability $P^o(s_2|s_1, a) = \frac{a}{10}$ and $P^o(s_3|s_1, a) = 1 - \frac{a}{10}$ by taking action $a$. From $s_2$, it can transit to $s_3$ and $s_4$ with probability $P^o(s_3|s_2, a) = 1 - \frac{a}{10}$ and $P^o(s_4|s_2, a) = \frac{a}{10}$ by taking action $a$. The $s_3$ and $s_4$ are absorbing states. The agent is rewarded $\frac{a}{20}$ by taking action $a$ at $s_0$, $s_1$ and $s_2$, rewarded 1 regardless of the action taken at $s_4$, and rewarded 0 regardless of the action taken at $s_3$.

The target environment Figure 5(b) is obtained by perturbing the first step in the source environment. To be specific, with perturbation rate $q$, the transition probability from $s_0$ to $s_3$ and $s_4$ is $P^w(s_3|s_0, a) = 0.1 + q \times (0.5 - \frac{a}{10})$ and $P^w(s_4|s_0, a) = (1 - q) \times (0.5 - \frac{a}{10})$, while $P^w(s_1|s_0, a) = 0.4 + \frac{a}{10}$ stays the same.

**Implementation**  We set $K = 1,000$ in Algorithm 1 and evaluate the learned policy in target environments with $q \in \{0, 0.05, 0.1, \cdots, 1\}$, respectively. In each target environment, the average reward among 500 runs is calculated for evaluation. All experimental results are based on 20 replications. The choice of uncertainty level $\rho$, regularizer $\beta$, and constant $c_{\text{bonus}}$ are provided in Table 2. For the non-robust algorithm, we simply set $\beta = 10000$ for Algorithm 1 instantiated with TV-distance defined regularization. It can be justified by Lemma D.1 that the extremely large regularization would not tolerate any perturbation, and thus the learned policy is basically the optimal policy under the source environment. The results are plotted in Figures 1(b) to 1(d).

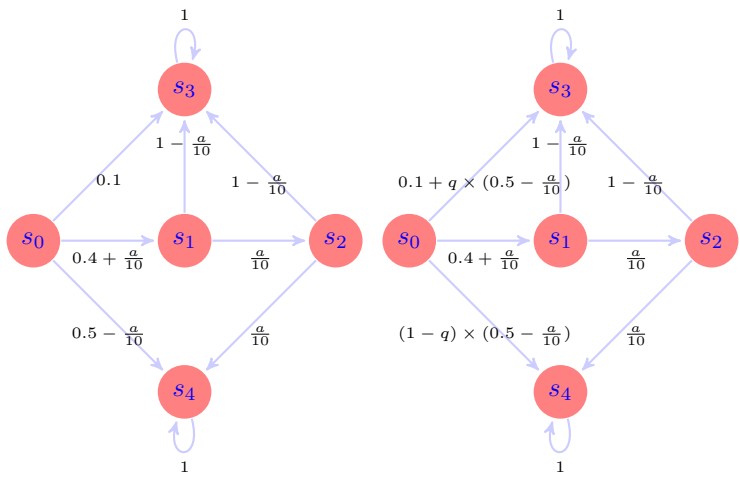

(a) The source RMDP environment.    (b) The target RMDP environment.

*Figure 5.* The source and target RMDP environments in Section 6.2, where the target environment is constructed by perturbing the first step.

### A.3. More Details on Section 6.3 (Learning the Frozen Lake Problem)

**Construction**  In this scenario, the agent's objective is to traverse a frozen lake from the Start (S) to the Goal (G) without falling into any Holes (H), navigating over the Frozen (F) surface. The agent's movement is influenced by a hyper-parameter $P_{\text{slip}} = 0.1$, which determines the probability of the agent successfully moving in the intended direction. Specifically, the agent moves towards the intended direction with a probability of $1 - P_{\text{slip}} = 0.9$, and with probability $P_{\text{slip}}/2 = 0.05$, it will veer off in either perpendicular direction. The Goal state is an absorbing state, once reached the agent will stay there. The agent earns a reward of 1 if and only if it is at the goal (G) at step $H - 1$. For evaluation, after the agent selects an intended action, with a probability of $P_{\text{perturb}}$, the agent actually takes action towards the opposite direction instead. The difficulty of this environment arises from two sources: 1) it is a sparse reward MDP, and 2) the influence onto the movement in the source environment makes the exploration of the goal state very hard.

**Implementation**  We use the default map in the OpenAI Gym library, which is illustrated in Example A.1, and set $H = 25$ and $K = 1,000$ in Algorithm 1. The hyperparameter $\rho$ in the constrained setting, $\beta$ in the regularized setting, and $c_{\text{bonus}}$ are

tuned from $\{0.001, 0.003, 0.01, 0.03, 0.1, 0.3, 1\}$, with the final choice presented in Table 2. All experimental results are based on 20 replications.

We implement several optimizations to accelerate the algorithm. First, we reuse the counts $n_h^k(s, a)$ from (4.1) across different steps, as the transition dynamics remain consistent throughout. Additionally, we observe that the inner optimization process, which is the primary bottleneck of the algorithm, does not need to be fully recalculated at each episode. For instance, if the agent falls into a hole after the first step, the information about the environment remains largely unchanged. To take advantage of this, we create a map where the keys are hashes of the optimization parameters and the values are the corresponding results, allowing us to reuse prior computations efficiently. We also applied Algorithm 2 to obtain the result in the CRMDP-TV setting, which can directly be derived from definition.

We assess the convergence of our algorithms by calculating the average reward among 500 runs in the single target environment with $P_{\text{perturb}} = 0.1$, of the policy $\pi^k$ obtained after each episode $k$ during the training. The convergence results are plotted in Figures 2(a) and 2(b), and the average training time is plotted in Figure 3(a). We also evaluate the robustness of the policy $\pi^K$ after $K$ episodes across various target environments with $P_{\text{perturb}} \in \{0, 0.05, 0.1, \cdots, 0.3\}$. For each target environment, we compute the average reward among 500 runs, and the results are shown in Figures 3(b) to 3(d).

**Example A.1** (Illustration of Frozen Lake environment). The environment of the Frozen Lake problem is illustrated as follows, where S denotes "Start", G denotes "Goal", H denotes "Hole" and F denotes "Frozen".

$$\begin{bmatrix} S & F & F & F & F & F & F & F \\ F & F & F & F & F & F & F & F \\ F & F & F & H & F & F & F & F \\ F & F & F & F & F & H & F & F \\ F & F & F & H & F & F & F & F \\ F & H & H & F & F & F & H & F \\ F & H & F & F & H & F & H & F \\ F & F & F & H & F & F & F & G \end{bmatrix}$$

### A.4. A More Computationally Efficient Solver for the CRMDP-TV Setting

As shown in Lemma C.1, the update formulation of Algorithm 1 in the CRMDP-TV setting involves solving an optimization problem in its dual formulation. To reduce the computational complexity, we introduce Algorithm 2, which simplifies this procedure.

We explain the rationale behind Algorithm 2 as follows. The original formulation of the CRMDP-TV problem is given by $Q_h(s, a) = r_h(s, a) + \inf\limits_{\text{TV}(P \| P_h^o) \leq \rho} \mathbb{E}_P[V_{h+1}](s, a)$. It is easy to see that the worst-case scenario is reached when the transition probabilities for states with the highest value functions are reduced by a total of $\rho$, and those for states with the lowest value functions are increased by $\rho$. This greedy approach avoids the need to solve the optimization problem for $\eta$ as described in Lemma C.1.

### A.5. Hyper-parameters for Experiments in Section 6

Here, we provide the hyper-parameters used in training in the experiments section. Note that we reformulate the bonus term as $c_{\text{bonus}}/\sqrt{K}$ in practical experiments.

---

**Algorithm 2** A more efficient solver for the CRMDP-TV setting

---

**Require:** robust set radius $\rho > 0$, transition array $P[S]$, value function array $V[S]$ ($S > 1$)

1: args = argsort($V$).
2: $V = V[\text{args}], P = P[\text{args}]$.
3: $\text{pnt}_0 = 1, \text{pnt}_1 = S$.
4: $\text{rho}_0 = 0, \text{rho}_1 = 0$.
5: **while** $\text{rho}_0 < \rho$ **do**
6:     tmp = $\min(\rho - \text{rho}_0, 1 - P[\text{pnt}_0])$.
7:     $P[\text{pnt}_0] = P[\text{pnt}_0] + \text{tmp}$.
8:     $\text{rho}_0 = \text{rho}_0 + \text{tmp}$.
9:     $\text{pnt}_0 = \text{pnt}_0 + 1$.
10: **end while**
11: **while** $\text{rho}_1 < \rho$ **do**
12:     tmp = $\min(\rho - \text{rho}_1, P[\text{pnt}_1])$.
13:     $P[\text{pnt}_1] = P[\text{pnt}_1] - \text{tmp}$.
14:     $\text{rho}_1 = \text{rho}_1 + \text{tmp}$.
15:     $\text{pnt}_1 = \text{pnt}_1 - 1$.
16: **end while**
17: Output: $\sum_{i=1}^{S} P[i] * V[i]$.

---

*Table 2.* hyper-parameters for Section 6.2 (Learning on Simulated RMDPs)

| Setting | $\rho$ **or** $\beta$ | $c_{\text{bonus}}$ |
|---|---|---|
| non-robust | – | 1 |
| constrained TV | 0.5 | 1 |
| constrained KL | 0.5 | 1 |
| constrained $\chi^2$ | 1 | 1 |
| regularized TV | 0.1 | 1 |
| regularized KL | 0.1 | 1 |
| regularized $\chi^2$ | 0.1 | 1 |

# B. Proof of Proposition 5.4

*Proof of Proposition 5.4.* We prove it by contradiction. We assume that there exists $s^*$, $a^*$ and $s'$ such that $V_{h+1}^{\pi,\rho}(s') > 0$ and $P_h^{w,\pi}(s'|s^*, a^*) > P_h^o(s'|s^*, a^*)$. We pick $\widetilde{s} \in \mathcal{S}_f$ arbitrarily and consider the following transition measure $P_h'$:

$$P_h'(s|s^*, a^*) = \begin{cases} P_h^{w,\pi}(s|s^*, a^*) & s \notin \{s', \widetilde{s}\}, \\ P_h^o(s'|s^*, a^*) & s = s', \\ P_h^{w,\pi}(\widetilde{s}|s^*, a^*) + P_h^{w,\pi}(s'|s^*, a^*) - P_h^o(s'|s^*, a^*) & s = \widetilde{s}. \end{cases}$$

It is easy to verify that $P_h' \in \Delta(\mathcal{S})$, $\text{TV}(P_h' \| P_h^o) \leq \text{TV}(P_h^{w,\pi} \| P_h^o)$, therefore $P_h'$ is a valid transition measure in the transition uncertainty set. Based on the definition of fail-states, we have $V_{h+1}^{\pi,\rho}(\widetilde{s}) = 0 < V_{h+1}^{\pi,\rho}(s')$ and thus $\mathbb{E}_{P_h'}[V_{h+1}^{\pi,\rho}] < \mathbb{E}_{P_h^{w,\pi}}[V_{h+1}^{\pi,\rho}]$, which contradicts the fact that $P_h^{w,\pi}$ is the worst-case transition. $\square$

# C. Proofs of Results in Constrained RMDPs

## C.1. Proof of Theorem 5.9 (Constrained TV Setting)

Before proving this theorem, we first present several technical lemmas that will be useful in the proof.

We first give the closed form solution of constrained TV update formulation. This dual formulation has also been proved by

*Table 3.* hyper-parameters for Section 6.3 (Learning the Frozen Lake Problem)

| Setting | $\rho$ or $\beta$ | $c_{\text{bonus}}$ |
|---|---|---|
| non-robust | – | 0.001 |
| constrained TV | 0.15 | 0.001 |
| constrained KL | 0.15 | 0.01 |
| constrained $\chi^2$ | 0.5 | 0.01 |
| regularized TV | 0.1 | 0.003 |
| regularized KL | 0.1 | 0.001 |
| regularized $\chi^2$ | 0.05 | 0.01 |

Iyengar (2005, Lemma 4.3), but the formulation of our result is slightly different from theirs. Note that Lu et al. (2024) used the same formulation, but their expression is incorrect by a factor of $\frac{1}{2}$. This error arises because they directly cited Yang et al. (2022), which employed the $L_1$ distance rather than the TV-distance. So we prove it again for the sake of completeness.

**Lemma C.1** (Dual formulation). For the optimization problem $Q_h(s,a) = r_h(s,a) + \inf_{\text{TV}(P \| P_h^o) \leq \rho} \mathbb{E}_P[V_{h+1}](s,a)$, we have its dual formulation as follows

$$Q_h = r_h - \inf_{\eta \in [0,H]} \left( \mathbb{E}_{P_h^o}[(\eta - V_{h+1}(s))_+] + \rho \left( \eta - \min_{s \in \mathcal{S}} V_{h+1}(s) \right)_+ - \eta \right). \tag{C.1}$$

*Proof.* Consider the optimization problem

$$Q_h = r_h + \inf_{\text{TV}(P \| P_h^o) \leq \rho} \mathbb{E}_P[V_{h+1}] = r_h + \inf_{\text{TV}(P \| P_h^o) \leq \rho} \sum_{s \in \mathcal{S}} P(s) V_{h+1}(s).$$

We denote $\varphi(t) = |t - 1|/2$, then the Lagrangian can be written as

$$\mathcal{L}(P, \eta) = \sum_{s \in S} P(s) V_{h+1}(s) + \nu \left( \sum_{s \in S} P_h^o(s) \varphi \left( \frac{P(s)}{P_h^o(s)} \right) - \rho \right) - \sum_{s \in \mathcal{S}} \lambda(s) P(s) + \eta \left( 1 - \sum_{s \in S} P(s) \right).$$

We denote $g(s) = P(s)/P_h^o(s)$, then we have

$$\inf_P \mathcal{L}(P, \eta) = -\sup_g \sum_{s \in \mathcal{S}} P_h^o(s)[g(s)[(\lambda(s) + \eta) - V_{h+1}(s)] - \nu \varphi(g(s))] - \nu\rho + \eta$$

$$= -\nu \mathbb{E}_{P_h^o} \left[ \varphi^* \left( \frac{(\lambda(s) + \eta) - V_{h+1}(s)}{\nu} \right) \right] - \nu\rho + \eta,$$

where the second equation is from the definition of dual function $\varphi^*(y) = \sup_x (y^\top x - \varphi(x))$. From $f$-divergence literature (Xu et al., 2023b), we know that for TV-distance,

$$\varphi^*(s) = \begin{cases} -\frac{1}{2} & \text{for } s \leq -\frac{1}{2}; \\ s & \text{for } -\frac{1}{2} < s \leq \frac{1}{2}; \\ +\infty & \text{for } s > \frac{1}{2}. \end{cases} \tag{C.2}$$

So we have

$$Q_h = r_h + \sup_{\nu \geq 0, \lambda \geq 0, \eta} \left( \inf_P \mathcal{L}(P, \eta) \right)$$

$$= r_h - \inf_{\nu \geq 0, \lambda \geq 0, \eta} \left( \nu \mathbb{E}_{P_h^o} \left[ \varphi^* \left( \frac{(\lambda(s) + \eta) - V_{h+1}(s)}{\nu} \right) \right] + \nu\rho - \eta \right)$$

$$= r_h - \inf_{\nu \geq 0, \lambda \geq 0, \eta, \frac{\lambda(s) + \eta - V_{h+1}(s)}{\nu} \leq \frac{1}{2}} \left( \nu \mathbb{E}_{P_h^o} \left[ \max \left( \frac{(\lambda(s) + \eta) - V_{h+1}(s)}{\nu}, -\frac{1}{2} \right) \right] + \nu\rho - \eta \right) \tag{C.3}$$

$$
= r_h - \inf_{\nu \geq 0, \lambda \geq 0, \eta, \frac{\lambda(s)+\eta-V_{h+1}(s)}{\nu} \leq \frac{1}{2}} \left( \mathbb{E}_{P_h^o} \left[ \left( (\lambda(s) + \eta) - V_{h+1}(s) + \frac{\nu}{2} \right)_+ \right] + \nu\rho - \eta - \frac{\nu}{2} \right)
$$

$$
= r_h - \inf_{\nu \geq 0, \lambda \geq 0, \eta', \nu \geq \lambda(s)+\eta'-V_{h+1}(s)} \left( \mathbb{E}_{P_h^o} [(\lambda(s) + \eta' - V_{h+1}(s))_+] + \nu\rho - \eta' \right) \tag{C.4}
$$

$$
= r_h - \inf_{\lambda \geq 0, \eta'} \left( \mathbb{E}_{P_h^o} [(\lambda(s) + \eta' - V_{h+1}(s))_+] + \rho \max_{s \in \mathcal{S}} (\lambda(s) + \eta' - V_{h+1}(s))_+ - \eta' \right) \tag{C.5}
$$

$$
= r_h - \inf_{\eta'} \left( \mathbb{E}_{P_h^o} [(\eta' - V_{h+1}(s))_+] + \rho \left( \eta' - \min_{s \in \mathcal{S}} V_{h+1}(s) \right)_+ - \eta' \right) \tag{C.6}
$$

$$
= r_h - \inf_{\eta' \in [0,H]} \left( \mathbb{E}_{P_h^o} [(\eta' - V_{h+1}(s))_+] + \rho \left( \eta' - \min_{s \in \mathcal{S}} V_{h+1}(s) \right)_+ - \eta' \right), \tag{C.7}
$$

where (C.3) follows from the definition of $\varphi^*$ (C.2), we redefine $\eta' = \eta + \frac{\nu}{2}$ in (C.4), (C.5) holds because the result increases monotonically with respect to $\nu$ thus the minimum value is attained at $\nu = \max_{s \in \mathcal{S}} (\lambda(s) + \eta' - V_{h+1}(s))_+$, (C.6) holds because the result increases monotonically with respect to $\lambda$, (C.7) holds because the result increases monotonically with respect to $\eta'$ when $\eta' \leq 0$ and increases monotonically with respect to $\eta'$ when $\eta' \geq H$. $\square$

In the next lemma, we prove the optimism of estimation $Q^k$, which helps control the estimation error $Q^* - Q^k$.

**Lemma C.2** (Optimism). If we set the bonus term as follows

$$
\text{bonus}_h^k(s, a) = 2H \sqrt{\frac{2S^2 \ln(12SAH^2K^2/\delta)}{n_h^{k-1}(s, a) \vee 1}} + \frac{1}{K}, \tag{C.8}
$$

then for any policy $\pi$ and any $(k, h, s, a) \in [K] \times [H] \times \mathcal{S} \times \mathcal{A}$, with probability at least $1 - 2\delta$, we have $Q_h^{k,\rho}(s, a) \geq Q_h^{\pi,\rho}(s, a)$. Specially, by setting $\pi = \pi^*$, we have $Q_h^{k,\rho}(s, a) \geq Q_h^{\pi^*,\rho}(s, a)$.

*Proof.* We prove this by induction. First, when $h = H + 1$, $Q_{H+1}^{k,\rho}(s, a) = 0 = Q_{H+1}^{\pi,\rho}(s, a)$ holds trivially.

Assume $Q_{h+1}^{k,\rho}(s, a) \geq Q_{h+1}^{\pi,\rho}(s, a)$ holds, since $\pi^k$ is the greedy policy, we have

$$
V_{h+1}^{k,\rho}(s) = Q_{h+1}^{k,\rho}(s, \pi_{h+1}^k(s)) \geq Q_{h+1}^{k,\rho}(s, \pi_{h+1}(s)) \geq Q_{h+1}^{\pi,\rho}(s, \pi_{h+1}(s)) = V_{h+1}^{\pi,\rho}(s),
$$

where the first inequality is because we choose $\pi^k$ as the greedy policy.

Recall that we denote $Q_h^{k,\rho}$ as the optimistic estimation in $k$-th episode, that is,

$$
Q_h^{k,\rho}(s, a) = \text{bonus}_h^k(s, a) + \hat{r}_h^k(s, a) + \inf_{\text{TV}(P \| \hat{P}_h^k) \leq \rho} \mathbb{E}_P \left[ V_{h+1}^{k,\rho} \right](s, a).
$$

We can infer that

$$
Q_h^{k,\rho} - Q_h^{\pi,\rho} = \text{bonus}_h^k + \hat{r}_h^k + \inf_{\text{TV}(P \| \hat{P}_h^k) \leq \rho} \mathbb{E}_P \left[ V_{h+1}^{k,\rho} \right] - r_h - \inf_{\text{TV}(P \| P_h^o) \leq \rho} \mathbb{E}_P \left[ V_{h+1}^{\pi,\rho} \right]
$$

$$
= \text{bonus}_h^k + \hat{r}_h^k - r_h + \inf_{\text{TV}(P \| \hat{P}_h^k) \leq \rho} \mathbb{E}_P \left[ V_{h+1}^{k,\rho} \right] - \inf_{\text{TV}(P \| P_h^o) \leq \rho} \mathbb{E}_P \left[ V_{h+1}^{k,\rho} \right]
$$

$$
+ \inf_{\text{TV}(P \| P_h^o) \leq \rho} \mathbb{E}_P \left[ V_{h+1}^{k,\rho} \right] - \inf_{\text{TV}(P \| P_h^o) \leq \rho} \mathbb{E}_P \left[ V_{h+1}^{\pi,\rho} \right]
$$

$$
\geq \text{bonus}_h^k + \hat{r}_h^k - r_h + \inf_{\text{TV}(P \| \hat{P}_h^k) \leq \rho} \mathbb{E}_P \left[ V_{h+1}^{k,\rho} \right] - \inf_{\text{TV}(P \| P_h^o) \leq \rho} \mathbb{E}_P \left[ V_{h+1}^{k,\rho} \right] \tag{C.9}
$$

$$
= \text{bonus}_h^k + \hat{r}_h^k - r_h - \inf_{\eta \in [0,H]} \left( \mathbb{E}_{\hat{P}_h^k} \left[ (\eta - V_{h+1}^{k,\rho}(s))_+ \right] + \rho \left( \eta - \min_{s \in \mathcal{S}} V_{h+1}^{k,\rho}(s) \right)_+ - \eta \right)
$$

$$
+ \inf_{\eta \in [0,H]} \left( \mathbb{E}_{P_h^o} \left[ (\eta - V_{h+1}^{k,\rho}(s))_+ \right] + \rho \left( \eta - \min_{s \in \mathcal{S}} V_{h+1}^{k,\rho}(s) \right)_+ - \eta \right) \tag{C.10}
$$

$$
\begin{aligned}
&\geq \text{bonus}_h^k + \widehat{r}_h^k - r_h + \inf_{\eta \in [0,H]} \left\{ \left( \mathbb{E}_{P_h^o}\left[(\eta - V_{h+1}^{k,\rho}(s))_+\right] + \rho\left(\eta - \min_{s \in \mathcal{S}} V_{h+1}^{k,\rho}(s)\right)_+ - \eta \right) \right. \\
&\qquad\qquad\qquad\quad \left. - \left( \mathbb{E}_{\widehat{P}_h^k}\left[(\eta - V_{h+1}^{k,\rho}(s))_+\right] + \rho\left(\eta - \min_{s \in \mathcal{S}} V_{h+1}^{k,\rho}(s)\right)_+ - \eta \right) \right\} \\
&= \text{bonus}_h^k + \widehat{r}_h^k - r_h + \inf_{\eta \in [0,H]} \left\{ \mathbb{E}_{P_h^o}\left[(\eta - V_{h+1}^{k,\rho}(s))_+\right] - \mathbb{E}_{\widehat{P}_h^k}\left[(\eta - V_{h+1}^{k,\rho}(s))_+\right] \right\} \\
&\geq \text{bonus}_h^k - \underbrace{\left| \widehat{r}_h^k - r_h \right|}_{(i)} - \underbrace{\sup_{\eta \in [0,H]} \left| \mathbb{E}_{P_h^o}\left[(\eta - V_{h+1}^{k,\rho}(s))_+\right] - \mathbb{E}_{\widehat{P}_h^k}\left[(\eta - V_{h+1}^{k,\rho}(s))_+\right] \right|}_{(ii)},
\end{aligned}
\tag{C.11}
$$

where (C.9) is from the induction assumption, we plug in the dual formulation (C.1) in (C.10).

For term (i) in (C.11), from Lemma G.1 and a union bound, with probability at least $1 - \delta$, we have

$$
\left| \widehat{r}_h^k(s,a) - r_h(s,a) \right| \leq \sqrt{\frac{\ln(2SAHK/\delta)}{2n_h^{k-1}(s,a) \vee 1}},
\tag{C.12}
$$

for any $(k, h, s, a) \in [K] \times [H] \times \mathcal{S} \times \mathcal{A}$.

We denote $V(\eta) = \left(\eta - V_{h+1}^{k,\rho}(s)\right)_+ \in [0, H]$ and $\mathcal{V} = \left\{ V \in \mathbb{R}^S : \|V\|_\infty \leq H \right\}$. To bound term (ii) in (C.11), we create a $\epsilon$-net $\mathcal{N}_\mathcal{V}(\epsilon)$ for $\mathcal{V}$. From Lemma G.4, it holds that $\ln |\mathcal{N}_\mathcal{V}(\epsilon)| \leq |S| \cdot \ln(3H/\epsilon)$.

Therefore, by the definition of $\mathcal{N}_\mathcal{V}(\epsilon)$, for any fixed $V$, there exists a $V' \in \mathcal{N}_\mathcal{V}(\epsilon)$ such that $\|V - V'\|_\infty \leq \epsilon$, that is

$$
\begin{aligned}
\left| \mathbb{E}_{P_h^o}[V] - \mathbb{E}_{\widehat{P}_h^k}[V] \right| &\leq \left| \mathbb{E}_{P_h^o}[V] - \mathbb{E}_{P_h^o}[V'] \right| + \left| \mathbb{E}_{P_h^o}[V'] - \mathbb{E}_{\widehat{P}_h^k}[V'] \right| + \left| \mathbb{E}_{\widehat{P}_h^k}[V'] - \mathbb{E}_{\widehat{P}_h^k}[V] \right| \\
&\leq \|P_h^o\|_1 \|V - V'\|_\infty + \left| \mathbb{E}_{P_h^o}[V'] - \mathbb{E}_{\widehat{P}_h^k}[V'] \right| + \left\| \widehat{P}_h^k \right\|_1 \|V - V'\|_\infty \\
&\leq \sup_{V' \in \mathcal{N}_\mathcal{V}(\epsilon)} \left| \mathbb{E}_{P_h^o}[V'] - \mathbb{E}_{\widehat{P}_h^k}[V'] \right| + 2\epsilon,
\end{aligned}
\tag{C.13}
$$

where the second inequality follows from the Holder's inequality.

For any fixed $V$, we apply Lemma G.3 and have

$$
\left| \mathbb{E}_{P_h^o}[V] - \mathbb{E}_{\widehat{P}_h^k}[V] \right| \leq \left\| P_h^o - \widehat{P}_h^k \right\|_1 \cdot \|V\|_\infty \leq H\sqrt{\frac{2S\ln(2/\delta)}{n_h^{k-1} \vee 1}},
\tag{C.14}
$$

with probability at least $1 - \delta$.

Then with probability at least $1 - \delta$, we have

$$
\begin{aligned}
\sup_{\eta \in [0,H]} \left| \mathbb{E}_{P_h^o}\left[(\eta - V_{h+1}^{k,\rho}(s))_+\right] - \mathbb{E}_{\widehat{P}_h^k}\left[(\eta - V_{h+1}^{k,\rho}(s))_+\right] \right| &\leq \sup_{\eta \in [0,H]} \left| \mathbb{E}_{P_h^o}[V(\eta)] - \mathbb{E}_{\widehat{P}_h^k}[V(\eta)] \right| \\
&\leq \sup_{V \in \mathcal{N}_\mathcal{V}(\epsilon)} \left| \mathbb{E}_{P_h^o}[V] - \mathbb{E}_{\widehat{P}_h^k}[V] \right| + 2\epsilon \tag{C.15} \\
&\leq H\sqrt{\frac{2S\ln(2SAHK|\mathcal{N}_\mathcal{V}(\epsilon)|/\delta)}{n_h^{k-1} \vee 1}} + 2\epsilon \tag{C.16} \\
&\leq H\sqrt{\frac{2S^2\ln(6SAH^2K/\epsilon\delta)}{n_h^{k-1} \vee 1}} + 2\epsilon \\
&= H\sqrt{\frac{2S^2\ln(12SAH^2K^2/\delta)}{n_h^{k-1} \vee 1}} + \frac{1}{K}. \tag{C.17}
\end{aligned}
$$

for any $(k, h, s, a) \in [K] \times [H] \times \mathcal{S} \times \mathcal{A}$, where (C.15) follows from (C.13), (C.16) is from (C.14) and a union bound, we set $\epsilon = 1/2K$ in (C.17).

Apply the union bound again and combine (C.11) with (C.12), (C.17), the definition of bonus and induction assumption. With probability at least $1 - 2\delta$, we have $Q_h^{k,\rho}(s, a) \geq Q_h^{\pi,\rho}(s, a)$ for any $(k, h, s, a) \in [K] \times [H] \times \mathcal{S} \times \mathcal{A}$. This completes the proof. $\qquad\square$

**Lemma C.3.** For any $k \in [K]$, we can bound the estimation error as follows

$$Q_1^{k,\rho} - Q_1^{\pi^k,\rho} \leq \sum_{h=1}^{H} \mathbb{E}_{\{P_h^{w,k}\}_{h=1}^H, \pi^k} \left[ 2\text{bonus}_h^k \right].$$

*Proof.* From the proof of Lemma C.2, we see that

$$\left| \widehat{r}_h^k(s, a) - r_h(s, a) \right| + \left| \inf_{\text{TV}(P\|\widehat{P}_h^k)\leq\rho} \mathbb{E}_P\left[V_{h+1}^{k,\rho}\right](s, a) - \inf_{\text{TV}(P\|P_h^o)\leq\rho} \mathbb{E}_P\left[V_{h+1}^{k,\rho}\right](s, a) \right| \leq \text{bonus}_h^k(s, a). \tag{C.18}$$

Recall that we define $P_h^{w,k} = \underset{\text{TV}(P\|P_h^o)\leq\rho}{\arg\min} \mathbb{E}_P\left[V_{h+1}^{\pi^k,\rho}\right]$ as the worst-case transition in Definition 5.3, we have

$$Q_h^{k,\rho} - Q_h^{\pi^k,\rho} = \text{bonus}_h^k + \widehat{r}_h^k + \inf_{\text{TV}(P\|\widehat{P}_h^k)\leq\rho} \mathbb{E}_P\left[V_{h+1}^{k,\rho}\right] - r_h - \inf_{\text{TV}(P\|P_h^o)\leq\rho} \mathbb{E}_P\left[V_{h+1}^{\pi^k,\rho}\right]$$

$$= \text{bonus}_h^k + \widehat{r}_h^k - r_h + \inf_{\text{TV}(P\|\widehat{P}_h^k)\leq\rho} \mathbb{E}_P\left[V_{h+1}^{k,\rho}\right] - \inf_{\text{TV}(P\|P_h^o)\leq\rho} \mathbb{E}_P\left[V_{h+1}^{k,\rho}\right]$$

$$\quad + \inf_{\text{TV}(P\|P_h^o)\leq\rho} \mathbb{E}_P\left[V_{h+1}^{k,\rho}\right] - \inf_{\text{TV}(P\|P_h^o)\leq\rho} \mathbb{E}_P\left[V_{h+1}^{\pi^k,\rho}\right]$$

$$\leq 2\text{bonus}_h^k + \inf_{\text{TV}(P\|P_h^o)\leq\rho} \mathbb{E}_P\left[V_{h+1}^{k,\rho}\right] - \inf_{\text{TV}(P\|P_h^o)\leq\rho} \mathbb{E}_P\left[V_{h+1}^{\pi^k,\rho}\right] \tag{C.19}$$

$$\leq 2\text{bonus}_h^k + \mathbb{E}_{P_h^{w,k}}\left[V_{h+1}^{k,\rho} - V_{h+1}^{\pi^k,\rho}\right] \tag{C.20}$$

$$= 2\text{bonus}_h^k + \mathbb{E}_{P_h^{w,k}, \pi^k}\left[Q_{h+1}^{k,\rho} - Q_{h+1}^{\pi^k,\rho}\right], \tag{C.21}$$

where the (C.19) holds because of (C.18), (C.20) and (C.21) use the definition of $P_h^{w,k}$ and $\pi^k$ accordingly. Apply (C.21) recursively, we can obtain the result. $\qquad\square$

Next, in Lemma C.9, we provide an upper bound on the sum of the expectations of $\sqrt{\frac{1}{n_h^{k-1}\vee 1}}$ under the worst-case environment. In order to prove this lemma, we follow a similar procedure to that of Zanette & Brunskill (2019), whose setting differs from ours as we consider the non-stationary dynamics.

**Lemma C.4.** (Failure Events) We define the following failure events:

$$F_k = \left\{ \exists s, a, h : n_h^{k-1}(s, a) \leq \frac{1}{2} \sum_{i<k} \text{d}_h^i(s, a) - \ln\left(\frac{SAHK}{\delta}\right) \right\}.$$

Then we have $P\left( \bigcup_{k=1}^{K} F_k \right) \leq 1 - \delta$.

*Proof.* Consider a fixed $s \in \mathcal{S}, a \in \mathcal{A}, h \in [H]$. We define $\mathcal{F}_k$ to be the $\sigma$-field induced by the first $k-1$ episodes and $X_k$ to be the indicator whether $(s, a)$ was visited in episode $k$ at step $h$. The probability $\mathbb{P}(s = s_h^k, a = a_h^k | \pi_k)$ of whether $X_k = 1$ is $\mathcal{F}_k$-measurable, therefore we can apply Lemma G.5 with $W = \ln(\frac{SAHK}{\delta})$. The proof is finished by applying a union bound over $s, a, h, k$. $\qquad\square$

**Definition C.5.** (The Good Set) We define

$$L_k = \left\{ (s, a, h) : \frac{1}{4} \sum_{i<k} \text{d}_h^i(s, a) \geq \ln\left(\frac{SAHK}{\delta}\right) + 1 \right\}.$$

**Lemma C.6.** (Visitation Ratio) Outside the failure event, if $(s, a, h) \in L_k$, we have

$$n_h^{k-1}(s, a) \geq \frac{1}{4} \sum_{i \leq k} d_h^i(s, a).$$

*Proof.* Outside the failure events defined in Lemma C.4, we have

$$
\begin{aligned}
n_h^{k-1}(s, a) &> \frac{1}{2} \sum_{i < k} d_h^i(s, a) - \ln\left(\frac{SAHK}{\delta}\right) \\
&= \frac{1}{4} \sum_{i < k} d_h^i(s, a) + \frac{1}{4} \sum_{i < k} d_h^i(s, a) - \ln\left(\frac{SAHK}{\delta}\right) \\
&\geq \frac{1}{4} \sum_{i < k} d_h^i(s, a) + 1 \geq \frac{1}{4} \sum_{i < k} d_h^i(s, a) + d_h^k(s, a) \geq \frac{1}{4} \sum_{i \leq k} d_h^i(s, a).
\end{aligned}
$$

where the second inequality uses $(s, a, h) \in L_k$ and the definition of $L_k$ in Definition C.5. $\qquad\square$

**Lemma C.7.** (Minimal Contribution) Outside the failure event, we have

$$\sum_{k=1}^{K} \sum_{(s,a,h) \notin L_k} d_h^k(s, a) = \widetilde{\mathcal{O}}(SAH).$$

*Proof.* We have

$$
\begin{aligned}
\sum_{k=1}^{K} \sum_{(s,a,h) \notin L_k} d_h^k(s, a) &= \sum_{(s,a,h)} \sum_{k=1}^{K} d_h^k(s, a) \mathbb{1}\{(s, a, h) \notin L_k\} \\
&\leq \sum_{(s,a,h)} \left( \sum_{k < K} d_h^k(s, a) \mathbb{1}\{(s, a, h) \notin L_k\} + 1 \right) \\
&< \sum_{(s,a,h)} \left( 4 \ln\left(\frac{SAHK}{\delta}\right) + 5 \right) \\
&= \widetilde{\mathcal{O}}(SAH).
\end{aligned}
$$

where the first inequality uses the definition of $L_k$ in Definition C.5. $\qquad\square$

**Lemma C.8.** (Visitation Ratio) Outside the failure event, it holds that

$$\sum_{k=1}^{K} \sum_{(s,a,h) \in L_k} \frac{d_h^k(s, a)}{n_h^{k-1}(s, a)} = \widetilde{\mathcal{O}}(SAH).$$

*Proof.* Outside the failure events defined in Lemma C.4, we have

$$
\begin{aligned}
\sum_{k=1}^{K} \sum_{(s,a,h) \in L_k} \frac{d_h^k(s, a)}{n_h^{k-1}(s, a)} &= \sum_{k=1}^{K} \sum_{(s,a,h)} \frac{d_h^k(s, a)}{n_h^{k-1}(s, a)} \mathbb{1}\{(s, a, h) \in L_k\} \\
&\leq 4 \sum_{k=1}^{K} \sum_{(s,a,h)} \frac{d_h^k(s, a)}{\sum_{i \leq k} d_h^i(s, a)} \mathbb{1}\{(s, a, h) \in L_k\},
\end{aligned}
$$

where the inequality follows from Lemma C.6.

Next, for any fixed $(h, s, a) \in L_k$ for some $k_0$, since $\sum\limits_{i<k} \mathrm{d}_h^i(s, a)$ is strictly increasing with $k$, there exists a critical episode $\widetilde{k} \leq k_0$ such that $(h, s, a) \in L_k$ holds for all $k \geq \widetilde{k}$ and $(h, s, a) \notin L_k$ holds for all $k < \widetilde{k}$. From the definition of $\widetilde{k}$ and Definition C.5, we know $\sum\limits_{i<\widetilde{k}} \mathrm{d}_h^i(s, a) \geq 4\ln(\frac{SAHK}{\delta}) + 4 > 4$. Therefore,

$$\sum_{k=1}^{K} \frac{\mathrm{d}_h^k(s, a)}{\sum\limits_{i\leq k} \mathrm{d}_h^i(s, a)} \mathbb{1}\{(s, a, h) \in L_k\} = \sum_{k=1}^{K} \frac{\mathrm{d}_h^k(s, a)}{\sum\limits_{i<\widetilde{k}} \mathrm{d}_h^i(s, a) + \sum\limits_{\widetilde{k}\leq i\leq k} \mathrm{d}_h^i(s, a)} \mathbb{1}\{(s, a, h) \in L_k\}$$

$$\leq \sum_{k=1}^{K} \frac{\mathrm{d}_h^k(s, a)}{4 + \sum\limits_{\widetilde{k}\leq i\leq k} \mathrm{d}_h^i(s, a)} \mathbb{1}\{(s, a, h) \in L_k\}$$

$$\leq \sum_{\widetilde{k}\leq k\leq K} \frac{\mathrm{d}_h^k(s, a)}{4 + \sum\limits_{\widetilde{k}\leq i\leq k} \mathrm{d}_h^i(s, a)},$$

where the third inequality comes from the definition of $\widetilde{k}$.

To simplify the notations, we define $v_1 = \mathrm{d}_h^{\widetilde{k}}(s, a), v_2 = \mathrm{d}_h^{\widetilde{k}+1}(s, a), \cdots, v_{K-\widetilde{k}+1} = \mathrm{d}_h^K(s, a)$. And in order to bound the above summation, we also define the functions $F(x) = \sum\limits_{i=1}^{\lfloor x \rfloor} v_i + v_{\lceil x \rceil}(x - \lfloor x \rfloor)$ and $f(x) = v_{\lceil x \rceil}$. It is easy to verify that the derivative of $F(x)$ is $f(x)$. Then we write

$$\sum_{\widetilde{k}\leq k\leq K} \frac{\mathrm{d}_h^k(s, a)}{4 + \sum\limits_{\widetilde{k}\leq i\leq k} \mathrm{d}_h^i(s, a)} = \sum_{k=1}^{K-\widetilde{k}+1} \frac{v_k}{4 + \sum\limits_{i=1}^{k} v_i} = \sum_{k=1}^{K-\widetilde{k}+1} \frac{f(k)}{4 + F(k)}.$$

Additionally, we have that $F(x) \leq \sum\limits_{i=1}^{\lfloor x \rfloor} v_i + v_{\lceil x \rceil}(\lceil x \rceil - \lfloor x \rfloor) = \sum\limits_{i=1}^{\lceil x \rceil} v_i = F(\lceil x \rceil)$ and $f(x) = f(\lceil x \rceil)$. Then, we have

$$\sum_{k=1}^{K-\widetilde{k}+1} \frac{f(k)}{4 + F(k)} = \int_0^{K-\widetilde{k}+1} \frac{f(\lceil x \rceil)}{4 + F(\lceil x \rceil)}\, \mathrm{d}x$$

$$\leq \int_0^{K-\widetilde{k}+1} \frac{f(x)}{4 + F(x)}\, \mathrm{d}x$$

$$= \ln(4 + F(K - \widetilde{k} + 1)) - \ln(4 + F(0)) \leq \widetilde{\mathcal{O}}(\ln(K)).$$

We obtain the result by summing over all the $(s, a, h)$ pairs. □

**Lemma C.9.** Outside the failure event, it holds that

$$\sum_{k=1}^{K}\sum_{h=1}^{H} \mathbb{E}_{P_h^{w,k}, \pi^k}\left[\sqrt{\frac{1}{n_h^{k-1}(s_t^k, \pi_k(s_t^k)) \vee 1}}\right] = \widetilde{\mathcal{O}}\left(\sqrt{C_{vr}SAH^2K} + C_{vr}SAH\right).$$

*Proof.* Outside the failure event, we can calculate as follows

$$\sum_{k=1}^{K}\sum_{h=1}^{H} \mathbb{E}_{P_h^{w,k}, \pi^k}\left[\sqrt{\frac{1}{n_h^{k-1}(s, a) \vee 1}}\right]$$

$$= \sum_{k=1}^{K}\sum_{(h,s,a)} \mathrm{q}_h^k(s, a)\sqrt{\frac{1}{n_h^{k-1}(s, a) \vee 1}}$$

$$\leq \sum_{k=1}^{K} \sum_{(h,s,a) \in L_k} q_h^k(s,a) \sqrt{\frac{1}{n_h^{k-1}(s,a) \vee 1}} + \sum_{k=1}^{K} \sum_{(h,s,a) \notin L_k} q_h^k(s,a) \tag{C.22}$$

$$= \sum_{k=1}^{K} \sum_{(h,s,a) \in L_k} q_h^k(s,a) \sqrt{\frac{1}{n_h^{k-1}(s,a)}} + \sum_{k=1}^{K} \sum_{(h,s,a) \notin L_k} q_h^k(s,a) \tag{C.23}$$

$$\leq \sum_{k=1}^{K} \sum_{(h,s,a) \in L_k} q_h^k(s,a) \sqrt{\frac{1}{n_h^{k-1}(s,a)}} + C_{vr} \sum_{k=1}^{K} \sum_{(h,s,a) \notin L_k} d_h^k(s,a) \tag{C.24}$$

$$\leq \sum_{k=1}^{K} \sum_{(h,s,a) \in L_k} q_h^k(s,a) \sqrt{\frac{1}{n_h^{k-1}(s,a)}} + \widetilde{\mathcal{O}}(C_{vr} SAH) \tag{C.25}$$

$$\leq \sqrt{\sum_{k=1}^{K} \sum_{(h,s,a)} q_h^k(s,a)} \sqrt{\sum_{k=1}^{K} \sum_{(h,s,a) \in L_k} \frac{q_h^k(s,a)}{n_h^{k-1}(s,a)}} + \widetilde{\mathcal{O}}(C_{vr} SAH) \tag{C.26}$$

$$\leq \sqrt{\sum_{k=1}^{K} \sum_{(h,s,a)} q_h^k(s,a)} \sqrt{C_{vr} \sum_{k=1}^{K} \sum_{(h,s,a) \in L_k} \frac{d_h^k(s,a)}{n_h^{k-1}(s,a)}} + \widetilde{\mathcal{O}}(C_{vr} SAH) \tag{C.27}$$

$$\leq \sqrt{KH} \cdot \sqrt{\widetilde{\mathcal{O}}(C_{vr} SAH)} + \widetilde{\mathcal{O}}(C_{vr} SAH) \tag{C.28}$$

$$= \widetilde{\mathcal{O}}(\sqrt{C_{vr} SAH^2 K} + C_{vr} SAH),$$

where (C.22) decomposes the summation into two parts and makes use of the fact that $\sqrt{\frac{1}{n_h^{k-1}(s,a) \vee 1}} \leq 1$, (C.23) holds because $n_h^{k-1}(s,a) \geq \ln(\frac{SAHK}{\delta}) + 1 \geq 1$ by combining Definition C.5 and Lemma C.6, (C.24) and (C.27) are from our assumption Assumption 5.5, (C.25) and (C.28) are from Lemma C.7 and Lemma C.8 accordingly, and (C.26) is the Cauchy-Schwartz inequality. $\qquad \square$

We are now ready to prove the main theorem that establishes the regret bound of ORBIT in the CRMDP-TV setting.

**Theorem C.10** (Restatement of Theorem 5.9 in TV-distance setting)**.** For CRMDP with $(s,a)$-rectangular TV-distance defined uncertainty set satisfying Assumption 5.5, with probability at least $1 - \delta$, the regret of Algorithm 1 satisfies

$$\text{Regret} = \widetilde{\mathcal{O}}(C_{vr} S^2 AH^2 + C_{vr}^{\frac{1}{2}} S^{\frac{3}{2}} A^{\frac{1}{2}} H^2 \sqrt{K}).$$

*Proof.* Setting $\delta' = \delta/3$ in Lemmas C.2 and C.4, then with probability at least $1 - \delta$, we get

$$\text{Regret} = \sum_{k=1}^{K} (V_1^{*,\rho} - V_1^{\pi^k,\rho})$$

$$= \sum_{k=1}^{K} (V_1^{*,\rho} - V_1^{k,\rho}) + \sum_{k=1}^{K} (V_1^{k,\rho} - V_1^{\pi^k,\rho})$$

$$\leq \sum_{k=1}^{K} \sum_{h=1}^{H} \mathbb{E}_{P_h^{w,k},\pi^k} [2\text{bonus}_h^k] \tag{C.29}$$

$$= 2 \sum_{k=1}^{K} \sum_{h=1}^{H} \mathbb{E}_{P_h^{w,k},\pi^k} \left[ 2H \sqrt{\frac{2S^2 \ln(12SAH^2K^2/\delta)}{n_h^{k-1}(s,a) \vee 1}} + \frac{1}{K} \right] \tag{C.30}$$

$$= \widetilde{\mathcal{O}}(C_{vr} S^2 AH^2 + C_{vr}^{\frac{1}{2}} S^{\frac{3}{2}} A^{\frac{1}{2}} H^2 \sqrt{K}), \tag{C.31}$$

where (C.29) is the combination of Lemma C.2 and Lemma C.3, we plug in the bonus (C.8) in (C.30), (C.31) is from Lemma C.9. $\qquad \square$

## C.2. Proof of Theorem 5.9 (Constrained KL Setting)

We first give the closed form solution of constrained KL update formulation. This dual formulation has also been proved by Iyengar (2005, Lemma 4.1), but the rage of $\nu$ in our result is more precise compared to theirs.

**Lemma C.11** (Dual formulation). For the optimization problem $Q_h(s,a) = r_h(s,a) + \inf\limits_{\mathrm{KL}(P\|P_h^o)\leq\rho} \mathbb{E}_P[V_{h+1}](s,a)$, we have its dual formulation as follows

$$Q_h = r_h - \inf_{\nu\in[0,\frac{H}{\rho}]} \left(\nu \ln \mathbb{E}_{P_h^o}\left[e^{-\nu^{-1}V_{h+1}}\right] + \nu\rho\right). \tag{C.32}$$

*Proof.* Consider the optimization problem

$$Q_h = r_h + \inf_{\mathrm{KL}(P\|P_h^o)\leq\rho} \mathbb{E}_P[V_{h+1}] = r_h + \inf_{\mathrm{KL}(P\|P_h^o)\leq\rho} \sum_{s\in\mathcal{S}} P(s)V_{h+1}(s).$$

The Lagrangian can be written as

$$\mathcal{L}(P,\eta) = \sum_{s\in S} P(s)V_{h+1}(s) + \nu\left(\sum_{s\in S} P(s)\ln\left(\frac{P(s)}{P_h^o(s)}\right) - \rho\right) - \sum_{s\in\mathcal{S}}\lambda(s)P(s) + \eta\left(1 - \sum_{s\in S} P(s)\right).$$

We set the derivative of $\mathcal{L}$ w.r.t. $P(s)$ to zero

$$\frac{\partial\mathcal{L}}{\partial P(s)} = V_{h+1}(s) + \nu\ln\left(\frac{P(s)}{P_h^o(s)}\right) + \nu - [\lambda(s) + \eta] = 0. \tag{C.33}$$

We denote $P'$ as the worst-case transition that satisfies (C.33), then we have

$$P'(s) = P_h^o(s)e^{-\nu^{-1}V_{h+1}(s)+\nu^{-1}[\lambda(s)+\eta]-1},$$

$$\inf_P \mathcal{L}(P,\eta) = -\nu\mathbb{E}_{P_h^o}\left[e^{-\nu^{-1}V_{h+1}(s)+\nu^{-1}[\lambda(s)+\eta]-1}\right] - \nu\rho + \eta.$$

Therefore,

$$\begin{aligned} Q_h &= r_h + \sup_{\nu\geq0,\lambda\geq0,\eta}\left(\inf_P \mathcal{L}(P,\eta)\right) \\ &= r_h - \inf_{\nu\geq0,\lambda\geq0,\eta}\left(\nu\mathbb{E}_{P_h^o}\left[e^{-\nu^{-1}V_{h+1}(s)+\nu^{-1}[\lambda(s)+\eta]-1}\right] + \nu\rho - \eta\right) \\ &= r_h - \inf_{\nu\geq0,\eta}\left(\nu\mathbb{E}_{P_h^o}\left[e^{-\nu^{-1}V_{h+1}(s)+\nu^{-1}\eta-1}\right] + \nu\rho - \eta\right) \tag{C.34} \\ &= r_h - \inf_{\nu\geq0}\left(\nu\ln\mathbb{E}_{P_h^o}\left[e^{-\nu^{-1}V_{h+1}}\right] + \nu\rho\right). \tag{C.35} \end{aligned}$$

where (C.34) holds because the result increases monotonically with respect to $\lambda$, (C.35) holds by calculating the derivation with respect to $\eta$ and thus setting $\eta = -\nu\left(\ln\mathbb{E}_{P_h^o}\left[e^{-\nu^{-1}V_{h+1}}\right] - 1\right)$.

We denote $\widetilde{\nu} = \operatorname*{argmin}_{\nu\geq0}\left(\nu\ln\mathbb{E}_{P_h^o}\left[e^{-\nu^{-1}V_{h+1}}\right] + \nu\rho\right)$, from the strong duality, it is easy to infer that

$$\widetilde{\nu}\ln\mathbb{E}_{P_h^o}\left[e^{-\widetilde{\nu}^{-1}V_{h+1}}\right] + \widetilde{\nu}\rho \leq 0. \tag{C.36}$$

And since $V_{h+1} \leq H$, we have

$$\widetilde{\nu}\ln\mathbb{E}_{P_h^o}\left[e^{-\widetilde{\nu}^{-1}V_{h+1}}\right] \geq -H. \tag{C.37}$$

Combine (C.36) and (C.37) into (C.35), we have $\widetilde{\nu} \in \left[0,\frac{H}{\rho}\right]$. That is,

$$\begin{aligned} Q_h &= r_h - \inf_{\nu\geq0}\left(\nu\ln\mathbb{E}_{P_h^o}\left[e^{-\nu^{-1}V_{h+1}}\right] + \nu\rho\right) \\ &= r_h - \inf_{\nu\in[0,\frac{H}{\rho}]}\left(\nu\ln\mathbb{E}_{P_h^o}\left[e^{-\nu^{-1}V_{h+1}}\right] + \nu\rho\right). \end{aligned}$$

This finishes the proof. $\qquad\square$

Similar to Lemma C.2, we prove the optimism of estimation $Q^k$ and control $Q^* - Q^k$.

**Lemma C.12** (Optimism). If we set the bonus term as follows

$$\text{bonus}_h^k(s, a) = \left(1 + \frac{2H\sqrt{S}}{\rho C_{MP}}\right)\sqrt{\frac{2\ln(2SAHK/\delta)}{n_h^{k-1}(s, a) \vee 1}}, \tag{C.38}$$

then for any policy $\pi$ and any $(k, h, s, a) \in [K] \times [H] \times \mathcal{S} \times \mathcal{A}$, with probability at least $1 - 2\delta$, we have $Q_h^{k,\rho}(s, a) \geq Q_h^{\pi,\rho}(s, a)$. Specially, by setting $\pi = \pi^*$, we have $Q_h^{k,\rho}(s, a) \geq Q_h^{\pi^*,\rho}(s, a)$.

*Proof.* We prove this by induction. First, when $h = H + 1$, $Q_{H+1}^{k,\rho}(s, a) = 0 = Q_{H+1}^{\pi,\rho}(s, a)$ holds trivially.

Assume $Q_{h+1}^{k,\rho}(s, a) \geq Q_{h+1}^{\pi,\rho}(s, a)$ holds, since $\pi^k$ is the greedy policy, we have

$$V_{h+1}^{k,\rho}(s) = Q_{h+1}^{k,\rho}(s, \pi_{h+1}^k(s)) \geq Q_{h+1}^{k,\rho}(s, \pi_{h+1}(s)) \geq Q_{h+1}^{\pi,\rho}(s, \pi_{h+1}(s)) = V_{h+1}^{\pi,\rho}(s),$$

where the first inequality is because we choose $\pi^k$ as the greedy policy.

Recall that we denote $Q_h^{k,\rho}$ as the optimistic estimation in $k$-th episode, that is,

$$Q_h^{k,\rho}(s, a) = \text{bonus}_h^k(s, a) + \widehat{r}_h^k(s, a) + \inf_{\text{KL}(P\|\widehat{P}_h^k) \leq \rho} \mathbb{E}_P\big[V_{h+1}^{k,\rho}\big](s, a).$$

We can infer that

$$\begin{aligned}
Q_h^{k,\rho} - Q_h^{\pi,\rho} &= \text{bonus}_h^k + \widehat{r}_h^k + \inf_{\text{KL}(P\|\widehat{P}_h^k) \leq \rho} \mathbb{E}_P\big[V_{h+1}^{k,\rho}\big] - r_h - \inf_{\text{KL}(P\|P_h^o) \leq \rho} \mathbb{E}_P\big[V_{h+1}^{\pi,\rho}\big] \\
&= \text{bonus}_h^k + \widehat{r}_h^k - r_h + \inf_{\text{KL}(P\|\widehat{P}_h^k) \leq \rho} \mathbb{E}_P\big[V_{h+1}^{k,\rho}\big] - \inf_{\text{KL}(P\|P_h^o) \leq \rho} \mathbb{E}_P\big[V_{h+1}^{k,\rho}\big] \\
&\quad + \inf_{\text{KL}(P\|P_h^o) \leq \rho} \mathbb{E}_P\big[V_{h+1}^{k,\rho}\big] - \inf_{\text{KL}(P\|P_h^o) \leq \rho} \mathbb{E}_P\big[V_{h+1}^{\pi,\rho}\big] \\
&\geq \text{bonus}_h^k + \widehat{r}_h^k - r_h + \inf_{\text{KL}(P\|\widehat{P}_h^k) \leq \rho} \mathbb{E}_P\big[V_{h+1}^{k,\rho}\big] - \inf_{\text{KL}(P\|P_h^o) \leq \rho} \mathbb{E}_P\big[V_{h+1}^{k,\rho}\big] \tag{C.39} \\
&= \text{bonus}_h^k + \widehat{r}_h^k - r_h + \inf_{\nu \in [0, \frac{H}{\rho}]} \big(\nu \ln \mathbb{E}_{P_h^o}\big[e^{-\nu^{-1}V_{h+1}^{k,\rho}}\big] + \nu\rho\big) \\
&\quad - \inf_{\nu \in [0, \frac{H}{\rho}]} \big(\nu \ln \mathbb{E}_{\widehat{P}_h^k}\big[e^{-\nu^{-1}V_{h+1}^{k,\rho}}\big] + \nu\rho\big) \tag{C.40} \\
&\geq \text{bonus}_h^k + \widehat{r}_h^k - r_h + \inf_{\nu \in [0, \frac{H}{\rho}]} \big(\nu \ln \mathbb{E}_{P_h^o}\big[e^{-\nu^{-1}V_{h+1}^{k,\rho}}\big] - \nu \ln \mathbb{E}_{\widehat{P}_h^k}\big[e^{-\nu^{-1}V_{h+1}^{k,\rho}}\big]\big) \\
&\geq \text{bonus}_h^k - \underbrace{\big|\widehat{r}_h^k - r_h\big|}_{(i)} - \underbrace{\sup_{\nu \in [0, \frac{H}{\rho}]} \big|\nu \ln \mathbb{E}_{\widehat{P}_h^k}\big[e^{-\nu^{-1}V_{h+1}^{k,\rho}}\big] - \nu \ln \mathbb{E}_{P_h^o}\big[e^{-\nu^{-1}V_{h+1}^{k,\rho}}\big]\big|}_{(ii)}, \tag{C.41}
\end{aligned}$$

where (C.39) is from the induction assumption, we plug in the dual formulation Lemma C.11 in (C.40).

For term (i) in (C.41), from Lemma G.1 and a union bound, with probability at least $1 - \delta$, we have

$$\big|\widehat{r}_h^k(s, a) - r_h(s, a)\big| \leq \sqrt{\frac{\ln(2SAHK/\delta)}{2n_h^{k-1}(s, a) \vee 1}}, \tag{C.42}$$

for any $(k, h, s, a) \in [K] \times [H] \times \mathcal{S} \times \mathcal{A}$.

To bound term (ii) in (C.11), we have

$$\sup_{\nu \in [0, \frac{H}{\rho}]} \big|\nu \ln \mathbb{E}_{\widehat{P}_h^k}\big[e^{-\nu^{-1}V_{h+1}^{k,\rho}}\big] - \nu \ln \mathbb{E}_{P_h^o}\big[e^{-\nu^{-1}V_{h+1}^{k,\rho}}\big]\big|$$

$$
= \sup_{\nu \in [0, \frac{H}{\rho}]} \left| \nu \ln \left( 1 + \frac{\sum_{s \in \mathcal{S}} \left( \widehat{P}_h^k(s) - P_h^o(s) \right) e^{-\nu^{-1} V_{h+1}^{k,\rho}(s)}}{\sum_{s \in \mathcal{S}} P_h^o(s) e^{-\nu^{-1} V_{h+1}^{k,\rho}(s)}} \right) \right|
$$

$$
\leq \sup_{\nu \in [0, \frac{H}{\rho}]} 2\nu \left| \frac{\sum_{s \in \mathcal{S}} \left( \widehat{P}_h^k(s) - P_h^o(s) \right) e^{-\nu^{-1} V_{h+1}^{k,\rho}(s)}}{\sum_{s \in \mathcal{S}} P_h^o(s) e^{-\nu^{-1} V_{h+1}^{k,\rho}(s)}} \right| \tag{C.43}
$$

$$
\leq \sup_{\nu \in [0, \frac{H}{\rho}]} 2\nu \cdot \max_{s \in \mathcal{S}, P_h^o(s) \neq 0} \left| \frac{\widehat{P}_h^k(s) - P_h^o(s)}{P_h^o(s)} \right| \tag{C.44}
$$

$$
\leq \frac{2H}{\rho C_{MP}} \cdot \max_{s \in \mathcal{S}} \left| \widehat{P}_h^k(s) - P_h^o(s) \right| \tag{C.45}
$$

$$
\leq \frac{2H}{\rho C_{MP}} \cdot \left\| \widehat{P}_h^k - P_h^o \right\|_1
$$

$$
\leq \frac{2H}{\rho C_{MP}} \sqrt{\frac{2S \ln(2SAHK/\delta)}{n_h^{k-1}(s,a) \vee 1}}, \tag{C.46}
$$

for any $(k, h, s, a) \in [K] \times [H] \times \mathcal{S} \times \mathcal{A}$, where (C.43) is because $\ln(1 + x) \leq 2|x|$, (C.44) follows from the Holder's inequality, noting that we have $n_h^{k-1}(s,a) = 0$ when $P_h^o(s) = 0$ and thus $\widehat{P}_h^k(s) = 0$ from (4.1), (C.45) uses Assumption 5.7, (C.46) is from (C.14) and a union bound.

Apply the union bound again and combine (C.41) with (C.42), (C.46), the definition of bonus and induction assumption. With probability at least $1 - 2\delta$, we have $Q_h^{k,\rho}(s,a) \geq Q_h^{\pi,\rho}(s,a)$ for any $(k, h, s, a) \in [K] \times [H] \times \mathcal{S} \times \mathcal{A}$. This completes the proof. □

With similar proof to Lemma C.3, we can control the item $Q^k - Q^{\pi^k}$.

**Lemma C.13.** For any $k \in [K]$, we can bound the estimation error as follows

$$
Q_1^{k,\rho} - Q_1^{\pi^k,\rho} \leq \sum_{h=1}^H \mathbb{E}_{\{P_h^{w,k}\}_{h=1}^H, \pi^k} \left[ 2\text{bonus}_h^k \right].
$$

*Proof.* From the proof of Lemma C.12, we see that

$$
\left| \widehat{r}_h^k(s,a) - r_h(s,a) \right| + \left| \inf_{\text{KL}(P \| \widehat{P}_h^k) \leq \rho} \mathbb{E}_P \left[ V_{h+1}^{k,\rho} \right](s,a) - \inf_{\text{KL}(P \| P_h^o) \leq \rho} \mathbb{E}_P \left[ V_{h+1}^{k,\rho} \right](s,a) \right| \leq \text{bonus}_h^k(s,a). \tag{C.47}
$$

Recall that we define $P_h^{w,k} = \operatorname*{argmin}_{\text{KL}(P \| P_h^o) \leq \rho} \mathbb{E}_P \left[ V_{h+1}^{\pi^k,\rho} \right]$ as the worst-case transition in Definition 5.3, we have

$$
Q_h^{k,\rho} - Q_h^{\pi^k,\rho} = \text{bonus}_h^k + \widehat{r}_h^k + \inf_{\text{KL}(P \| \widehat{P}_h^k) \leq \rho} \mathbb{E}_P \left[ V_{h+1}^{k,\rho} \right] - r_h - \inf_{\text{KL}(P \| P_h^o) \leq \rho} \mathbb{E}_P \left[ V_{h+1}^{\pi^k,\rho} \right]
$$

$$
\leq 2\text{bonus}_h^k + \inf_{\text{KL}(P \| P_h^o) \leq \rho} \mathbb{E}_P \left[ V_{h+1}^{k,\rho} \right] - \inf_{\text{KL}(P \| P_h^o) \leq \rho} \mathbb{E}_P \left[ V_{h+1}^{\pi^k,\rho} \right] \tag{C.48}
$$

$$
= 2\text{bonus}_h^k + \mathbb{E}_{P_h^{w,k}, \pi^k} \left[ Q_{h+1}^{k,\rho} - Q_{h+1}^{\pi^k,\rho} \right]. \tag{C.49}
$$

where (C.48) uses (C.47). Apply (C.49) recursively, we can obtain the result. □

Combine everything together the same way as the proof of Theorem C.10, we have

**Theorem C.14** (Restatement of Theorem 5.9 in KL-divergence setting). For CRMDP with $(s,a)$-rectangular KL-divergence defined uncertainty set satisfying Assumption 5.5 and Assumption 5.7, with probability at least $1 - \delta$, the regret of Algorithm 1 satisfies

$$\text{Regret} = \widetilde{\mathcal{O}}\bigg(\bigg(1 + \frac{H\sqrt{S}}{\rho C_{MP}}\bigg)\big(C_{vr}SAH + C_{vr}^{\frac{1}{2}}S^{\frac{1}{2}}A^{\frac{1}{2}}H\sqrt{K}\big)\bigg).$$

*Proof.* Setting $\delta' = \delta/3$ in Lemmas C.4 and C.12, then with probability at least $1 - \delta$, we get

$$\text{Regret} = \sum_{k=1}^{K}\big(V_1^{*,\rho} - V_1^{\pi^k,\rho}\big)$$

$$= \sum_{k=1}^{K}\big(V_1^{*,\rho} - V_1^{k,\rho}\big) + \sum_{k=1}^{K}\big(V_1^{k,\rho} - V_1^{\pi^k,\rho}\big)$$

$$\leq \sum_{k=1}^{K}\sum_{h=1}^{H}\mathbb{E}_{P_h^{w,k},\pi^k}\big[2\text{bonus}_h^k\big] \tag{C.50}$$

$$= 2\sum_{k=1}^{K}\sum_{h=1}^{H}\mathbb{E}_{P_h^{w,k},\pi^k}\bigg[\bigg(1 + \frac{2H\sqrt{S}}{\rho C_{MP}}\bigg)\sqrt{\frac{2\ln(2SAHK/\delta)}{n_h^{k-1}(s,a)\vee 1}}\bigg] \tag{C.51}$$

$$= \widetilde{\mathcal{O}}\bigg(\bigg(1 + \frac{H\sqrt{S}}{\rho C_{MP}}\bigg)\big(C_{vr}SAH + C_{vr}^{\frac{1}{2}}S^{\frac{1}{2}}A^{\frac{1}{2}}H\sqrt{K}\big)\bigg), \tag{C.52}$$

where (C.50) is the combination of Lemma C.12 and Lemma C.13, we plug in the bonus (C.38) in (C.51), (C.52) is from Lemma C.9. $\qquad\square$

## C.3. Proof of Theorem 5.9 (Constrained $\chi^2$ Setting)

We first give the closed form solution of constrained $\chi^2$ update formulation. This dual formulation has also been proved by Iyengar (2005, Lemma 4.2), but the rage of $\lambda$ in our result is more precise compared to theirs.

**Lemma C.15** (Dual formulation). For the optimization problem $Q_h(s,a) = r_h(s,a) + \inf_{\chi^2(P\|P_h^o)\leq\rho}\mathbb{E}_P[V_{h+1}](s,a)$, we have its dual formulation as follows

$$Q_h = r_h + \sup_{\boldsymbol{\lambda}\in[0,H]}\bigg(\mathbb{E}_{P_h^o}\big[V_{h+1} - \boldsymbol{\lambda}\big] - \sqrt{\rho\text{Var}_{P_h^o}(V_{h+1} - \boldsymbol{\lambda})}\bigg). \tag{C.53}$$

*Proof.* Consider the optimization problem

$$Q_h = r_h + \inf_{\chi^2(P\|P_h^o)\leq\rho}\mathbb{E}_P[V_{h+1}] = r_h + \inf_{\chi^2(P\|P_h^o)\leq\rho}\sum_{s\in\mathcal{S}}P(s)V_{h+1}(s).$$

The Lagrangian can be written as

$$\mathcal{L}(P,\eta) = \sum_{s\in S}P(s)V_{h+1}(s) + \nu\bigg(\sum_{s\in S}P_h^o(s)\bigg(\frac{P(s)}{P_h^o(s)} - 1\bigg)^2 - \rho\bigg) - \sum_{s\in\mathcal{S}}\boldsymbol{\lambda}(s)P(s) + \eta\bigg(1 - \sum_{s\in S}P(s)\bigg).$$

We set the derivative of $\mathcal{L}$ w.r.t. $P(s)$ to zero

$$\frac{\partial\mathcal{L}}{\partial P(s)} = V_{h+1}(s) + 2\nu\bigg(\frac{P(s)}{P_h^o(s)} - 1\bigg) - [\boldsymbol{\lambda}(s) + \eta] = 0. \tag{C.54}$$

We denote $P'$ as the worst-case transition that satisfies (C.54), then we have

$$P'(s) = P_h^o(s)\bigg(1 - \frac{V_{h+1}(s) - [\boldsymbol{\lambda}(s) + \eta]}{2\nu}\bigg),$$

$$\inf_P \mathcal{L}(P, \eta) = -\mathbb{E}_{P_h^o}\left[\frac{1}{4\nu}(V_{h+1}(s) - [\boldsymbol{\lambda}(s) + \eta])^2 - (V_{h+1}(s) - [\boldsymbol{\lambda}(s) + \eta])\right] - \nu\rho + \eta.$$

Therefore,

$$
\begin{aligned}
Q_h &= r_h + \sup_{\nu \geq 0, \boldsymbol{\lambda} \geq 0, \eta}\left(\inf_P \mathcal{L}(P, \eta)\right) \\
&= r_h - \inf_{\nu \geq 0, \boldsymbol{\lambda} \geq 0, \eta}\left(\mathbb{E}_{P_h^o}\left[\frac{1}{4\nu}(V_{h+1}(s) - [\boldsymbol{\lambda}(s) + \eta])^2 - (V_{h+1}(s) - [\boldsymbol{\lambda}(s) + \eta])\right] + \nu\rho - \eta\right) \\
&= r_h + \sup_{\nu \geq 0, \boldsymbol{\lambda} \geq 0}\left(\mathbb{E}_{P_h^o}[V_{h+1} - \boldsymbol{\lambda}] - \frac{1}{4\nu}\mathrm{Var}_{P_h^o}(V_{h+1} - \boldsymbol{\lambda}) - \nu\rho\right) && \text{(C.55)} \\
&= r_h + \sup_{\boldsymbol{\lambda} \geq 0}\left(\mathbb{E}_{P_h^o}[V_{h+1} - \boldsymbol{\lambda}] - \sqrt{\rho\mathrm{Var}_{P_h^o}(V_{h+1} - \boldsymbol{\lambda})}\right) && \text{(C.56)} \\
&= r_h + \sup_{\boldsymbol{\lambda} \in [0, H]}\left(\mathbb{E}_{P_h^o}[V_{h+1} - \boldsymbol{\lambda}] - \sqrt{\rho\mathrm{Var}_{P_h^o}(V_{h+1} - \boldsymbol{\lambda})}\right), && \text{(C.57)}
\end{aligned}
$$

where (C.55) holds by calculating the derivation with respect to $\eta$ and thus setting $\eta = \mathbb{E}_{P_h^o}[V_{h+1} - \boldsymbol{\lambda}]$, (C.56) is from the basic inequality $a + b \geq 2\sqrt{ab}$, (C.57) holds because the result increases monotonically with respect to $\boldsymbol{\lambda}$ when $\boldsymbol{\lambda} \geq H$. $\square$

Similar to Lemma C.2, we prove the optimism of estimation $Q^k$ and control $Q^* - Q^k$.

**Lemma C.16** (Optimism). If we set the bonus term as follows

$$\mathrm{bonus}_h^k(s, a) = 3H\sqrt{\frac{2S^2 \ln(192SAH^3K^3/\delta)}{n_h^{k-1}(s, a) \vee 1}} + \frac{2}{K}, \tag{C.58}$$

then for any policy $\pi$ and any $(k, h, s, a) \in [K] \times [H] \times \mathcal{S} \times \mathcal{A}$, with probability at least $1 - 3\delta$, we have $Q_h^{k,\rho}(s, a) \geq Q_h^{\pi,\rho}(s, a)$. Specially, by setting $\pi = \pi^*$, we have $Q_h^{k,\rho}(s, a) \geq Q_h^{\pi^*,\rho}(s, a)$.

*Proof.* We prove this by induction. First, when $h = H + 1$, $Q_{H+1}^{k,\rho}(s, a) = 0 = Q_{H+1}^{\pi,\rho}(s, a)$ holds trivially.

Assume $Q_{h+1}^{k,\rho}(s, a) \geq Q_{h+1}^{\pi,\rho}(s, a)$ holds, since $\pi^k$ is the greedy policy, we have

$$V_{h+1}^{k,\rho}(s) = Q_{h+1}^{k,\rho}(s, \pi_{h+1}^k(s)) \geq Q_{h+1}^{k,\rho}(s, \pi_{h+1}(s)) \geq Q_{h+1}^{\pi,\rho}(s, \pi_{h+1}(s)) = V_{h+1}^{\pi,\rho}(s),$$

where the first inequality is because we choose $\pi^k$ as the greedy policy.

Recall that we denote $Q_h^{k,\rho}$ as the optimistic estimation in $k$-th episode, that is,

$$Q_h^{k,\rho}(s, a) = \mathrm{bonus}_h^k(s, a) + \widehat{r}_h^k(s, a) + \inf_{\chi^2(P\|\widehat{P}_h^k) \leq \rho}\mathbb{E}_P[V_{h+1}^{k,\rho}](s, a).$$

We can infer that

$$
\begin{aligned}
Q_h^{k,\rho} - Q_h^{\pi,\rho} &= \mathrm{bonus}_h^k + \widehat{r}_h^k + \inf_{\chi^2(P\|\widehat{P}_h^k) \leq \rho}\mathbb{E}_P[V_{h+1}^{k,\rho}] - r_h - \inf_{\chi^2(P\|P_h^o) \leq \rho}\mathbb{E}_P[V_{h+1}^{\pi,\rho}] \\
&= \mathrm{bonus}_h^k + \widehat{r}_h^k - r_h + \inf_{\chi^2(P\|\widehat{P}_h^k) \leq \rho}\mathbb{E}_P[V_{h+1}^{k,\rho}] - \inf_{\chi^2(P\|P_h^o) \leq \rho}\mathbb{E}_P[V_{h+1}^{k,\rho}] \\
&\quad + \inf_{\chi^2(P\|P_h^o) \leq \rho}\mathbb{E}_P[V_{h+1}^{k,\rho}] - \inf_{\chi^2(P\|P_h^o) \leq \rho}\mathbb{E}_P[V_{h+1}^{\pi,\rho}] \\
&\geq \mathrm{bonus}_h^k + \widehat{r}_h^k - r_h + \inf_{\chi^2(P\|\widehat{P}_h^k) \leq \rho}\mathbb{E}_P[V_{h+1}^{k,\rho}] - \inf_{\chi^2(P\|P_h^o) \leq \rho}\mathbb{E}_P[V_{h+1}^{k,\rho}] && \text{(C.59)} \\
&= \mathrm{bonus}_h^k + \widehat{r}_h^k - r_h + \sup_{\boldsymbol{\lambda} \in [0, H]}\left(\mathbb{E}_{\widehat{P}_h^k}[V_{h+1}^{k,\rho} - \boldsymbol{\lambda}] - \sqrt{\rho\mathrm{Var}_{\widehat{P}_h^k}(V_{h+1}^{k,\rho} - \boldsymbol{\lambda})}\right)
\end{aligned}
$$

$$- \sup_{\boldsymbol{\lambda} \in [0,H]} \left( \mathbb{E}_{P_h^o} \left[ V_{h+1}^{k,\rho} - \boldsymbol{\lambda} \right] - \sqrt{\rho \mathrm{Var}_{P_h^o} \left( V_{h+1}^{k,\rho} - \boldsymbol{\lambda} \right)} \right) \tag{C.60}$$

$$\geq \mathrm{bonus}_h^k + \widehat{r}_h^k - r_h + \inf_{\boldsymbol{\lambda} \in [0,H]} \left\{ \left( \mathbb{E}_{\widehat{P}_h^k} \left[ V_{h+1}^{k,\rho} - \boldsymbol{\lambda} \right] - \sqrt{\rho \mathrm{Var}_{\widehat{P}_h^k} \left( V_{h+1}^{k,\rho} - \boldsymbol{\lambda} \right)} \right) \right.$$

$$\left. - \left( \mathbb{E}_{P_h^o} \left[ V_{h+1}^{k,\rho} - \boldsymbol{\lambda} \right] - \sqrt{\rho \mathrm{Var}_{P_h^o} \left( V_{h+1}^{k,\rho} - \boldsymbol{\lambda} \right)} \right) \right\}$$

$$\geq \mathrm{bonus}_h^k - \left| \widehat{r}_h^k - r_h \right| - \sup_{\boldsymbol{\lambda} \in [0,H]} \left| \left( \mathbb{E}_{\widehat{P}_h^k} \left[ V_{h+1}^{k,\rho} - \boldsymbol{\lambda} \right] - \sqrt{\rho \mathrm{Var}_{\widehat{P}_h^k} \left( V_{h+1}^{k,\rho} - \boldsymbol{\lambda} \right)} \right) \right.$$

$$\left. - \left( \mathbb{E}_{P_h^o} \left[ V_{h+1}^{k,\rho} - \boldsymbol{\lambda} \right] - \sqrt{\rho \mathrm{Var}_{P_h^o} \left( V_{h+1}^{k,\rho} - \boldsymbol{\lambda} \right)} \right) \right|$$

$$\geq \mathrm{bonus}_h^k - \underbrace{\left| \widehat{r}_h^k - r_h \right|}_{(i)} - \underbrace{\sup_{\boldsymbol{\lambda} \in [0,H]} \left| \mathbb{E}_{\widehat{P}_h^k} \left[ V_{h+1}^{k,\rho} - \boldsymbol{\lambda} \right] - \mathbb{E}_{P_h^o} \left[ V_{h+1}^{k,\rho} - \boldsymbol{\lambda} \right] \right|}_{(ii)}$$

$$- \underbrace{\sqrt{\rho} \sup_{\boldsymbol{\lambda} \in [0,H]} \left| \sqrt{\mathrm{Var}_{\widehat{P}_h^k} \left( V_{h+1}^{k,\rho} - \boldsymbol{\lambda} \right)} - \sqrt{\mathrm{Var}_{P_h^o} \left( V_{h+1}^{k,\rho} - \boldsymbol{\lambda} \right)} \right|}_{(iii)}, \tag{C.61}$$

where (C.59) is from the induction assumption, we plug in the dual formulation Lemma C.15 in (C.60), (C.61) is because $\sup f(x) + \sup g(x) \geq \sup (f-g)(x)$.

For term (i) in (C.61), from Lemma G.1 and a union bound, with probability at least $1 - \delta$, we have

$$\left| \widehat{r}_h^k(s,a) - r_h(s,a) \right| \leq \sqrt{\frac{\ln(2SAHK/\delta)}{2n_h^{k-1}(s,a) \vee 1}}, \tag{C.62}$$

for any $(k,h,s,a) \in [K] \times [H] \times \mathcal{S} \times \mathcal{A}$.

We denote $V(\boldsymbol{\lambda}) = V_{h+1}^{k,\rho} - \boldsymbol{\lambda} \in [-H, H]$ and $\mathcal{V} = \left\{ V \in \mathbb{R}^S : \|V\|_\infty \leq H \right\}$. To bound term (ii) in (C.61), we create a $\epsilon$-net $\mathcal{N}_\mathcal{V}(\epsilon)$ for $\mathcal{V}$. From Lemma G.4, it holds that $\ln |\mathcal{N}_\mathcal{V}(\epsilon)| \leq |S| \cdot \ln(3H/\epsilon)$. Then follow the same proof as (C.17), with probability at least $1 - \delta$, we have

$$\sup_{\boldsymbol{\lambda} \in [0,H]} \left| \mathbb{E}_{\widehat{P}_h^k} \left[ V_{h+1}^{k,\rho} - \boldsymbol{\lambda} \right] - \mathbb{E}_{P_h^o} \left[ V_{h+1}^{k,\rho} - \boldsymbol{\lambda} \right] \right| \leq \sup_{V \in \mathcal{N}_\mathcal{V}(\epsilon)} \left| \mathbb{E}_{P_h^o}[V] - \mathbb{E}_{\widehat{P}_h^k}[V] \right| + 2\epsilon$$

$$\leq H \sqrt{\frac{2S^2 \ln(12SAH^2K^2/\delta)}{n_h^{k-1} \vee 1}} + \frac{1}{K}, \tag{C.63}$$

for any $(k,h,s,a) \in [K] \times [H] \times \mathcal{S} \times \mathcal{A}$.

For term (iii) in (C.61), by the definition of $\mathcal{N}_\mathcal{V}(\epsilon)$, for any fixed $V$, there exists a $V' \in \mathcal{N}_\mathcal{V}(\epsilon)$ such that $\|V - V'\|_\infty \leq \epsilon$, that is

$$\left| \sqrt{\mathrm{Var}_{P_h^o}(V)} - \sqrt{\mathrm{Var}_{\widehat{P}_h^k}(V)} \right|$$

$$\leq \left| \sqrt{\mathrm{Var}_{P_h^o}(V)} - \sqrt{\mathrm{Var}_{P_h^o}(V')} \right| + \left| \sqrt{\mathrm{Var}_{P_h^o}(V')} - \sqrt{\mathrm{Var}_{\widehat{P}_h^k}(V')} \right| + \left| \sqrt{\mathrm{Var}_{\widehat{P}_h^k}(V')} - \sqrt{\mathrm{Var}_{\widehat{P}_h^k}(V)} \right|$$

$$\leq \sqrt{\left| \mathrm{Var}_{P_h^o}(V) - \mathrm{Var}_{P_h^o}(V') \right|} + \left| \sqrt{\mathrm{Var}_{P_h^o}(V')} - \sqrt{\mathrm{Var}_{\widehat{P}_h^k}(V')} \right| + \sqrt{\left| \mathrm{Var}_{\widehat{P}_h^k}(V') - \mathrm{Var}_{\widehat{P}_h^k}(V) \right|}$$

$$\leq \sqrt{\left| \mathbb{E}_{P_h^o}[V^2 - V'^2] \right|} + \sqrt{\left| \mathbb{E}_{P_h^o}^2[V] - \mathbb{E}_{P_h^o}^2[V'] \right|} + \sqrt{\left| \mathbb{E}_{\widehat{P}_h^k}[V'^2 - V^2] \right|} + \sqrt{\left| \mathbb{E}_{\widehat{P}_h^k}^2[V'] - \mathbb{E}_{\widehat{P}_h^k}^2[V] \right|}$$

$$\quad + \left| \sqrt{\mathrm{Var}_{P_h^o}(V')} - \sqrt{\mathrm{Var}_{\widehat{P}_h^k}(V')} \right|$$

$$\leq \left| \sqrt{\mathrm{Var}_{P_h^o}(V')} - \sqrt{\mathrm{Var}_{\widehat{P}_h^k}(V')} \right| + 4\sqrt{2H\epsilon}$$

$$\leq \sup_{V' \in \mathcal{N}_\mathcal{V}(\epsilon)} \left| \sqrt{\mathrm{Var}_{P_h^o}(V')} - \sqrt{\mathrm{Var}_{\widehat{P}_h^k}(V')} \right| + 4\sqrt{2H\epsilon}, \tag{C.64}$$

where the second inequality follows from $\left|\sqrt{x} - \sqrt{y}\right| \leq \sqrt{|x-y|}$ and the third inequality follows from $\sqrt{x+y} \leq \sqrt{x} + \sqrt{y}$.

For any fixed $V$, we apply Lemma G.2 and have

$$\left|\sqrt{\operatorname{Var}_{P_h^o}(V)} - \sqrt{\operatorname{Var}_{\widehat{P}_h^k}(V)}\right| \leq H\sqrt{\frac{2\ln(2/\delta)}{n_h^{k-1} \vee 1}}. \tag{C.65}$$

with probability at least $1-\delta$.

Then with probability at least $1-\delta$, we have

$$\sup_{\boldsymbol{\lambda} \in [0,H]} \left|\sqrt{\operatorname{Var}_{\widehat{P}_h^k}(V_{h+1}^{k,\rho} - \boldsymbol{\lambda})} - \sqrt{\operatorname{Var}_{P_h^o}(V_{h+1}^{k,\rho} - \boldsymbol{\lambda})}\right| \leq \sup_{\boldsymbol{\lambda} \in [0,H]} \left|\sqrt{\operatorname{Var}_{P_h^o}(V(\boldsymbol{\lambda}))} - \sqrt{\operatorname{Var}_{\widehat{P}_h^k}(V(\boldsymbol{\lambda}))}\right|$$

$$\leq \sup_{V \in \mathcal{N}_{\mathcal{V}}(\epsilon)} \left|\sqrt{\operatorname{Var}_{P_h^o}(V)} - \sqrt{\operatorname{Var}_{\widehat{P}_h^k}(V)}\right| + 4\sqrt{2H\epsilon} \tag{C.66}$$

$$\leq H\sqrt{\frac{2\ln(2SAHK|\mathcal{N}_{\mathcal{V}}(\epsilon)|/\delta)}{n_h^{k-1} \vee 1}} + 4\sqrt{2H\epsilon} \tag{C.67}$$

$$\leq H\sqrt{\frac{2S\ln(6SAH^2K/\epsilon\delta)}{n_h^{k-1} \vee 1}} + 4\sqrt{2H\epsilon}$$

$$= H\sqrt{\frac{2S\ln(192SAH^3K^3/\delta)}{n_h^{k-1} \vee 1}} + \frac{1}{K}, \tag{C.68}$$

for any $(k,h,s,a) \in [K] \times [H] \times \mathcal{S} \times \mathcal{A}$, where (C.66) follows from (C.64), (C.67) is from (C.65) and a union bound, we set $\epsilon = 1/32HK^2$ in (C.68).

Apply the union bound again and combine (C.61) with (C.62), (C.63), (C.68), the definition of bonus and induction assumption. With probability at least $1-3\delta$, we have $Q_h^{k,\rho}(s,a) \geq Q_h^{\pi,\rho}(s,a)$ for any $(k,h,s,a) \in [K] \times [H] \times \mathcal{S} \times \mathcal{A}$. This completes the proof. $\qquad\square$

With similar proof to Lemma C.3, we can control the item $Q^k - Q^{\pi^k}$.

**Lemma C.17.** For any $k \in [K]$, we can bound the estimation error as follows

$$Q_1^{k,\rho} - Q_1^{\pi^k,\rho} \leq \sum_{h=1}^{H} \mathbb{E}_{\{P_h^{w,k}\}_{h=1}^H, \pi^k}\left[2\text{bonus}_h^k\right].$$

*Proof.* From the proof of Lemma C.12, we see that

$$\left|\widehat{r}_h^k(s,a) - r_h(s,a)\right| + \left|\inf_{\chi^2(P\|\widehat{P}_h^k)\leq\rho} \mathbb{E}_P\left[V_{h+1}^{k,\rho}\right](s,a) - \inf_{\chi^2(P\|P_h^o)\leq\rho} \mathbb{E}_P\left[V_{h+1}^{k,\rho}\right](s,a)\right| \leq \text{bonus}_h^k(s,a). \tag{C.69}$$

Recall that we define $P_h^{w,k} = \operatorname*{argmin}_{\chi^2(P\|P_h^o)\leq\rho} \mathbb{E}_P\left[V_{h+1}^{\pi^k,\rho}\right]$ as the worst-case transition in Definition 5.3, we have

$$Q_h^{k,\rho} - Q_h^{\pi^k,\rho} = \text{bonus}_h^k + \widehat{r}_h^k + \inf_{\chi^2(P\|\widehat{P}_h^k)\leq\rho} \mathbb{E}_P\left[V_{h+1}^{k,\rho}\right] - r_h - \inf_{\chi^2(P\|P_h^o)\leq\rho} \mathbb{E}_P\left[V_{h+1}^{\pi^k,\rho}\right]$$

$$\leq 2\text{bonus}_h^k + \inf_{\chi^2(P\|P_h^o)\leq\rho} \mathbb{E}_P\left[V_{h+1}^{k,\rho}\right] - \inf_{\chi^2(P\|P_h^o)\leq\rho} \mathbb{E}_P\left[V_{h+1}^{\pi^k,\rho}\right] \tag{C.70}$$

$$= 2\text{bonus}_h^k + \mathbb{E}_{P_h^{w,k}, \pi^k}\left[Q_{h+1}^{k,\rho} - Q_{h+1}^{\pi^k,\rho}\right], \tag{C.71}$$

where (C.70) uses (C.69). Apply (C.71) recursively, we can obtain the result. $\qquad\square$

**Theorem C.18** (Restatement of Theorem 5.9 in $\chi^2$-divergence setting). For CRMDP with $(s, a)$-rectangular $\chi^2$-divergence defined uncertainty set satisfying Assumption 5.5, with probability at least $1 - \delta$, the regret of Algorithm 1 satisfies

$$\text{Regret} = \widetilde{\mathcal{O}}\big(C_{vr}S^2AH^2 + C_{vr}^{\frac{1}{2}}S^{\frac{3}{2}}A^{\frac{1}{2}}H^2\sqrt{K}\big).$$

*Proof.* Setting $\delta' = \delta/4$ in Lemmas C.4 and C.16, then with probability at least $1 - \delta$, we get

$$\text{Regret} = \sum_{k=1}^{K} \big(V_1^{*,\rho} - V_1^{\pi^k,\rho}\big) = \sum_{k=1}^{K} \big(V_1^{*,\rho} - V_1^{k,\rho}\big) + \sum_{k=1}^{K} \big(V_1^{k,\rho} - V_1^{\pi^k,\rho}\big)$$

$$\leq \sum_{k=1}^{K}\sum_{h=1}^{H} \mathbb{E}_{P_h^{w,k}, \pi^k}\big[2\text{bonus}_h^k\big] \tag{C.72}$$

$$= 2\sum_{k=1}^{K}\sum_{h=1}^{H} \mathbb{E}_{P_h^{w,k}, \pi^k}\left[3H\sqrt{\frac{2S^2\ln(192SAH^3K^3/\delta)}{n_h^{k-1}(s,a) \vee 1}} + \frac{2}{K}\right] \tag{C.73}$$

$$= \widetilde{\mathcal{O}}\big(C_{vr}S^2AH^2 + C_{vr}^{\frac{1}{2}}S^{\frac{3}{2}}A^{\frac{1}{2}}H^2\sqrt{K}\big), \tag{C.74}$$

where (C.72) is the combination of Lemma C.16 and Lemma C.17, we plug in the bonus (C.58) in (C.73), (C.74) is from Lemma C.9. $\qquad\square$

## D. Proofs of Results in Regularized RMDPs

### D.1. Proof of Theorem 5.16 (Regularized TV Setting)

We first give the closed form solution of regularized TV update formulation. This dual formulation has also been proved by Panaganti et al. (2024), but our formulation is more concise.

**Lemma D.1** (Dual formulation). For the optimization problem $Q_h(s, a) = r_h(s, a) + \inf_{P \in \Delta(S)} \big(\mathbb{E}_P[V_{h+1}] + \beta\text{TV}\big(P\|P_h^o\big)\big)(s, a)$, we have its dual formulation as follows

$$Q_h = r_h - \mathbb{E}_{P_h^o}\left[\left(\min_{s \in \mathcal{S}} V_{h+1}(s) + \beta - V_{h+1}(s)\right)_+\right] + \left(\min_{s \in \mathcal{S}} V_{h+1}(s) + \beta\right). \tag{D.1}$$

*Proof.* Consider the optimization problem

$$Q_h = r_h + \inf_{P \in \Delta(S)} \big(\mathbb{E}_P[V_{h+1}] + \beta\text{TV}\big(P\|P_h^o\big)\big)$$

$$= r_h + \inf_{P \in \Delta(S)} \left(\sum_{s \in \mathcal{S}} P(s)V_{h+1}(s) + \beta P_h^o(s)\varphi\left(\frac{P(s)}{P_h^o(s)}\right)\right).$$

where $\varphi(t) = |t - 1|/2$, and then the Lagrangian can be written as

$$\mathcal{L}(P, \eta) = \sum_{s \in S} P(s)V_{h+1}(s) + \beta\sum_{s \in S} P_h^o(s)\varphi\left(\frac{P(s)}{P_h^o(s)}\right) - \sum_{s \in \mathcal{S}}\lambda(s)P(s) + \eta\left(1 - \sum_{s \in S} P(s)\right).$$

We denote $g(s) = P(s)/P_h^o(s)$, then we have

$$\inf_P \mathcal{L}(P, \eta) = -\sup_g \sum_{s \in \mathcal{S}} P_h^o(s)\big[g(s)\big[(\lambda(s) + \eta) - V_{h+1}(s)\big] - \beta\varphi(g(s))\big] + \eta$$

$$= -\beta\mathbb{E}_{P_h^o}\left[\varphi^*\left(\frac{(\lambda(s) + \eta) - V_{h+1}(s)}{\beta}\right)\right] + \eta,$$

where the second equation is from the definition of dual function $\varphi^*(y) = \sup_x (y^\top x - \varphi(x))$.

So we have

$$
\begin{aligned}
Q_h &= r_h + \sup_{\lambda \geq 0, \eta} \left( \inf_P \mathcal{L}(P, \eta) \right) \\
&= r_h - \inf_{\lambda \geq 0, \eta} \left( \beta \mathbb{E}_{P_h^o} \left[ \varphi^* \left( \frac{(\lambda(s) + \eta) - V_{h+1}(s)}{\beta} \right) \right] - \eta \right) \\
&= r_h - \inf_{\lambda \geq 0, \eta, \frac{\lambda(s) + \eta - V_{h+1}(s)}{\beta} \leq \frac{1}{2}} \left( \beta \mathbb{E}_{P_h^o} \left[ \max \left( \frac{(\lambda(s) + \eta) - V_{h+1}(s)}{\beta}, -\frac{1}{2} \right) \right] - \eta \right) \quad \text{(D.2)} \\
&= r_h - \inf_{\lambda \geq 0, \eta, \frac{\lambda(s) + \eta - V_{h+1}(s)}{\beta} \leq \frac{1}{2}} \left( \mathbb{E}_{P_h^o} \left[ \left( (\lambda(s) + \eta) - V_{h+1}(s) + \frac{\beta}{2} \right)_+ \right] - \eta - \frac{\beta}{2} \right) \\
&= r_h - \inf_{\lambda \geq 0, \eta', \lambda(s) + \eta' - V_{h+1}(s) \leq \beta} \left( \mathbb{E}_{P_h^o} \left[ \left( \lambda(s) + \eta' - V_{h+1}(s) \right)_+ \right] - \eta' \right) \quad \text{(D.3)} \\
&= r_h - \inf_{\eta' \leq V_{h+1}(s) + \beta} \left( \mathbb{E}_{P_h^o} \left[ (\eta' - V_{h+1}(s))_+ \right] - \eta' \right) \quad \text{(D.4)} \\
&= r_h - \mathbb{E}_{P_h^o} \left[ \left( \min_{s \in \mathcal{S}} V_{h+1}(s) + \beta - V_{h+1}(s) \right)_+ \right] + \left( \min_{s \in \mathcal{S}} V_{h+1}(s) + \beta \right), \quad \text{(D.5)}
\end{aligned}
$$

where (D.2) follows from the definition of $\varphi^*$ (C.2), we redefine $\eta' = \eta + \frac{\beta}{2}$ in (D.3), (D.4) holds because the result increases monotonically with respect to $\lambda$, (D.5) holds because the result increases monotonically with respect to $\eta'$. □

Similar to Lemma C.2, we prove the optimism of estimation $Q^k$ and control $Q^* - Q^k$.

**Lemma D.2** (Optimism). If we set the bonus term as follows

$$
\text{bonus}_h^k(s, a) = 2H \sqrt{\frac{2S \ln(2SAHK/\delta)}{n_h^{k-1}(s, a) \vee 1}}, \quad \text{(D.6)}
$$

then for any policy $\pi$ and any $(k, h, s, a) \in [K] \times [H] \times \mathcal{S} \times \mathcal{A}$, with probability at least $1 - 2\delta$, we have $Q_h^{k,\beta}(s, a) \geq Q_h^{\pi,\beta}(s, a)$. Specially, by setting $\pi = \pi^*$, we have $Q_h^{k,\beta}(s, a) \geq Q_h^{\pi^*,\beta}(s, a)$.

*Proof.* We prove this by induction. First, when $h = H + 1$, $Q_{H+1}^{k,\beta}(s, a) = 0 = Q_{H+1}^{\pi,\beta}(s, a)$ holds trivially.

Assume $Q_{h+1}^{k,\beta}(s, a) \geq Q_{h+1}^{\pi,\beta}(s, a)$ holds, since $\pi^k$ is the greedy policy, we have

$$
V_{h+1}^{k,\beta}(s) = Q_{h+1}^{k,\beta}(s, \pi_{h+1}^k(s)) \geq Q_{h+1}^{k,\beta}(s, \pi_{h+1}(s)) \geq Q_{h+1}^{\pi,\beta}(s, \pi_{h+1}(s)) = V_{h+1}^{\pi,\beta}(s),
$$

where the first inequality is because we choose $\pi^k$ as the greedy policy.

Recall that we denote $Q_h^{k,\beta}$ as the optimistic estimation in $k$-th episode, that is,

$$
Q_h^{k,\beta}(s, a) = \text{bonus}_h^k(s, a) + \widehat{r}_h^k(s, a) + \inf_{P \in \Delta(S)} \left( \mathbb{E}_P \left[ V_{h+1}^{k,\beta} \right] + \beta \text{TV}(P \| \widehat{P}_h^k) \right)(s, a).
$$

We can infer that

$$
\begin{aligned}
Q_h^{k,\beta} - Q_h^{\pi,\beta} &= \text{bonus}_h^k + \widehat{r}_h^k + \inf_{P \in \Delta(S)} \left( \mathbb{E}_P \left[ V_{h+1}^{k,\beta} \right] + \beta \text{TV}(P \| \widehat{P}_h^k) \right) - r_h - \inf_{P \in \Delta(S)} \left( \mathbb{E}_P \left[ V_{h+1}^{\pi,\beta} \right] + \beta \text{TV}(P \| P_h^o) \right) \\
&= \text{bonus}_h^k + \widehat{r}_h^k - r_h + \inf_{P \in \Delta(S)} \left( \mathbb{E}_P \left[ V_{h+1}^{k,\beta} \right] + \beta \text{TV}(P \| \widehat{P}_h^k) \right) - \inf_{P \in \Delta(S)} \left( \mathbb{E}_P \left[ V_{h+1}^{k,\beta} \right] + \beta \text{TV}(P \| P_h^o) \right) \\
&\quad + \inf_{P \in \Delta(S)} \left( \mathbb{E}_P \left[ V_{h+1}^{k,\beta} \right] + \beta \text{TV}(P \| P_h^o) \right) - \inf_{P \in \Delta(S)} \left( \mathbb{E}_P \left[ V_{h+1}^{\pi,\beta} \right] + \beta \text{TV}(P \| P_h^o) \right)
\end{aligned}
$$

$$
\geq \text{bonus}_h^k + \widehat{r}_h^k - r_h + \inf_{P \in \Delta(S)} \left( \mathbb{E}_P\big[V_{h+1}^{k,\beta}\big] - \mathbb{E}_P\big[V_{h+1}^{\pi,\beta}\big] \right)
$$

$$
+ \inf_{P \in \Delta(S)} \left( \mathbb{E}_P\big[V_{h+1}^{k,\beta}\big] + \beta \text{TV}\big(P\|\widehat{P}_h^k\big) \right) - \inf_{P \in \Delta(S)} \left( \mathbb{E}_P\big[V_{h+1}^{k,\beta}\big] + \beta \text{TV}\big(P\|P_h^o\big) \right) \tag{D.7}
$$

$$
\geq \text{bonus}_h^k + \widehat{r}_h^k - r_h + \inf_{P \in \Delta(S)} \left( \mathbb{E}_P\big[V_{h+1}^{k,\beta}\big] + \beta \text{TV}\big(P\|\widehat{P}_h^k\big) \right) - \inf_{P \in \Delta(S)} \left( \mathbb{E}_P\big[V_{h+1}^{k,\beta}\big] + \beta \text{TV}\big(P\|P_h^o\big) \right) \tag{D.8}
$$

$$
= \text{bonus}_h^k + \widehat{r}_h^k - r_h - \mathbb{E}_{P_h^o}\left[ \left( \min_{s \in \mathcal{S}} V_{h+1}^{k,\beta}(s) + \beta - V_{h+1}^{k,\beta}(s) \right)_+ \right] + \left( \min_{s \in \mathcal{S}} V_{h+1}^{k,\beta}(s) + \beta \right)
$$

$$
+ \mathbb{E}_{\widehat{P}_h^k}\left[ \left( \min_{s \in \mathcal{S}} V_{h+1}^{k,\beta}(s) + \beta - V_{h+1}^{k,\beta}(s) \right)_+ \right] - \left( \min_{s \in \mathcal{S}} V_{h+1}^{k,\beta}(s) + \beta \right) \tag{D.9}
$$

$$
= \text{bonus}_h^k + \widehat{r}_h^k - r_h - \mathbb{E}_{P_h^o}\left[ \left( \min_{s \in \mathcal{S}} V_{h+1}^{k,\beta}(s) + \beta - V_{h+1}^{k,\beta}(s) \right)_+ \right] + \mathbb{E}_{\widehat{P}_h^k}\left[ \left( \min_{s \in \mathcal{S}} V_{h+1}^{k,\beta}(s) + \beta - V_{h+1}^{k,\beta}(s) \right)_+ \right]
$$

$$
\geq \text{bonus}_h^k - \underbrace{\big|\widehat{r}_h^k - r_h\big|}_{\text{(i)}} - \underbrace{\left| \mathbb{E}_{P_h^o}\left[ \left( \min_{s \in \mathcal{S}} V_{h+1}^{k,\beta}(s) + \beta - V_{h+1}^{k,\beta}(s) \right)_+ \right] - \mathbb{E}_{\widehat{P}_h^k}\left[ \left( \min_{s \in \mathcal{S}} V_{h+1}^{k,\beta}(s) + \beta - V_{h+1}^{k,\beta}(s) \right)_+ \right] \right|}_{\text{(ii)}},
$$

$$
\tag{D.10}
$$

where (D.7) is from $\inf f(x) - \inf g(x) \geq \inf(f - g)(x)$, (D.8) is from the induction assumption, we plug in the dual formulation (D.1) in (D.9).

For term (i) in (D.10), from Lemma G.1 and a union bound, with probability at least $1 - \delta$, we have

$$
\big|\widehat{r}_h^k(s,a) - r_h(s,a)\big| \leq \sqrt{\frac{\ln(2SAHK/\delta)}{2n_h^{k-1}(s,a) \vee 1}} \tag{D.11}
$$

for any $(k,h,s,a) \in [K] \times [H] \times \mathcal{S} \times \mathcal{A}$.

For term (ii) in (D.10), follow the same proof as (C.14) and a union bound, with probability at least $1 - \delta$, we have

$$
\left| \mathbb{E}_{P_h^o}\left[ \left( \min_{s \in \mathcal{S}} V_{h+1}^{k,\beta}(s) + \beta - V_{h+1}^{k,\beta}(s) \right)_+ \right] - \mathbb{E}_{\widehat{P}_h^k}\left[ \left( \min_{s \in \mathcal{S}} V_{h+1}^{k,\beta}(s) + \beta - V_{h+1}^{k,\beta}(s) \right)_+ \right] \right| \leq H\sqrt{\frac{2S\ln(2SAHK/\delta)}{n_h^{k-1} \vee 1}} \tag{D.12}
$$

for any $(k,h,s,a) \in [K] \times [H] \times \mathcal{S} \times \mathcal{A}$.

Apply the union bound again and combine (D.10) with (D.11), (D.12), the definition of bonus and induction assumption. With probability at least $1 - 2\delta$, we have $Q_h^{k,\rho}(s,a) \geq Q_h^{\pi,\rho}(s,a)$ for any $(k,h,s,a) \in [K] \times [H] \times \mathcal{S} \times \mathcal{A}$. This completes the proof. $\qquad \square$

With similar proof to Lemma C.3, we can control the item $Q^k - Q^{\pi^k}$.

**Lemma D.3.** For any $k \in [K]$, we can bound the estimation error as follows

$$
Q_1^{k,\beta} - Q_1^{\pi^k,\beta} \leq \sum_{h=1}^{H} \mathbb{E}_{\{P_h^{w,k}\}_{h=1}^H, \pi^k}\left[ 2\text{bonus}_h^k \right].
$$

*Proof.* From the proof of Lemma D.2, we see that

$$
\big|\widehat{r}_h^k(s,a) - r_h(s,a)\big| + \Big| \inf_{P \in \Delta(S)} \big( \mathbb{E}_P\big[V_{h+1}^{k,\beta}\big] + \beta \text{TV}\big(P\|\widehat{P}_h^k\big) \big)(s,a)
$$

$$
- \inf_{P \in \Delta(S)} \big( \mathbb{E}_P\big[V_{h+1}^{k,\beta}\big] + \beta \text{TV}\big(P\|P_h^o\big) \big)(s,a) \Big| \leq \text{bonus}_h^k(s,a). \tag{D.13}
$$

Recall that we define $P_h^{w,k} = \underset{P \in \Delta(S)}{\operatorname{argmin}} \big(\mathbb{E}_P\big[V_{h+1}^{k,\beta}\big] + \beta \mathrm{TV}\big(P\|P_h^o\big)\big)$ as the worst-case transition in Definition 5.3, we have

$$Q_h^{k,\beta} - Q_h^{\pi^k,\beta} = \mathrm{bonus}_h^k + \widehat{r}_h^k + \inf_{P \in \Delta(S)} \big(\mathbb{E}_P\big[V_{h+1}^{k,\beta}\big] + \beta \mathrm{TV}\big(P\|\widehat{P}_h^k\big)\big) - r_h - \inf_{P \in \Delta(S)} \big(\mathbb{E}_P\big[V_{h+1}^{\pi^k,\beta}\big] + \beta \mathrm{TV}\big(P\|P_h^o\big)\big)$$

$$\leq 2\mathrm{bonus}_h^k + \inf_{P \in \Delta(S)} \big(\mathbb{E}_P\big[V_{h+1}^{k,\beta}\big] + \beta \mathrm{TV}\big(P\|P_h^o\big)\big) - \inf_{P \in \Delta(S)} \big(\mathbb{E}_P\big[V_{h+1}^{\pi^k,\beta}\big] + \beta \mathrm{TV}\big(P\|P_h^o\big)\big) \tag{D.14}$$

$$= 2\mathrm{bonus}_h^k + \mathbb{E}_{P_h^{w,k},\pi^k}\big[Q_{h+1}^{k,\beta} - Q_{h+1}^{\pi^k,\beta}\big]. \tag{D.15}$$

where (D.14) uses (D.13). Apply (D.15) recursively, we can obtain the result. $\qquad\square$

**Theorem D.4** (Restatement of Theorem 5.16 in TV-distance setting). *For RRMDP with $(s,a)$-rectangular TV-distance defined regularization term satisfying Assumption 5.5, with probability at least $1 - \delta$, the regret of Algorithm 1 satisfies*

$$\mathrm{Regret} = \widetilde{\mathcal{O}}\big(C_{vr}S^{\frac{3}{2}}AH^2 + C_{vr}^{\frac{1}{2}}SA^{\frac{1}{2}}H^2\sqrt{K}\big).$$

*Proof.* Setting $\delta' = \delta/3$ in Lemmas C.4 and D.2, then with probability at least $1 - \delta$, we get

$$\mathrm{Regret} = \sum_{k=1}^{K}\big(V_1^{*,\beta} - V_1^{\pi^k,\beta}\big) = \sum_{k=1}^{K}\big(V_1^{*,\beta} - V_1^{k,\beta}\big) + \sum_{k=1}^{K}\big(V_1^{k,\beta} - V_1^{\pi^k,\beta}\big)$$

$$\leq \sum_{k=1}^{K}\sum_{h=1}^{H}\mathbb{E}_{P_h^{w,k},\pi^k}\big[2\mathrm{bonus}_h^k\big] \tag{D.16}$$

$$= 2\sum_{k=1}^{K}\sum_{h=1}^{H}\mathbb{E}_{P_h^{w,k},\pi^k}\left[2H\sqrt{\frac{2S\ln(2SAHK/\delta)}{n_h^{k-1}(s,a)\vee 1}}\right] \tag{D.17}$$

$$= \widetilde{\mathcal{O}}\big(C_{vr}S^{\frac{3}{2}}AH^2 + C_{vr}^{\frac{1}{2}}SA^{\frac{1}{2}}H^2\sqrt{K}\big). \tag{D.18}$$

where (D.16) is the combination of Lemma D.2 and Lemma D.3, we plug in the bonus (D.6) in (D.17), (D.18) is from Lemma C.9. $\qquad\square$

## D.2. Proof of Theorem 5.16 (Regularized KL Setting)

We first give the closed form solution of regularized KL update formulation. This dual formulation has also been proved by Panaganti et al. (2024), but our formulation is more concise. Zhang et al. (2024) applied the equivalence between regularized RMDPs and risk-sensitive MDPs to obtain the same result, which is different from our proof.

**Lemma D.5** (Dual formulation). *For the optimization problem* $Q_h(s,a) = r_h(s,a) + \inf_{P \in \Delta(S)} \big(\mathbb{E}_P[V_{h+1}] + \beta \mathrm{KL}\big(P\|P_h^o\big)\big)(s,a)$, *we have its dual formulation as follows*

$$Q_h = r_h - \beta \ln \mathbb{E}_{P_h^o}\big[e^{-\beta^{-1}V_{h+1}}\big]. \tag{D.19}$$

*Proof.* Consider the optimization problem

$$Q_h = r_h + \inf_{P \in \Delta(S)} \big(\mathbb{E}_P[V_{h+1}] + \beta \mathrm{KL}\big(P\|P_h^o\big)\big)$$

$$= r_h + \inf_{P \in \Delta(S)} \left(\sum_{s \in \mathcal{S}} P(s)V_{h+1}(s) + \beta P(s)\ln\frac{P(s)}{P_h^o(s)}\right).$$

The Lagrangian can be written as

$$\mathcal{L}(P,\eta) = \sum_{s \in S}\left[P(s)V_{h+1}(s) + \beta P(s)\ln\left(\frac{P(s)}{P_h^o(s)}\right)\right] - \sum_{s \in \mathcal{S}}\lambda(s)P(s) + \eta\left(1 - \sum_{s \in S}P(s)\right).$$

We set the derivative of $\mathcal{L}$ w.r.t. $P(s)$ to zero

$$\frac{\partial \mathcal{L}}{\partial P(s)} = V_{h+1}(s) + \beta \ln\left(\frac{P(s)}{P_h^o(s)}\right) + \beta - [\lambda(s) + \eta] = 0. \tag{D.20}$$

We denote $P'$ as the worst-case transition that satisfies (D.20), then we have

$$P'(s) = P_h^o(s)e^{-\beta^{-1}V_{h+1}(s)+\beta^{-1}[\lambda(s)+\eta]-1}, \tag{D.21}$$

$$\inf_P \mathcal{L}(P,\eta) = -\beta \mathbb{E}_{P_h^o}\left[e^{-\beta^{-1}V_{h+1}(s)+\beta^{-1}[\lambda(s)+\eta]-1}\right] + \eta.$$

We have

$$\begin{aligned}
Q_h &= r_h + \sup_{\lambda \geq 0, \eta}\left(\inf_P \mathcal{L}(P,\eta)\right) \\
&= r_h - \inf_{\lambda \geq 0, \eta}\left(\beta \mathbb{E}_{P_h^o}\left[e^{-\beta^{-1}V_{h+1}(s)+\beta^{-1}[\lambda(s)+\eta]-1}\right] - \eta\right) \\
&= r_h - \inf_\eta \left(\beta \mathbb{E}_{P_h^o}\left[e^{-\beta^{-1}V_{h+1}(s)+\beta^{-1}\eta-1}\right] - \eta\right) \tag{D.22} \\
&= r_h - \beta \ln \mathbb{E}_{P_h^o}\left[e^{-\beta^{-1}V_{h+1}}\right]. \tag{D.23}
\end{aligned}$$

where (D.22) holds because the result increases monotonically with respect to $\lambda$, (D.23) holds by calculating the derivation with respect to $\eta$ and thus setting $\eta = -\beta\left(\ln \mathbb{E}_{P_h^o}\left[e^{-\beta^{-1}V_{h+1}}\right] - 1\right)$. $\qquad\square$

Similar to Lemma C.2, we prove the optimism of estimation $Q^k$ and control $Q^* - Q^k$.

**Lemma D.6** (Optimism). If we set the bonus term as follows

$$\text{bonus}_h^k(s,a) = \left(1 + \beta e^{\beta^{-1}H}\sqrt{S}\right)\sqrt{\frac{2\ln(2SAHK/\delta)}{n_h^{k-1}(s,a) \vee 1}}, \tag{D.24}$$

then for any policy $\pi$ and any $(k,h,s,a) \in [K] \times [H] \times \mathcal{S} \times \mathcal{A}$, with probability at least $1 - 2\delta$, we have $Q_h^{k,\beta}(s,a) \geq Q_h^{\pi,\beta}(s,a)$. Specially, by setting $\pi = \pi^*$, we have $Q_h^{k,\beta}(s,a) \geq Q_h^{\pi^*,\beta}(s,a)$.

*Proof.* We prove this by induction. First, when $h = H + 1$, $Q_{H+1}^{k,\beta}(s,a) = 0 = Q_{H+1}^{\pi,\beta}(s,a)$ holds trivially.

Assume $Q_{h+1}^{k,\beta}(s,a) \geq Q_{h+1}^{\pi,\beta}(s,a)$ holds, since $\pi^k$ is the greedy policy, we have

$$V_{h+1}^{k,\beta}(s) = Q_{h+1}^{k,\beta}(s,\pi_{h+1}^k(s)) \geq Q_{h+1}^{k,\beta}(s,\pi_{h+1}(s)) \geq Q_{h+1}^{\pi,\beta}(s,\pi_{h+1}(s)) = V_{h+1}^{\pi,\beta}(s),$$

where the first inequality is because we choose $\pi^k$ as the greedy policy.

Recall that we denote $Q_h^{k,\beta}$ as the optimistic estimation in $k$-th episode, that is,

$$Q_h^{k,\beta}(s,a) = \text{bonus}_h^k(s,a) + \widehat{r}_h^k(s,a) + \inf_{P \in \Delta(S)}\left(\mathbb{E}_P\left[V_{h+1}^{k,\beta}\right] + \beta\text{KL}\left(P\|\widehat{P}_h^k\right)\right)(s,a).$$

We can infer that

$$\begin{aligned}
Q_h^{k,\beta} - Q_h^{\pi,\beta} &= \text{bonus}_h^k + \widehat{r}_h^k + \inf_{P \in \Delta(S)}\left(\mathbb{E}_P\left[V_{h+1}^{k,\beta}\right] + \beta\text{KL}\left(P\|\widehat{P}_h^k\right)\right) - r_h - \inf_{P \in \Delta(S)}\left(\mathbb{E}_P\left[V_{h+1}^{\pi,\beta}\right] + \beta\text{KL}\left(P\|P_h^o\right)\right) \\
&= \text{bonus}_h^k + \widehat{r}_h^k - r_h + \inf_{P \in \Delta(S)}\left(\mathbb{E}_P\left[V_{h+1}^{k,\beta}\right] + \beta\text{KL}\left(P\|\widehat{P}_h^k\right)\right) - \inf_{P \in \Delta(S)}\left(\mathbb{E}_P\left[V_{h+1}^{k,\beta}\right] + \beta\text{KL}\left(P\|P_h^o\right)\right) \\
&\quad + \inf_{P \in \Delta(S)}\left(\mathbb{E}_P\left[V_{h+1}^{k,\beta}\right] + \beta\text{KL}\left(P\|P_h^o\right)\right) - \inf_{P \in \Delta(S)}\left(\mathbb{E}_P\left[V_{h+1}^{\pi,\beta}\right] + \beta\text{KL}\left(P\|P_h^o\right)\right)
\end{aligned}$$

$$\geq \text{bonus}_h^k + \widehat{r}_h^k - r_h + \inf_{P \in \Delta(S)} \left( \mathbb{E}_P\big[V_{h+1}^{k,\beta}\big] - \mathbb{E}_P\big[V_{h+1}^{\pi,\beta}\big] \right)$$

$$+ \inf_{P \in \Delta(S)} \left( \mathbb{E}_P\big[V_{h+1}^{k,\beta}\big] + \beta \text{KL}\big(P\|\widehat{P}_h^k\big) \right) - \inf_{P \in \Delta(S)} \left( \mathbb{E}_P\big[V_{h+1}^{k,\beta}\big] + \beta \text{KL}\big(P\|P_h^o\big) \right) \tag{D.25}$$

$$\geq \text{bonus}_h^k + \widehat{r}_h^k - r_h + \inf_{P \in \Delta(S)} \left( \mathbb{E}_P\big[V_{h+1}^{k,\beta}\big] + \beta \text{KL}\big(P\|\widehat{P}_h^k\big) \right) - \inf_{P \in \Delta(S)} \left( \mathbb{E}_P\big[V_{h+1}^{k,\beta}\big] + \beta \text{KL}\big(P\|P_h^o\big) \right)$$

$$\tag{D.26}$$

$$= \text{bonus}_h^k + \widehat{r}_h^k - r_h + \beta \ln \mathbb{E}_{P_h^o}\big[e^{-\beta^{-1}V_{h+1}^{k,\beta}}\big] - \beta \ln \mathbb{E}_{\widehat{P}_h^k}\big[e^{-\beta^{-1}V_{h+1}^{k,\beta}}\big] \tag{D.27}$$

$$\geq \text{bonus}_h^k - \underbrace{\big|\widehat{r}_h^k - r_h\big|}_{(i)} - \underbrace{\beta\big|\ln \mathbb{E}_{\widehat{P}_h^k}\big[e^{-\beta^{-1}V_{h+1}^{k,\beta}}\big] - \ln \mathbb{E}_{P_h^o}\big[e^{-\beta^{-1}V_{h+1}^{k,\beta}}\big]\big|}_{(ii)}, \tag{D.28}$$

where (D.25) is from $\inf f(x) - \inf g(x) \geq \inf(f - g)(x)$, (D.26) is from the induction assumption, we plug in the dual formulation (D.19) in (D.27).

For term (i) in (D.28), from Lemma G.1 and a union bound, with probability at least $1 - \delta$, we have

$$\big|\widehat{r}_h^k(s,a) - r_h(s,a)\big| \leq \sqrt{\frac{\ln(2SAHK/\delta)}{2n_h^{k-1}(s,a) \vee 1}} \tag{D.29}$$

for any $(k,h,s,a) \in [K] \times [H] \times S \times A$.

For term (ii) in (D.28), following the same proof as (C.14) and a union bound, with probability at least $1 - \delta$, we have

$$\big| \ln \mathbb{E}_{\widehat{P}_h^k}\big[e^{-\beta^{-1}V_{h+1}^{k,\beta}}\big] - \ln \mathbb{E}_{P_h^o}\big[e^{-\beta^{-1}V_{h+1}^{k,\beta}}\big]\big| \leq e^{\beta^{-1}H} \sqrt{\frac{2S\ln(2SAHK/\delta)}{n_h^{k-1} \vee 1}} \tag{D.30}$$

for any $(k,h,s,a) \in [K] \times [H] \times S \times A$.

Apply the union bound again and combine (D.28) with (D.29), (D.30), the definition of bonus and induction assumption. With probability at least $1 - 2\delta$, we have $Q_h^{k,\rho}(s,a) \geq Q_h^{\pi,\rho}(s,a)$ for any $(k,h,s,a) \in [K] \times [H] \times S \times A$. This completes the proof. $\qquad\square$

With similar proof to Lemma C.13, we can control the item $Q^k - Q^{\pi^k}$.

**Lemma D.7.** For any $k \in [K]$, we can bound the estimation error as follows

$$Q_1^{k,\beta} - Q_1^{\pi^k,\beta} \leq \sum_{h=1}^{H} \mathbb{E}_{\{P_h^{w,k}\}_{h=1}^H, \pi^k}\big[2\text{bonus}_h^k\big].$$

*Proof.* From the proof of Lemma D.6, we see that

$$\big|\widehat{r}_h^k(s,a) - r_h(s,a)\big| + \big| \inf_{P \in \Delta(S)} \big(\mathbb{E}_P\big[V_{h+1}^{k,\beta}\big] + \beta \text{KL}\big(P\|\widehat{P}_h^k\big)\big)(s,a)$$

$$- \inf_{P \in \Delta(S)} \big(\mathbb{E}_P\big[V_{h+1}^{k,\beta}\big] + \beta \text{KL}\big(P\|P_h^o\big)\big)(s,a)\big| \leq \text{bonus}_h^k(s,a). \tag{D.31}$$

Recall that we define $P_h^{w,k} = \underset{P \in \Delta(S)}{\text{argmin}}\big(\mathbb{E}_P\big[V_{h+1}^{k,\beta}\big] + \beta \text{KL}\big(P\|P_h^o\big)\big)$ as the worst-case transition in Definition 5.3, we have

$$Q_h^{k,\beta} - Q_h^{\pi^k,\beta} = \text{bonus}_h^k + \widehat{r}_h^k + \inf_{P \in \Delta(S)} \big(\mathbb{E}_P\big[V_{h+1}^{k,\beta}\big] + \beta \text{KL}\big(P\|\widehat{P}_h^k\big)\big) - r_h - \inf_{P \in \Delta(S)} \big(\mathbb{E}_P\big[V_{h+1}^{\pi^k,\beta}\big] + \beta \text{KL}\big(P\|P_h^o\big)\big)$$

$$\leq 2\text{bonus}_h^k + \inf_{P \in \Delta(S)} \big(\mathbb{E}_P\big[V_{h+1}^{k,\beta}\big] + \beta \text{KL}\big(P\|P_h^o\big)\big) - \inf_{P \in \Delta(S)} \big(\mathbb{E}_P\big[V_{h+1}^{\pi^k,\beta}\big] + \beta \text{KL}\big(P\|P_h^o\big)\big) \tag{D.32}$$

$$= 2\text{bonus}_h^k + \mathbb{E}_{P_h^{w,k}, \pi^k}\big[Q_{h+1}^{k,\beta} - Q_{h+1}^{\pi^k,\beta}\big]. \tag{D.33}$$

where (D.32) uses (D.31). Apply (D.33) recursively, we can obtain the result. $\qquad\square$

**Theorem D.8** (Restatement of Theorem 5.16 in KL-divergence setting). For RRMDP with $(s,a)$-rectangular KL-divergence defined regularization term satisfying Assumption 5.5, with probability at least $1 - \delta$, the regret of Algorithm 1 satisfies

$$\text{Regret} = \widetilde{\mathcal{O}}\big(\big(1 + \beta e^{\beta^{-1}H}\sqrt{S}\big)\big(C_{vr}SAH + C_{vr}^{\frac{1}{2}}S^{\frac{1}{2}}A^{\frac{1}{2}}H\sqrt{K}\big)\big).$$

*Proof.* Setting $\delta' = \delta/3$ in Lemmas C.4 and D.6, then with probability at least $1 - \delta$, we get

$$\text{Regret} = \sum_{k=1}^{K}\big(V_1^{*,\beta} - V_1^{\pi^k,\beta}\big) = \sum_{k=1}^{K}\big(V_1^{*,\beta} - V_1^{k,\beta}\big) + \sum_{k=1}^{K}\big(V_1^{k,\beta} - V_1^{\pi^k,\beta}\big)$$

$$\leq \sum_{k=1}^{K}\sum_{h=1}^{H}\mathbb{E}_{P_h^{w,k},\pi^k}\big[2\text{bonus}_h^k\big] \tag{D.34}$$

$$= 2\sum_{k=1}^{K}\sum_{h=1}^{H}\mathbb{E}_{P_h^{w,k},\pi^k}\left[\big(1 + \beta e^{\beta^{-1}H}\sqrt{S}\big)\sqrt{\frac{2\ln(2SAHK/\delta)}{n_h^{k-1}(s,a)\vee 1}}\right] \tag{D.35}$$

$$= \widetilde{\mathcal{O}}\big(\big(1 + \beta e^{\beta^{-1}H}\sqrt{S}\big)\big(C_{vr}SAH + C_{vr}^{\frac{1}{2}}S^{\frac{1}{2}}A^{\frac{1}{2}}H\sqrt{K}\big)\big). \tag{D.36}$$

where (D.34) is the combination of Lemma D.6 and Lemma D.7, we plug in the bonus (D.24) in (D.35), (D.36) is from Lemma C.9. □

## D.3. Proof of Theorem 5.16 (Regularized $\chi^2$ Setting)

We first give the closed form solution of regularized $\chi^2$ update formulation. This dual formulation has also been proved by Panaganti et al. (2024), but our formulation is more concise.

**Lemma D.9** (Dual formulation). For the optimization problem $Q_h(s,a) = r_h(s,a) + \inf_{P\in\Delta(S)}\big(\mathbb{E}_P[V_{h+1}] + \beta\chi^2\big(P\|P_h^o\big)\big)(s,a)$, we have its dual formulation as follows

$$Q_h = r_h + \sup_{\boldsymbol{\lambda}\in[0,H]}\big(\mathbb{E}_{P_h^o}\big[V_{h+1} - \boldsymbol{\lambda}\big] - \frac{1}{4\beta}\text{Var}_{P_h^o}(V_{h+1} - \boldsymbol{\lambda})\big). \tag{D.37}$$

*Proof.* Consider the optimization problem

$$Q_h = r_h + \inf_{P\in\Delta(S)}\big(\mathbb{E}_P[V_{h+1}] + \beta\chi^2\big(P\|P_h^o\big)\big)$$

$$= r_h + \inf_{P\in\Delta(S)}\sum_{s\in S}\left(P(s)V_{h+1}(s) + \beta P_h^o(s)\Big(\frac{P(s)}{P_h^o(s)} - 1\Big)^2\right).$$

The Lagrangian can be written as

$$\mathcal{L}(P,\eta) = \sum_{s\in S}\left[P(s)V_{h+1}(s) + \beta P_h^o(s)\Big(\frac{P(s)}{P_h^o(s)} - 1\Big)^2\right] - \sum_{s\in\mathcal{S}}\boldsymbol{\lambda}(s)P(s) + \eta\Big(1 - \sum_{s\in S}P(s)\Big).$$

We set the derivative of $\mathcal{L}$ w.r.t. $P(s)$ to zero

$$\frac{\partial\mathcal{L}}{\partial P(s)} = V_{h+1}(s) + 2\beta\Big(\frac{P(s)}{P_h^o(s)} - 1\Big) - [\boldsymbol{\lambda}(s) + \eta] = 0. \tag{D.38}$$

We denote $P'$ as the worst-case transition that satisfies (D.38), then we have

$$P'(s) = P_h^o(s)\Big(1 - \frac{V_{h+1}(s) - [\boldsymbol{\lambda}(s) + \eta]}{2\beta}\Big), \tag{D.39}$$

$$\inf_P \mathcal{L}(P, \eta) = -\mathbb{E}_{P_h^o}\left[\frac{1}{4\beta}\left(V_{h+1}(s) - [\boldsymbol{\lambda}(s) + \eta]\right)^2 - \left(V_{h+1}(s) - [\boldsymbol{\lambda}(s) + \eta]\right)\right] + \eta.$$

We have

$$
\begin{aligned}
Q_h &= r_h + \sup_{\boldsymbol{\lambda} \geq 0, \eta}\left(\inf_P \mathcal{L}(P, \eta)\right) \\
&= r_h - \inf_{\boldsymbol{\lambda} \geq 0, \eta}\left(\mathbb{E}_{P_h^o}\left[\frac{1}{4\beta}\left(V_{h+1}(s) - [\boldsymbol{\lambda}(s) + \eta]\right)^2 - \left(V_{h+1}(s) - [\boldsymbol{\lambda}(s) + \eta]\right)\right] - \eta\right) \\
&= r_h + \sup_{\boldsymbol{\lambda} \geq 0}\left(\mathbb{E}_{P_h^o}[V_{h+1} - \boldsymbol{\lambda}] - \frac{1}{4\beta}\mathrm{Var}_{P_h^o}(V_{h+1} - \boldsymbol{\lambda})\right) \hspace{2cm} \text{(D.40)} \\
&= r_h + \sup_{\boldsymbol{\lambda} \in [0, H]}\left(\mathbb{E}_{P_h^o}[V_{h+1} - \boldsymbol{\lambda}] - \frac{1}{4\beta}\mathrm{Var}_{P_h^o}(V_{h+1} - \boldsymbol{\lambda})\right), \hspace{1cm} \text{(D.41)}
\end{aligned}
$$

where (D.40) holds by calculating the derivation with respect to $\eta$ and thus setting $\eta = \mathbb{E}_{P_h^o}[V_{h+1} - \boldsymbol{\lambda}]$, (D.41) holds because the result increases monotonically with respect to $\boldsymbol{\lambda}$ when $\boldsymbol{\lambda} \geq H$. $\qquad\square$

Similar to Lemma C.2, we prove the optimism of estimation $Q^k$ and control $Q^* - Q^k$.

**Lemma D.10** (Optimism). If we set the bonus term as follows

$$\mathrm{bonus}_h^k(s, a) = 5H^2\sqrt{\frac{2S^2\ln(48SAH^3K^2/\delta)}{n_h^{k-1}(s, a) \vee 1}} + \frac{2}{K}, \hspace{2cm} \text{(D.42)}$$

then for any policy $\pi$ and any $(k, h, s, a) \in [K] \times [H] \times \mathcal{S} \times \mathcal{A}$, with probability at least $1 - 3\delta$, we have $Q_h^{k,\beta}(s, a) \geq Q_h^{\pi,\beta}(s, a)$. Specially, by setting $\pi = \pi^*$, we have $Q_h^{k,\beta}(s, a) \geq Q_h^{\pi^*,\beta}(s, a)$.

*Proof.* We prove this by induction. First, when $h = H + 1$, $Q_{H+1}^{k,\beta}(s, a) = 0 = Q_{H+1}^{\pi,\beta}(s, a)$ holds trivially.

Assume $Q_{h+1}^{k,\beta}(s, a) \geq Q_{h+1}^{\pi,\beta}(s, a)$ holds, since $\pi^k$ is the greedy policy, we have

$$V_{h+1}^{k,\beta}(s) = Q_{h+1}^{k,\beta}(s, \pi_{h+1}^k(s)) \geq Q_{h+1}^{k,\beta}(s, \pi_{h+1}(s)) \geq Q_{h+1}^{\pi,\beta}(s, \pi_{h+1}(s)) = V_{h+1}^{\pi,\beta}(s),$$

where the first inequality is because we choose $\pi^k$ as the greedy policy.

Recall that we denote $Q_h^{k,\beta}$ as the optimistic estimation in $k$-th episode, that is,

$$Q_h^{k,\beta}(s, a) = \mathrm{bonus}_h^k(s, a) + \widehat{r}_h^k(s, a) + \inf_{P \in \Delta(S)}\left(\mathbb{E}_P[V_{h+1}^{k,\beta}] + \beta\chi^2(P\|\widehat{P}_h^k)\right)(s, a).$$

We can infer that

$$
\begin{aligned}
Q_h^{k,\beta} - Q_h^{\pi,\beta} &= \mathrm{bonus}_h^k + \widehat{r}_h^k + \inf_{P \in \Delta(S)}\left(\mathbb{E}_P[V_{h+1}^{k,\beta}] + \beta\chi^2(P\|\widehat{P}_h^k)\right) - r_h - \inf_{P \in \Delta(S)}\left(\mathbb{E}_P[V_{h+1}^{\pi,\beta}] + \beta\chi^2(P\|P_h^o)\right) \\
&= \mathrm{bonus}_h^k + \widehat{r}_h^k - r_h + \inf_{P \in \Delta(S)}\left(\mathbb{E}_P[V_{h+1}^{k,\beta}] + \beta\chi^2(P\|\widehat{P}_h^k)\right) - \inf_{P \in \Delta(S)}\left(\mathbb{E}_P[V_{h+1}^{k,\beta}] + \beta\chi^2(P\|P_h^o)\right) \\
&\quad + \inf_{P \in \Delta(S)}\left(\mathbb{E}_P[V_{h+1}^{k,\beta}] + \beta\chi^2(P\|P_h^o)\right) - \inf_{P \in \Delta(S)}\left(\mathbb{E}_P[V_{h+1}^{\pi,\beta}] + \beta\chi^2(P\|P_h^o)\right) \\
&\geq \mathrm{bonus}_h^k + \widehat{r}_h^k - r_h + \inf_{P \in \Delta(S)}\left(\mathbb{E}_P[V_{h+1}^{k,\beta}] - \mathbb{E}_P[V_{h+1}^{\pi,\beta}]\right) \\
&\quad + \inf_{P \in \Delta(S)}\left(\mathbb{E}_P[V_{h+1}^{k,\beta}] + \beta\chi^2(P\|\widehat{P}_h^k)\right) - \inf_{P \in \Delta(S)}\left(\mathbb{E}_P[V_{h+1}^{k,\beta}] + \beta\chi^2(P\|P_h^o)\right) \hspace{0.5cm} \text{(D.43)} \\
&\geq \mathrm{bonus}_h^k + \widehat{r}_h^k - r_h + \inf_{P \in \Delta(S)}\left(\mathbb{E}_P[V_{h+1}^{k,\beta}] + \beta\chi^2(P\|\widehat{P}_h^k)\right) - \inf_{P \in \Delta(S)}\left(\mathbb{E}_P[V_{h+1}^{k,\beta}] + \beta\chi^2(P\|P_h^o)\right)
\end{aligned}
$$
$$\text{(D.44)}$$

$$= \text{bonus}_h^k + \widehat{r}_h^k - r_h + \sup_{\boldsymbol{\lambda} \in [0,H]} \left( \mathbb{E}_{\widehat{P}_h^k} \left[ V_{h+1}^{k,\beta} - \boldsymbol{\lambda} \right] - \frac{1}{4\beta} \text{Var}_{\widehat{P}_h^k} \left( V_{h+1}^{k,\beta} - \boldsymbol{\lambda} \right) \right)$$

$$- \sup_{\boldsymbol{\lambda} \in [0,H]} \left( \mathbb{E}_{P_h^o} \left[ V_{h+1}^{k,\beta} - \boldsymbol{\lambda} \right] - \frac{1}{4\beta} \text{Var}_{P_h^o} \left( V_{h+1}^{k,\beta} - \boldsymbol{\lambda} \right) \right) \tag{D.45}$$

$$\geq \text{bonus}_h^k + \widehat{r}_h^k - r_h + \inf_{\boldsymbol{\lambda} \in [0,H]} \left\{ \left( \mathbb{E}_{\widehat{P}_h^k} \left[ V_{h+1}^{k,\beta} - \boldsymbol{\lambda} \right] - \frac{1}{4\beta} \text{Var}_{\widehat{P}_h^k} \left( V_{h+1}^{k,\beta} - \boldsymbol{\lambda} \right) \right) \right.$$

$$\left. - \left( \mathbb{E}_{P_h^o} \left[ V_{h+1}^{k,\beta} - \boldsymbol{\lambda} \right] - \frac{1}{4\beta} \text{Var}_{P_h^o} \left( V_{h+1}^{k,\beta} - \boldsymbol{\lambda} \right) \right) \right\}$$

$$\geq \text{bonus}_h^k - \left| \widehat{r}_h^k - r_h \right| - \sup_{\boldsymbol{\lambda} \in [0,H]} \left| \left( \mathbb{E}_{\widehat{P}_h^k} \left[ V_{h+1}^{k,\beta} - \boldsymbol{\lambda} \right] - \frac{1}{4\beta} \text{Var}_{\widehat{P}_h^k} \left( V_{h+1}^{k,\beta} - \boldsymbol{\lambda} \right) \right) \right.$$

$$\left. - \left( \mathbb{E}_{P_h^o} \left[ V_{h+1}^{k,\beta} - \boldsymbol{\lambda} \right] - \frac{1}{4\beta} \text{Var}_{P_h^o} \left( V_{h+1}^{k,\beta} - \boldsymbol{\lambda} \right) \right) \right|$$

$$\geq \text{bonus}_h^k - \underbrace{\left| \widehat{r}_h^k - r_h \right|}_{(i)} - \underbrace{\sup_{\boldsymbol{\lambda} \in [0,H]} \left| \mathbb{E}_{\widehat{P}_h^k} \left[ V_{h+1}^{k,\beta} - \boldsymbol{\lambda} \right] - \mathbb{E}_{P_h^o} \left[ V_{h+1}^{k,\beta} - \boldsymbol{\lambda} \right] \right|}_{(ii)}$$

$$- \underbrace{\frac{1}{4\beta} \sup_{\boldsymbol{\lambda} \in [0,H]} \left| \text{Var}_{\widehat{P}_h^k} \left( V_{h+1}^{k,\beta} - \boldsymbol{\lambda} \right) - \text{Var}_{P_h^o} \left( V_{h+1}^{k,\beta} - \boldsymbol{\lambda} \right) \right|}_{(iii)}, \tag{D.46}$$

where (D.43) is from $\inf f(x) - \inf g(x) \geq \inf(f - g)(x)$, (D.44) is from the induction assumption, we plug in the dual formulation (D.37) in (D.45).

For term (i) in (D.46), from Lemma G.1 and a union bound, with probability at least $1 - \delta$, we have

$$\left| \widehat{r}_h^k(s,a) - r_h(s,a) \right| \leq \sqrt{\frac{\ln(2SAHK/\delta)}{2n_h^{k-1}(s,a) \vee 1}}, \tag{D.47}$$

for any $(k, h, s, a) \in [K] \times [H] \times \mathcal{S} \times \mathcal{A}$.

To bound term (ii) in (D.46), following the same discussion as (C.63), with probability at least $1 - \delta$, we have

$$\sup_{\boldsymbol{\lambda} \in [0,H]} \left| \mathbb{E}_{\widehat{P}_h^k} \left[ V_{h+1}^{k,\rho} - \boldsymbol{\lambda} \right] - \mathbb{E}_{P_h^o} \left[ V_{h+1}^{k,\rho} - \boldsymbol{\lambda} \right] \right| \leq \sup_{V \in \mathcal{N}_{\mathcal{V}}(\epsilon)} \left| \mathbb{E}_{P_h^o}[V] - \mathbb{E}_{\widehat{P}_h^k}[V] \right| + 2\epsilon$$

$$\leq H \sqrt{\frac{2S^2 \ln(12SAH^2K^2/\delta)}{n_h^{k-1} \vee 1}} + \frac{1}{K}, \tag{D.48}$$

for any $(k, h, s, a) \in [K] \times [H] \times \mathcal{S} \times \mathcal{A}$.

We denote $V(\boldsymbol{\lambda}) = V_{h+1}^{k,\rho} - \boldsymbol{\lambda} \in [-H, H]$ and $\mathcal{V} = \left\{ V \in \mathbb{R}^S : \|V\|_\infty \leq H \right\}$. To bound term (iii) in (D.46), we create a $\epsilon$-net $\mathcal{N}_{\mathcal{V}}(\epsilon)$ for $\mathcal{V}$. From Lemma G.4, it holds that $\ln |\mathcal{N}_{\mathcal{V}}(\epsilon)| \leq |S| \cdot \ln(3H/\epsilon)$.

Therefore, by the definition of $\mathcal{N}_{\mathcal{V}}(\epsilon)$, for any fixed $V$, there exists a $V' \in \mathcal{N}_{\mathcal{V}}(\epsilon)$ such that $\|V - V'\|_\infty \leq \epsilon$, that is

$$\left| \text{Var}_{P_h^o}(V) - \text{Var}_{\widehat{P}_h^k}(V) \right| \leq \left| \text{Var}_{P_h^o}(V) - \text{Var}_{P_h^o}(V') \right| + \left| \text{Var}_{P_h^o}(V') - \text{Var}_{\widehat{P}_h^k}(V') \right| + \left| \text{Var}_{\widehat{P}_h^k}(V') - \text{Var}_{\widehat{P}_h^k}(V) \right|$$

$$\leq \left| \mathbb{E}_{P_h^o} \left[ V^2 - V'^2 \right] \right| + \left| \mathbb{E}_{P_h^o}^2[V] - \mathbb{E}_{P_h^o}^2[V'] \right| + \left| \mathbb{E}_{\widehat{P}_h^k} \left[ V'^2 - V^2 \right] \right| + \left| \mathbb{E}_{\widehat{P}_h^k}^2[V'] - \mathbb{E}_{\widehat{P}_h^k}^2[V] \right|$$

$$+ \left| \text{Var}_{P_h^o}(V') - \text{Var}_{\widehat{P}_h^k}(V') \right|$$

$$\leq \left| \text{Var}_{P_h^o}(V') - \text{Var}_{\widehat{P}_h^k}(V') \right| + 8H\epsilon$$

$$\leq \sup_{V' \in N_{\mathcal{V}}} \left| \text{Var}_{P_h^o}(V') - \text{Var}_{\widehat{P}_h^k}(V') \right| + 8H\epsilon. \tag{D.49}$$

For any fixed $V$, following the same analysis as (C.14), we have

$$\left| \text{Var}_{P_h^o}(V) - \text{Var}_{\widehat{P}_h^k}(V) \right| = \left| \left( \mathbb{E}_{P_h^o}[V^2] - \mathbb{E}_{P_h^o}^2[V] \right) - \left( \mathbb{E}_{\widehat{P}_h^k}[V^2] - \mathbb{E}_{\widehat{P}_h^k}^2[V] \right) \right|$$

$$\leq \left|\mathbb{E}_{P_h^o}[V^2] - \mathbb{E}_{\widehat{P}_h^k}[V^2]\right| + \left|\mathbb{E}_{P_h^o}^2[V] - \mathbb{E}_{\widehat{P}_h^k}^2[V]\right|$$

$$\leq H^2 \sqrt{\frac{2S \ln(2/\delta)}{n_h^{k-1} \vee 1}} + \left(\mathbb{E}_{P_h^o}[V] + \mathbb{E}_{\widehat{P}_h^k}[V]\right) \cdot \left|\mathbb{E}_{P_h^o}[V] - \mathbb{E}_{\widehat{P}_h^k}[V]\right|$$

$$\leq H^2 \sqrt{\frac{2S \ln(2/\delta)}{n_h^{k-1} \vee 1}} + 2H^2 \sqrt{\frac{2S \ln(2/\delta)}{n_h^{k-1} \vee 1}}$$

$$\leq 3H^2 \sqrt{\frac{2S \ln(2/\delta)}{n_h^{k-1} \vee 1}} \tag{D.50}$$

with probability at least $1 - \delta$.

Then with probability at least $1 - \delta$, we have

$$\sup_{\boldsymbol{\lambda} \in [0,H]} \left|\mathrm{Var}_{\widehat{P}_h^k}\left(V_{h+1}^{k,\beta} - \boldsymbol{\lambda}\right) - \mathrm{Var}_{P_h^o}\left(V_{h+1}^{k,\beta} - \boldsymbol{\lambda}\right)\right| \leq \sup_{\boldsymbol{\lambda} \in [0,H]} \left|\mathrm{Var}_{P_h^o}(V(\boldsymbol{\lambda})) - \mathrm{Var}_{\widehat{P}_h^k}(V(\boldsymbol{\lambda}))\right|$$

$$\leq \sup_{V \in \mathcal{N}_{\mathcal{V}}(\epsilon)} \left|\mathrm{Var}_{P_h^o}(V) - \mathrm{Var}_{\widehat{P}_h^k}(V)\right| + 8H\epsilon \tag{D.51}$$

$$\leq 3H^2 \sqrt{\frac{2S \ln(2SAHK|\mathcal{N}_{\mathcal{V}}|/\delta)}{n_h^{k-1} \vee 1}} + 8H\epsilon \tag{D.52}$$

$$\leq 3H^2 \sqrt{\frac{2S^2 \ln(6SAH^2K/\epsilon\delta)}{n_h^{k-1} \vee 1}} + 8H\epsilon$$

$$= 3H^2 \sqrt{\frac{2S^2 \ln(48SAH^3K^2/\delta)}{n_h^{k-1} \vee 1}} + \frac{1}{K}, \tag{D.53}$$

for any $(k,h,s,a) \in [K] \times [H] \times \mathcal{S} \times \mathcal{A}$, where (D.51) follows from (D.49), (D.52) is from (D.50) and a union bound, we set $\epsilon = \frac{1}{8HK}$ in (D.53).

Apply the union bound again and combine (D.46) with (D.47), (D.48), (D.53), the definition of bonus and induction assumption. With probability at least $1 - 3\delta$, we have $Q_h^{k,\rho}(s,a) \geq Q_h^{\pi,\rho}(s,a)$ for any $(k,h,s,a) \in [K] \times [H] \times \mathcal{S} \times \mathcal{A}$. This completes the proof. $\qquad\square$

With similar proof to Lemma C.3, we can control the item $Q^k - Q^{\pi^k}$.

**Lemma D.11.** For any $k \in [K]$, we can bound the estimation error as follows

$$Q_1^{k,\beta} - Q_1^{\pi^k,\beta} \leq \sum_{h=1}^{H} \mathbb{E}_{\{P_h^{w,k}\}_{h=1}^H, \pi^k}\left[2\mathrm{bonus}_h^k\right].$$

*Proof.* From the proof of Lemma D.2, we see that

$$\left|\widehat{r}_h^k(s,a) - r_h(s,a)\right| + \left|\inf_{P \in \Delta(S)} \left(\mathbb{E}_P\left[V_{h+1}^{k,\beta}\right] + \beta\chi^2\left(P\|\widehat{P}_h^k\right)\right)(s,a)\right.$$

$$\left. - \inf_{P \in \Delta(S)} \left(\mathbb{E}_P\left[V_{h+1}^{k,\beta}\right] + \beta\chi^2\left(P\|P_h^o\right)\right)(s,a)\right| \leq \mathrm{bonus}_h^k(s,a). \tag{D.55}$$

Recall that we define $P_h^{w,k} = \underset{P \in \Delta(S)}{\mathrm{argmin}} \left(\mathbb{E}_P\left[V_{h+1}^{k,\beta}\right] + \beta\chi^2\left(P\|P_h^o\right)\right)$ as the worst-case transition in Definition 5.3, we have

$$Q_h^{k,\beta} - Q_h^{\pi^k,\beta} = \mathrm{bonus}_h^k + \widehat{r}_h^k + \inf_{P \in \Delta(S)} \left(\mathbb{E}_P\left[V_{h+1}^{k,\beta}\right] + \beta\chi^2\left(P\|\widehat{P}_h^k\right)\right) - r_h - \inf_{P \in \Delta(S)} \left(\mathbb{E}_P\left[V_{h+1}^{\pi^k,\beta}\right] + \beta\chi^2\left(P\|P_h^o\right)\right)$$

$$\leq 2\mathrm{bonus}_h^k + \inf_{P \in \Delta(S)} \left(\mathbb{E}_P\left[V_{h+1}^{k,\beta}\right] + \beta\chi^2\left(P\|P_h^o\right)\right) - \inf_{P \in \Delta(S)} \left(\mathbb{E}_P\left[V_{h+1}^{\pi^k,\beta}\right] + \beta\chi^2\left(P\|P_h^o\right)\right) \tag{D.56}$$

$$= 2\text{bonus}_h^k + \mathbb{E}_{P_h^{w,k},\pi^k}\big[Q_{h+1}^{k,\beta} - Q_{h+1}^{\pi^k,\beta}\big]. \tag{D.57}$$

where (D.56) uses (D.55). Apply (D.57) recursively, we can obtain the result. $\qquad\square$

**Theorem D.12** (Restatement of Theorem 5.16 in $\chi^2$-divergence setting). For RRMDP with $(s,a)$-rectangular $\chi^2$-divergence defined regularization term satisfying Assumption 5.5, with probability at least $1 - \delta$, the regret of Algorithm 1 satisfies

$$\text{Regret} = \widetilde{\mathcal{O}}\big(C_{vr}S^2AH^3 + C_{vr}^{\frac{1}{2}}S^{\frac{3}{2}}A^{\frac{1}{2}}H^3\sqrt{K}\big).$$

*Proof.* Setting $\delta' = \delta/4$ in Lemmas C.4 and D.10, then with probability at least $1 - \delta$, we get

$$\text{Regret} = \sum_{k=1}^{K}\big(V_1^{*,\beta} - V_1^{\pi^k,\beta}\big) = \sum_{k=1}^{K}\big(V_1^{*,\beta} - V_1^{k,\beta}\big) + \sum_{k=1}^{K}\big(V_1^{k,\beta} - V_1^{\pi^k,\beta}\big)$$

$$\le \sum_{k=1}^{K}\sum_{h=1}^{H}\mathbb{E}_{P_h^{w,k},\pi^k}\big[2\text{bonus}_h^k\big] \tag{D.58}$$

$$= 2\sum_{k=1}^{K}\sum_{h=1}^{H}\mathbb{E}_{P_h^{w,k},\pi^k}\left[5H^2\sqrt{\frac{2S^2\ln(48SAH^3K^2/\delta)}{n_h^{k-1}(s,a)\vee 1}} + \frac{2}{K}\right] \tag{D.59}$$

$$= \widetilde{\mathcal{O}}\big(C_{vr}S^2AH^3 + C_{vr}^{\frac{1}{2}}S^{\frac{3}{2}}A^{\frac{1}{2}}H^3\sqrt{K}\big), \tag{D.60}$$

where (D.58) is the combination of Lemma D.10 and Lemma D.11, we plug in the bonus (D.42) in (D.59), (D.60) is from Lemma C.9. $\qquad\square$

# E. Proofs of Results in Lower Bounds

## E.1. Proof of Lower Bounds on the Regret

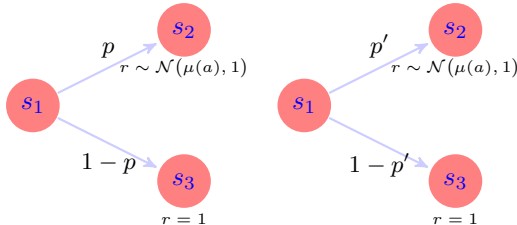

(a) The nominal RMDP.     (b) The perturbed RMDP.

*Figure 6.* Constructions of the nominal environment and the worst-case environment in the proofs of regret lower bounds, the value on each arrow represents the transition probability.

**Theorem E.1** (Combination of Theorem 5.12 and Theorem 5.18). For CRMDPs with TV, KL and $\chi^2$ divergence defined uncertainty sets and RRMDPs with TV, KL and $\chi^2$ divergence defined regularization terms, for any learning algorithm $\xi$, there exists a RMDP $\mathcal{M}$ satisfying Assumption 5.5, such that $\text{Regret}_{\mathcal{M}}(\xi, K) = \Omega\big(C_{vr}^{\frac{1}{2}}\sqrt{K}\big)$.

*Proof.* The proof here follows the high level idea in Lattimore & Szepesvári (2020, Theorem 15.2). We consider two RMDPs $\mathcal{M}_1$ and $\mathcal{M}_2$, as illustrated in Figure 6, where $H = 2$, $\mathcal{S} = \{s_1, s_2, s_3\}$ and $\mathcal{A} = \{a_1, a_2, \cdots, a_{|\mathcal{A}|}\}$. The only difference between these two RMDPs is the mean reward at $s_2$. $s_1$ is always the initial state, and can transit to $s_2$ with fixed probability $p$ and $s_3$ with fixed probability $1 - p$ regardless of the action. We assume that $K > |\mathcal{A}|$ and $p \ge |\mathcal{A}|/K$ to facilitate the constructions, this bound becomes loose as $K$ grows large.

The agent receives a reward drawn from a unit normal distribution, $r \sim \mathcal{N}(\mu(a), 1)$, at state $s_2$, and a fixed reward $r = 1$ at state $s_3$, where $\mu(a) \in [0, 1)$. This choice of $\mu(a)$ ensures that the robust value function satisfies $V^{\pi,\rho}(s_2) < V^{\pi,\rho}(s_3)$,

implying that the transition probability from $s_1$ to $s_2$ will not decrease in the worst-case environment compared to the nominal environment. In $\mathcal{M}_1$, the mean reward vector at $s_2$ is given by $\mu_1 = (\Delta, 0, \cdots, 0) \in \mathbb{R}^{|\mathcal{A}|}$ and $\Delta < \frac{1}{2}$ is a constant to be specified.

We introduce some notations that will be useful in the following discussion. The agent follows a learning algorithm $\xi$ and, in the $k$-th episode, selects a policy $\pi^k$ to interact with the environment and collect a trajectory $(o^k, a^k, r^k)$, where $o^k$ denotes the state to which the agent transitions from $s_1$ after taking an arbitrary action; $a^k$ denotes the action the agent takes at step $h = 2$, i.e., at state $s_2$ or $s_3$; and $r^k$ denotes the corresponding reward received at $s_2$ or $s_3$, all in episode $k$. The joint distribution over the trajectories collected across all $K$ episodes, $\{(o^k, a^k, r^k)\}_{k=1}^K$, induces a distribution denoted by $\mathbb{P}_i^o$.

The random variables corresponding to $o^k, a^k, r^k$ are denoted by $O^k, A^k, R^k$, respectively. We use $\mathbb{E}_i^o$ to denote the expectation under $\mathbb{P}_i^o$. As similarly noted by Auer et al. (2002), it suffices to consider deterministic strategies without loss of generality. Let $S_j(K)$ denote the number of episodes in which the agent selects action $a_j$ if visiting state $s_2$ according to the learned policy

$$S_j(K) = \sum_{k=1}^K \mathbb{1}\{\pi^k(s_2) = a_j\}, \tag{E.1}$$

and let $T_j(K)$ denote the number of episodes in which action $a_j$ is actually taken at state $s_2$,

$$T_j(K) = \sum_{k=1}^K \mathbb{1}\{O^k = s_2, A^k = a_j\}. \tag{E.2}$$

We set $c = \underset{j > 1}{\arg\min}\, \mathbb{E}_1^o[T_j(n)]$ as the least chosen action (excluding $a_1$) under environment $\mathcal{M}_1$. Since the expected number of times the agent transitions from $s_1$ to $s_2$ over $K$ episodes is $Kp$, and given the selection of $c$, it follows that

$$\mathbb{E}_1^o[T_c(n)] \leq \frac{Kp}{|\mathcal{A}| - 1}. \tag{E.3}$$

We now define the mean reward at $s_2$ in environment $\mathcal{M}_2$ as

$$\mu_2 = (\Delta, 0, \cdots, 0, \underset{c\text{-th}}{2\Delta}, 0, \cdots, 0).$$

That is, $\mu_2(a_j) = \mu_1(a_j)$ for all $j \neq c$ and $\mu_2(a_c) = 2\Delta$. Clearly, the optimal policy at state $s_2$ selects action $a_1$ in $\mathcal{M}_1$ and action $a_c$ in $\mathcal{M}_2$.

Recall the definition of $C_{vr}$ in Assumption 5.5 and the visitation measure notations introduced in Definition 5.3. Since the visitation measure to $s_1$ is always 1, and the visitation measure to $s_3$, which is equivalently the transition probability from $s_1$ to $s_3$, does not increase in the worst-case environment compared to the nominal one, the maximum visitation measure ratio can only be attained at state $s_2$. In the nominal environment, the visitation measure to $s_2$ is $d^\pi(s_2) = p$ for any policy $\pi$. In the worst-case environment of $\mathcal{M}_1$, let $\widetilde{p} = \max_\pi q^\pi(s_2)$ denote the maximum visitation measure to $s_2$, that is, the largest transition probability from $s_1$ to $s_2$ across all policies, under both the CRMDP and RRMDP settings. In what follows, we show that this value $\widetilde{p}$ remains unchanged in $\mathcal{M}_2$.

To proceed, we present the following lemma, which provides a lower bound on the sum of the total regret in environments $\mathcal{M}_1$ and $\mathcal{M}_2$.

**Lemma E.2.** For TV, KL and $\chi^2$ divergence defined CRMDP and RRMDP settings, there exists a corresponding constant $\zeta$ such that

$$\mathbb{E}\big[\text{Regret}_{\mathcal{M}_1}(\xi, K) + \text{Regret}_{\mathcal{M}_2}(\xi, K)\big] \geq \zeta \cdot \widetilde{p} \cdot \exp\big(-\text{KL}\big(\mathbb{P}_1^o, \mathbb{P}_2^o\big)\big) K\Delta, \tag{E.4}$$

$$\text{where} \quad \zeta = \begin{cases} \frac{1}{4} & \text{for CRMDP settings,} \\ \frac{1}{4} & \text{for RRMDP-TV setting,} \\ \frac{e^{-\beta^{-1}}}{4(1+\beta^{-1})} & \text{for RRMDP-KL setting,} \\ \frac{1}{16} & \text{for RRMDP-}\chi^2 \text{ setting.} \end{cases}$$

We present the following lemma to simplify the expression for $\mathrm{KL}\big(\mathbb{P}_1^o, \mathbb{P}_2^o\big)$.

**Lemma E.3.** For the distributions $\mathbb{P}_1^o$ and $\mathbb{P}_2^o$, the following property holds:

$$\mathrm{KL}\big(\mathbb{P}_1^o, \mathbb{P}_2^o\big) = \sum_{j=1}^{|\mathcal{A}|} \mathbb{E}_1^o[T_j(K)] \cdot \mathrm{KL}\big(P_{r_{\mathcal{M}_1}(s_2, a_j)}, P_{r_{\mathcal{M}_2}(s_2, a_j)}\big), \tag{E.5}$$

where $T_j(K)$ is defined in (E.2).

From the construction of reward function, we can simplify (E.5) as

$$\begin{aligned}
\mathrm{KL}\big(\mathbb{P}_1^o, \mathbb{P}_2^o\big) &= \sum_{j=1}^{|\mathcal{A}|} \mathbb{E}_1^o[T_j(K)] \cdot \mathrm{KL}(r_{\mathcal{M}_1}(s_2, a_j), r_{\mathcal{M}_2}(s_2, a_j)) \\
&= \mathbb{E}_1^o[T_c(K)] \cdot \mathrm{KL}(\mathcal{N}(0, 1), \mathcal{N}(2\Delta, 1)) \\
&= 2\Delta^2 \cdot \mathbb{E}_1^o[T_c(K)] \tag{E.6} \\
&\leq \frac{2\Delta^2 K p}{|\mathcal{A}| - 1}, \tag{E.7}
\end{aligned}$$

where (E.6) follows from Lemma G.7, and (E.7) applies the result of (E.3).

We now return to bounding (E.4). By plugging in (E.7) and setting $\Delta = \sqrt{(|\mathcal{A}| - 1)/(4Kp)}$, which satisfies $\Delta < 1/2$ due to the given range of $K$ and $p$, we obtain

$$\begin{aligned}
2\max\big\{\mathbb{E}\big[\mathrm{Regret}_{\mathcal{M}_1}(\xi, K)\big], \mathbb{E}\big[\mathrm{Regret}_{\mathcal{M}_2}(\xi, K)\big]\big\} &\geq \mathbb{E}\big[\mathrm{Regret}_{\mathcal{M}_1}(\xi, K) + \mathrm{Regret}_{\mathcal{M}_2}(\xi, K)\big] \\
&\geq \zeta \cdot \widetilde{p} \cdot \exp\big(-\mathrm{KL}\big(\mathbb{P}_1^o, \mathbb{P}_2^o\big)\big) K\Delta \\
&\geq \zeta \cdot \widetilde{p} \cdot \exp\Big(-\frac{2\Delta^2 K p}{|\mathcal{A}| - 1}\Big) K\Delta \\
&\geq \Omega\Big(\frac{e^{-\frac{1}{2}}\sqrt{\widetilde{p}}}{2}\sqrt{C_{vr}(|\mathcal{A}| - 1)K}\Big) \tag{E.8} \\
&= \Omega\big(C_{vr}^{\frac{1}{2}}\sqrt{K}\big).
\end{aligned}$$

This finishes the proof. $\qquad\square$

## E.2. Proof of Lemma 5.14

To establish the results for hard instances, we instantiate the constructed example in Theorem E.1 by appropriately selecting the nominal transition $p$ and the uncertainty set radius $\rho$. We assume that $K \geq 2^{2|\mathcal{A}|+1}|\mathcal{A}|$ throughout the proof.

**For TV-distance.** We set $p = \frac{1}{2^{2|\mathcal{A}|+1} - 2} \geq |\mathcal{A}|/K$ and $\rho = \frac{1}{2}$. First, we solve the worst-case transition in the first step explicitly. Follow the definition of TV-distance, we have

$$\widetilde{p} = \operatorname*{argmin}_{p'} p' V^\pi(s_2) + (1 - p') \tag{E.9}$$
$$\text{s.t.} \quad p' - p \leq \rho.$$

Since (E.9) decrease monotonically with $\widetilde{p}$, the solution of this optimization problem is $\widetilde{p} = p + \rho = \frac{2^{2|\mathcal{A}|}}{2^{2|\mathcal{A}|+1} - 2}$. Therefore, $C_{vr} = \widetilde{p}/p = 2^{2|\mathcal{A}|}$.

From (E.8), we see that the expected regret for constructed environments $\mathcal{M}_1$ and $\mathcal{M}_2$ satisfies

$$\max\big\{\mathbb{E}\big[\mathrm{Regret}_{\mathcal{M}_1}(\xi, K)\big], \mathbb{E}\big[\mathrm{Regret}_{\mathcal{M}_2}(\xi, K)\big]\big\} \geq \Omega\Big(\frac{e^{-\frac{1}{2}}\sqrt{\widetilde{p}}}{2}\sqrt{C_{vr}(|\mathcal{A}| - 1)K}\Big) = \Omega\big(2^{|\mathcal{A}|}\sqrt{K}\big).$$

This finishes the proof.

**For $\chi^2$-divergence.** We set $p = \frac{1}{2^{2|\mathcal{A}|+1}} \geq |\mathcal{A}|/K$ and $\rho = \frac{(2^{2|\mathcal{A}|}-1)^2}{2^{2|\mathcal{A}|+1}-1}$. First, we solve the worst-case transition in the first step explicitly. Follow the definition of $\chi^2$-divergence, we have

$$\widetilde{p} = \operatorname*{argmin}_{p'} p'V^\pi(s_2) + (1 - p') \tag{E.10}$$

$$\text{s.t.} \quad p\left(\frac{p'}{p} - 1\right)^2 + (1 - p)\left(\frac{1 - p'}{1 - p} - 1\right)^2 \leq \rho.$$

Since (E.10) decrease monotonically with $\widetilde{p}$, the solution of this optimization problem is $\widetilde{p} = p + \sqrt{\rho p(1 - p)} = \frac{1}{2}$. Therefore, $C_{vr} = \widetilde{p}/p = 2^{2|\mathcal{A}|}$.

From (E.8), we see that the expected regret for constructed environments $\mathcal{M}_1$ and $\mathcal{M}_2$ satisfies

$$\max\left\{\mathbb{E}\big[\mathrm{Regret}_{\mathcal{M}_1}(\xi, K)\big], \mathbb{E}\big[\mathrm{Regret}_{\mathcal{M}_2}(\xi, K)\big]\right\} \geq \Omega\left(\frac{e^{-\frac{1}{2}}\sqrt{\widetilde{p}}}{2}\sqrt{C_{vr}(|\mathcal{A}| - 1)K}\right) = \Omega\big(2^{|\mathcal{A}|}\sqrt{K}\big).$$

This finishes the proof.

**For KL-divergence.** We set $p = \frac{1}{2^{2|\mathcal{A}|+1}} \geq |\mathcal{A}|/K$ and $\rho = \frac{(2^{2|\mathcal{A}|}-1)^2}{2^{2|\mathcal{A}|+1}-1}$. First, we solve the worst-case transition in the first step explicitly. Follow the definition of KL-divergence, we have

$$\widetilde{p} = \operatorname*{argmin}_{p'} p'V^\pi(s_2) + (1 - p')$$

$$\text{s.t.} \quad p'\log\left(\frac{p'}{p}\right) + (1 - p)\log\left(\frac{1 - p'}{1 - p}\right) \leq \rho.$$

Though it is difficult to obtain a closed-form solution for this optimization problem, Lemma G.8 implies that its optimal value is no less than that of the previous $\chi^2$-based optimization problem. That is, $\widetilde{p} \geq p + \sqrt{\rho p(1 - p)} = \frac{1}{2}$. Therefore, $C_{vr} = \widetilde{p}/p \geq 2^{2|\mathcal{A}|}$.

From (E.8), we see that the expected regret for constructed environments $\mathcal{M}_1$ and $\mathcal{M}_2$ satisfies

$$\max\left\{\mathbb{E}\big[\mathrm{Regret}_{\mathcal{M}_1}(\xi, K)\big], \mathbb{E}\big[\mathrm{Regret}_{\mathcal{M}_2}(\xi, K)\big]\right\} \geq \Omega\left(\frac{e^{-\frac{1}{2}}\sqrt{\widetilde{p}}}{2}\sqrt{C_{vr}(|\mathcal{A}| - 1)K}\right) = \Omega\big(2^{|\mathcal{A}|}\sqrt{K}\big).$$

This finishes the proof.

## F. Proofs of Supporting Lemmas in the Proofs of Lower Bounds

### F.1. Proof of Lemma E.2

We provide the proofs for the three CRMDP settings in Appendix F.1.1, as they share a similar underlying structure. The proofs for the RRMDP-TV, RRMDP-KL, and RRMDP-$\chi^2$ settings are given in Appendix F.1.2, Appendix F.1.3, and Appendix F.1.4, respectively.

#### F.1.1. THE CRMDP SETTINGS

First, we consider the optimization problem for the worst-case transition from state $s_1$ to states $s_2$ and $s_3$, given by

$$P^w = \operatorname*{argmin}_{p':\mathrm{D}(P^w\|P^o)\leq\rho} \big[p' \cdot V^{\pi,\rho}(s_2) + (1 - p') \cdot V^{\pi,\rho}(s_3)\big]$$

$$= \operatorname*{argmin}_{p':\mathrm{D}(P^w\|P^o)\leq\rho} \big[p' \cdot V^{\pi,\rho}(s_2) + (1 - p')\big],$$

$$\text{where} \quad P^w = (p', 1 - p') \quad \text{and} \quad P^o = (p, 1 - p),$$

where the second equality holds because $V^{\pi,\rho}(s_2) < 1$ and $V^{\pi,\rho}(s_3) = 1$ by construction. As a result, the objective function $p' \cdot V^{\pi,\rho}(s_2) + (1 - p')$ decreases monotonically with respect to $p'$. Therefore, the worst-case transition probability is characterized by $\widetilde{p} = \sup\{p' : \mathrm{D}(P^w \| P^o) \leq \rho\}$, where D denotes an $f$-divergence (TV, KL or $\chi^2$). This implies that $\widetilde{p}$ remains invariant across different policies $\pi$ and environments $\mathcal{M}_1$ and $\mathcal{M}_2$.

For environment $\mathcal{M}_1$, the total regret incurred by selecting a suboptimal policy $\pi^k(s_2) \neq a_1$ in each episode can be bounded as

$$\mathbb{E}\big[\text{Regret}_{\mathcal{M}_1}(\xi, K)\big] = \sum_{k=1}^{K} \mathbb{E}_1^o\big[V^{\pi^*,\rho}(s_1) - V^{\pi^k,\rho}(s_1)\big]$$

$$= \sum_{k=1}^{K} \mathbb{E}_1^o\big[\big(\widetilde{p} \cdot V^{\pi^*,\rho}(s_2) + (1 - \widetilde{p}) \cdot V^{\pi^*,\rho}(s_3)\big) - \big(\widetilde{p} \cdot V^{\pi^k,\rho}(s_2) + (1 - \widetilde{p}) \cdot V^{\pi^k,\rho}(s_3)\big)\big] \quad \text{(F.1)}$$

$$= \sum_{k=1}^{K} \mathbb{E}_1^o\big[\widetilde{p} \cdot \big(V^{\pi^*,\rho}(s_2) - V^{\pi^k,\rho}(s_2)\big)\big]$$

$$= \sum_{k=1}^{K} \mathbb{E}_1^o\big[\widetilde{p} \cdot \mathbb{1}\{\pi^k(s_2) \neq a_1\}\Delta\big] \quad \text{(F.2)}$$

$$= \widetilde{p} \cdot \Delta \cdot \mathbb{E}_1^o\bigg[\sum_{k=1}^{K} \mathbb{1}\{\pi^k(s_2) \neq a_1\}\bigg]$$

$$= \widetilde{p} \cdot \Delta \cdot \mathbb{E}_1^o[K - S_1(K)] \quad \text{(F.3)}$$

$$\geq \frac{\widetilde{p}K\Delta}{2}\mathbb{P}_1^o\Big(S_1(K) \leq \frac{K}{2}\Big), \quad \text{(F.4)}$$

where (F.1) follows from the definition of the robust value function under the CRMDP setting, using $\widetilde{p}$ as the worst-case transition probability, (F.2) relies on the construction of the reward function at state $s = s_2$, (F.3) uses the definition of $S_j(K)$ as given in (E.1), and (F.4) follows from Markov's inequality.

For environment $\mathcal{M}_2$, following the same analysis, the total regret incurred by selecting a suboptimal policy $\pi^k(s_2) \neq a_c$ in each episode can be bounded as

$$\mathbb{E}\big[\text{Regret}_{\mathcal{M}_2}(\xi, K)\big] = \sum_{k=1}^{K} \mathbb{E}_2^o\big[V^{\pi^*,\rho}(s_1) - V^{\pi^k,\rho}(s_1)\big]$$

$$= \sum_{k=1}^{K} \mathbb{E}_2^o\big[\widetilde{p} \cdot \big(V^{\pi^*,\rho}(s_2) - V^{\pi^k,\rho}(s_2)\big)\big]$$

$$= \sum_{k=1}^{K} \mathbb{E}_2^o\bigg[\widetilde{p} \cdot \bigg(\mathbb{1}\{\pi^k(s_2) = a_1\}\Delta + \sum_{j>1, j\neq c} \mathbb{1}\{\pi^k(s_2) = a_j\}\Delta\bigg)\bigg]$$

$$\geq \sum_{k=1}^{K} \mathbb{E}_2^o\big[\widetilde{p} \cdot \big(\mathbb{1}\{\pi^k(s_2) = a_1\}\Delta\big)\big]$$

$$= \widetilde{p} \cdot \Delta \cdot \mathbb{E}_2^o\bigg[\sum_{k=1}^{K} \mathbb{1}\{\pi^k(s_2) = a_1\}\bigg]$$

$$= \widetilde{p} \cdot \Delta \cdot \mathbb{E}_2^o[S_1(K)]$$

$$\geq \frac{\widetilde{p}K\Delta}{2}\mathbb{P}_2^o\Big(S_1(K) > \frac{K}{2}\Big). \quad \text{(F.5)}$$

By combining (F.4) and (F.5) with Lemma G.6, we obtain

$$\mathbb{E}\big[\text{Regret}_{\mathcal{M}_1}(\xi, K) + \text{Regret}_{\mathcal{M}_2}(\xi, K)\big] \geq \frac{\widetilde{p}K\Delta}{2}\left(\mathbb{P}_1^o\Big(S_1(K) \leq \frac{K}{2}\Big) + \mathbb{P}_1^o\Big(S_2(K) > \frac{K}{2}\Big)\right)$$

$$\geq \frac{\widetilde{p}K\Delta}{4}\exp\big(-\text{KL}\big(\mathbb{P}_1^o, \mathbb{P}_2^o\big)\big).$$

We complete the proof by setting $\zeta = \frac{1}{4}$.

### F.1.2. THE RRMDP-TV SETTING

For the RRMDP-TV setting, the robust value function at state $s_1$, denoted by $V^{\pi,\beta}(s_1)$, is defined as

$$V^{\pi,\beta}(s_1) = \inf_{p'}\big\{p' \cdot V^{\pi,\beta}(s_2) + (1-p') \cdot V^{\pi,\beta}(s_3) + \beta|p' - p|\big\}$$

$$= \inf_{p'}\big\{p' \cdot V^{\pi,\beta}(s_2) + (1-p') + \beta(p' - p)\big\},$$

where the second equality holds because $V^{\pi,\beta}(s_2) < 1$ and $V^{\pi,\beta}(s_3) = 1$ by construction. Given that $V_1^{\pi,\beta}(s_2)$ takes value in $\{0, \Delta, 2\Delta\}$, we set $\beta \in [0, 1-2\Delta]$, so that $V_2^{\pi,\beta}(s_2) - 1 + \beta \leq 0$. Therefore, the term $\big[V_2^{\pi,\beta}(s_2) - 1 + \beta\big]p'$ decreases monotonically with $p'$, and the infimum is attained at $\widetilde{p} = 1$. Consequently, we obtain

$$V^{\pi,\beta}(s_1) = V^{\pi,\beta}(s_2) + \beta(1-p).$$

For environment $\mathcal{M}_1$, the total regret incurred by selecting a suboptimal policy $\pi^k(s_2) \neq a_1$ in each episode can be bounded as

$$\mathbb{E}\big[\text{Regret}_{\mathcal{M}_1}(\xi, K)\big] = \sum_{k=1}^{K}\mathbb{E}_1^o\big[V^{\pi^*,\beta}(s_1) - V^{\pi^k,\beta}(s_1)\big]$$

$$= \sum_{k=1}^{K}\mathbb{E}_1^o\big[\mathbb{1}\{\pi^k(s_2) \neq a_1\} \cdot \big((\Delta + \beta(1-p)) - \beta(1-p)\big)\big]$$

$$= \widetilde{p} \cdot \Delta \cdot \mathbb{E}_1^o\left[\sum_{k=1}^{K}\mathbb{1}\{\pi^k(s_2) \neq a_1\}\right] \tag{F.6}$$

$$= \widetilde{p} \cdot \Delta \cdot \mathbb{E}_1^o[K - S_1(K)] \tag{F.7}$$

$$\geq \frac{\widetilde{p}K\Delta}{2}\mathbb{P}_1^o\Big(S_1(K) \leq \frac{K}{2}\Big), \tag{F.8}$$

where (F.6) holds because $\widetilde{p} = 1$ as previously derived, (F.7) uses the definition of $S_j(K)$ as given in (E.1), and (F.8) follows from Markov's inequality.

For environment $\mathcal{M}_2$, following the same analysis, the total regret incurred by selecting a suboptimal policy $\pi^k(s_2) \neq a_c$ in each episode can be bounded as

$$\mathbb{E}\big[\text{Regret}_{\mathcal{M}_2}(\xi, K)\big] = \sum_{k=1}^{K}\mathbb{E}_2^o\big[V^{\pi^*,\beta}(s_1) - V^{\pi^k,\beta}(s_1)\big]$$

$$= \sum_{k=1}^{K}\mathbb{E}_2^o\Big[\mathbb{1}\{\pi^k(s_2) = a_1\} \cdot \big((2\Delta + \beta(1-p)) - (\Delta + \beta(1-p))\big)$$

$$+ \sum_{j>1, j\neq c}\mathbb{1}\{\pi^k(s_2) = a_j\} \cdot \big((2\Delta + \beta(1-p)) - \beta(1-p)\big)\Big]$$

$$\geq \widetilde{p} \cdot \Delta \cdot \mathbb{E}_2^o\left[\sum_{k=1}^{K}\mathbb{1}\{\pi^k(s_2) = a_1\}\right]$$

$$= \widetilde{p} \cdot \Delta \cdot \mathbb{E}_2^o[S_1(K)]$$

$$\geq \frac{\widetilde{p}K\Delta}{2}\mathbb{P}_2^o\Big(S_1(K) > \frac{K}{2}\Big). \tag{F.9}$$

By combining (F.4) and (F.5) with Lemma G.6, we obtain

$$\mathbb{E}\big[\mathrm{Regret}_{\mathcal{M}_1}(\xi, K) + \mathrm{Regret}_{\mathcal{M}_2}(\xi, K)\big] \geq \frac{\widetilde{p}K\Delta}{2}\Big(\mathbb{P}_1^o\Big(S_1(K) \leq \frac{K}{2}\Big) + \mathbb{P}_2^o\Big(S_1(K) > \frac{K}{2}\Big)\Big)$$

$$\geq \frac{\widetilde{p}K\Delta}{4}\exp\big(-\mathrm{KL}\big(\mathbb{P}_1^o, \mathbb{P}_2^o\big)\big).$$

We complete the proof by setting $\zeta = \frac{1}{4}$.

### F.1.3. THE RRMDP-KL SETTING

For the RRMDP-KL setting, the robust value function at state $s_1$, denoted by $V^{\pi,\beta}(s_1)$, is defined as

$$V^{\pi,\beta}(s_1) = \inf_{p'}\left\{p' \cdot V^{\pi,\beta}(s_2) + (1-p') \cdot V^{\pi,\beta}(s_3) + \beta\Big[p'\log\Big(\frac{p'}{p}\Big) + (1-p')\log\Big(\frac{1-p'}{1-p}\Big)\Big]\right\}. \tag{F.10}$$

Using the construction that $V^{\pi,\beta}(s_3) = 1$ and setting the derivative of the objective function in (F.10) with respect to $p'$ to zero, we obtain

$$p' = \frac{pe^{-\beta^{-1}V^{\pi,\beta}(s_2)}}{pe^{-\beta^{-1}V^{\pi,\beta}(s_2)} + (1-p)e^{-\beta^{-1}}}. \tag{F.11}$$

The worst-case transition probability $\widetilde{p}$ is defined as the maximum value of $p'$ over all policies $\pi$. Since (F.11) is monotonically decreasing in $V^{\pi,\beta}(s_2)$, we set $V^{\pi,\beta}(s_2) = 0$ in (F.11) to obtain

$$\widetilde{p} = \frac{p}{p + (1-p)e^{-\beta^{-1}}}. \tag{F.12}$$

Finally, substituting (F.11) back into (F.10), we derive the expression for the robust value function $V^{\pi,\beta}(s_1)$ as

$$V^{\pi,\beta}(s_1) = -\beta\log\Big(pe^{-\beta^{-1}V^{\pi,\beta}(s_2)} + (1-p)e^{-\beta^{-1}}\Big). \tag{F.13}$$

For the environment $\mathcal{M}_1$, if $\pi^k(s_2) \neq a_1$, then the episodic regret can be bounded by

$$V^{\pi^*,\beta}(s_1) - V^{\pi^k,\beta}(s_1) = -\beta\log\Big(pe^{-\beta^{-1}\Delta} + (1-p)e^{-\beta^{-1}}\Big) + \beta\log\Big(p + (1-p)e^{-\beta^{-1}}\Big)$$

$$= -\beta\log\Big(\frac{pe^{-\beta^{-1}\Delta} + (1-p)e^{-\beta^{-1}}}{p + (1-p)e^{-\beta^{-1}}}\Big)$$

$$= -\beta\log\Big(1 + \frac{p(e^{-\beta^{-1}\Delta} - 1)}{p + (1-p)e^{-\beta^{-1}}}\Big)$$

$$\geq \beta\Big(\frac{p}{p + (1-p)e^{-\beta^{-1}}}\Big)(1 - e^{-\beta^{-1}\Delta}) \tag{F.14}$$

$$\geq \beta\Big(\frac{p}{p + (1-p)e^{-\beta^{-1}}}\Big)\Big(\frac{\beta^{-1}\Delta}{1 + \beta^{-1}\Delta}\Big) \tag{F.15}$$

$$\geq \widetilde{p}\Big(\frac{1}{1 + \beta^{-1}}\Big)\Delta, \tag{F.16}$$

where (F.14) is because $\log(1 + x) \leq x$ for $x \geq -1$, (F.15) is because $1 - e^{-x} \geq \frac{x}{1+x}$ for $x \geq 0$, (F.16) plugs in (F.12) and uses the selection that $\Delta \leq \frac{1}{2}$.

Summing (F.16) over all $K$ episodes, we obtain

$$
\begin{aligned}
\mathbb{E}\big[\mathrm{Regret}_{\mathcal{M}_1}(\xi, K)\big] &= \sum_{k=1}^{K} \mathbb{E}_1^o\big[V^{\pi^*, \beta}(s_1) - V^{\pi^k, \beta}(s_1)\big] \\
&= \sum_{k=1}^{K} \mathbb{E}_1^o\bigg[\mathbb{1}\{\pi^k(s_2) \neq a_1\} \cdot \widetilde{p}\Big(\frac{1}{1 + \beta^{-1}}\Big)\Delta\bigg] \\
&= \frac{\widetilde{p}\Delta}{1 + \beta^{-1}} \cdot \mathbb{E}_1^o\bigg[\sum_{k=1}^{K} \mathbb{1}\{\pi^k(s_2) \neq a_1\}\bigg] \\
&= \frac{\widetilde{p}\Delta}{1 + \beta^{-1}} \cdot \mathbb{E}_1^o[K - S_1(K)] &\text{(F.17)} \\
&\geq \frac{\widetilde{p}K\Delta}{2(1 + \beta^{-1})}\mathbb{P}_1^o\Big(S_1(K) \leq \frac{K}{2}\Big), &\text{(F.18)}
\end{aligned}
$$

where (F.17) uses the definition of $S_j(K)$ as given in (E.1), and (F.18) follows from Markov's inequality.

For the environment $\mathcal{M}_2$, if $\pi^k(s_2) = a_1$, then the episodic regret can be bounded by

$$
\begin{aligned}
V^{\pi^*, \beta}(s_1) - V^{\pi^k, \beta}(s_1) &= -\beta \log\Big(pe^{-2\beta^{-1}\Delta} + (1-p)e^{-\beta^{-1}}\Big) + \beta \log\Big(pe^{-\beta^{-1}\Delta} + (1-p)e^{-\beta^{-1}}\Big) \\
&= -\beta \log\bigg(\frac{pe^{-2\beta^{-1}\Delta} + (1-p)e^{-\beta^{-1}}}{pe^{-\beta^{-1}\Delta} + (1-p)e^{-\beta^{-1}}}\bigg) \\
&= -\beta \log\bigg(1 + \frac{p(e^{-2\beta^{-1}\Delta} - e^{-\beta^{-1}\Delta})}{pe^{-\beta^{-1}\Delta} + (1-p)e^{-\beta^{-1}}}\bigg) \\
&\geq \beta\bigg(\frac{p}{pe^{-\beta^{-1}\Delta} + (1-p)e^{-\beta^{-1}}}\bigg)(1 - e^{-\beta^{-1}\Delta})e^{-\beta^{-1}\Delta} \\
&\geq \beta e^{-\beta^{-1}}\bigg(\frac{p}{p + (1-p)e^{-\beta^{-1}}}\bigg)\bigg(\frac{\beta^{-1}\Delta}{1 + \beta^{-1}\Delta}\bigg) &\text{(F.19)} \\
&\geq \widetilde{p}\Big(\frac{e^{-\beta^{-1}}}{1 + \beta^{-1}}\Big)\Delta, &\text{(F.20)}
\end{aligned}
$$

where (F.19) uses $e^{-\beta^{-1}} \leq e^{-\beta^{-1}\Delta} \leq 1$, (F.20) plugs in (F.12) and uses the selection that $\Delta \leq \frac{1}{2}$.

Summing (F.20) over all $K$ episodes, following the same analysis, we obtain

$$
\begin{aligned}
\mathbb{E}\big[\mathrm{Regret}_{\mathcal{M}_2}(\xi, K)\big] &= \sum_{k=1}^{K} \mathbb{E}_2^o\big[V^{\pi^*, \beta}(s_1) - V^{\pi^k, \beta}(s_1)\big] \\
&\geq \sum_{k=1}^{K} \mathbb{E}_2^o\bigg[\mathbb{1}\{\pi^k(s_2) = a_1\} \cdot \widetilde{p}\Big(\frac{e^{-\beta^{-1}}}{1 + \beta^{-1}}\Big)\Delta\bigg] \\
&= \frac{e^{-\beta^{-1}}\widetilde{p}\Delta}{1 + \beta^{-1}} \cdot \mathbb{E}_2^o\bigg[\sum_{k=1}^{K} \mathbb{1}\{\pi^k(s_2) = a_1\}\bigg] \\
&= \frac{e^{-\beta^{-1}}\widetilde{p}\Delta}{1 + \beta^{-1}} \cdot \mathbb{E}_2^o[S_1(K)] \\
&\geq \frac{e^{-\beta^{-1}}\widetilde{p}K\Delta}{2(1 + \beta^{-1})}\mathbb{P}_2^o\Big(S_1(K) > \frac{K}{2}\Big). &\text{(F.21)}
\end{aligned}
$$

By combining (F.18) and (F.21) with Lemma G.6, we obtain

$$
\mathbb{E}\big[\mathrm{Regret}_{\mathcal{M}_1}(\xi, K) + \mathrm{Regret}_{\mathcal{M}_2}(\xi, K)\big] \geq \frac{e^{-\beta^{-1}}\widetilde{p}K\Delta}{2(1 + \beta^{-1})}\Big(\mathbb{P}_1^o\Big(S_1(K) \leq \frac{K}{2}\Big) + \mathbb{P}_2^o\Big(S_1(K) > \frac{K}{2}\Big)\Big)
$$

$$\geq \frac{e^{-\beta^{-1}}\widetilde{p}K\Delta}{4(1+\beta^{-1})}\exp\big(-\mathrm{KL}\big(\mathbb{P}_1^o,\mathbb{P}_2^o\big)\big).$$

We complete the proof by setting $\zeta = \frac{e^{-\beta^{-1}}}{4(1+\beta^{-1})}$.

### F.1.4. THE RRMDP-$\chi^2$ SETTING

For the RRMDP-$\chi^2$ setting, the robust value function at state $s_1$, denoted by $V^{\pi,\beta}(s_1)$, is defined as

$$V^{\pi,\beta}(s_1) = \inf_{p'}\left\{p'\cdot V^{\pi,\beta}(s_2) + (1-p')\cdot V^{\pi,\beta}(s_3) + \beta\left[p\left(\frac{p'}{p}-1\right)^2 + (1-p)\left(\frac{1-p'}{1-p}-1\right)^2\right]\right\}. \tag{F.22}$$

Using the construction that $V^{\pi,\beta}(s_3) = 1$ and setting the derivative of the objective function in (F.22) with respect to $p'$ to zero, we obtain

$$p' = p + \frac{p(1-p)\big(1 - V^{\pi,\beta}(s_2)\big)}{2\beta}. \tag{F.23}$$

The worst-case transition probability $\widetilde{p}$ is defined as the maximum value of $p'$ over all policies $\pi$. Since (F.23) is monotonically decreasing in $V^{\pi,\beta}(s_2)$, we set $V^{\pi,\beta}(s_2) = 0$ in (F.23) to obtain

$$\widetilde{p} = p + \frac{p(1-p)}{2\beta}. \tag{F.24}$$

Finally, substituting (F.23) back into (F.22), we derive the expression for the robust value function $V^{\pi,\beta}(s_1)$ as

$$V^{\pi,\beta}(s_1) = pV^{\pi,\beta}(s_2) + (1-p) - \frac{p(1-p)\big(1 - V^{\pi,\beta}(s_2)\big)^2}{4\beta}. \tag{F.25}$$

For the environment $\mathcal{M}_1$, if $\pi^k(s_2) \neq a_1$, then the episodic regret can be bounded by

$$V^{\pi^*,\beta}(s_1) - V^{\pi,\beta}(s_1) = \left(p\Delta + (1-p) - \frac{p(1-p)(1-\Delta)^2}{4\beta}\right) - \left((1-p) - \frac{p(1-p)}{4\beta}\right)$$

$$= p\Delta + \frac{p(1-p)}{4\beta}(2\Delta - \Delta^2)$$

$$\geq \frac{1}{2}p\Delta + \frac{p(1-p)}{4\beta}\Delta \tag{F.26}$$

$$\geq \frac{1}{2}\widetilde{p}\Delta, \tag{F.27}$$

where (F.26) uses the selection that $\Delta \leq \frac{1}{2}$ and (F.27) plugs in (F.24).

Summing (F.27) over all $K$ episodes, we obtain

$$\mathbb{E}\big[\mathrm{Regret}_{\mathcal{M}_1}(\xi,K)\big] = \sum_{k=1}^{K}\mathbb{E}_1^o\big[V^{\pi^*,\beta}(s_1) - V^{\pi^k,\beta}(s_1)\big]$$

$$= \sum_{k=1}^{K}\mathbb{E}_1^o\left[\mathbb{1}\{\pi^k(s_2) \neq a_1\}\cdot\frac{\widetilde{p}}{2}\Delta\right]$$

$$= \frac{\widetilde{p}\Delta}{2}\cdot\mathbb{E}_1^o\left[\sum_{k=1}^{K}\mathbb{1}\{\pi^k(s_2) \neq a_1\}\right]$$

$$= \frac{\widetilde{p}\Delta}{2}\cdot\mathbb{E}_1^o[K - S_1(K)] \tag{F.28}$$

$$\geq \frac{\widetilde{p}K\Delta}{4} \mathbb{P}_1^o\Big(S_1(K) \leq \frac{K}{2}\Big), \tag{F.29}$$

where (F.28) uses the definition of $S_j(K)$ as given in (E.1), and (F.29) follows from Markov's inequality.

For the environment $\mathcal{M}_2$, if $\pi^k(s_2) = a_1$, then the episodic regret can be bounded by

$$
\begin{aligned}
V^{\pi^*,\beta}(s_1) - V^{\pi,\beta}(s_1) &\geq \left(2p\Delta + (1-p) - \frac{p(1-p)(1-2\Delta)^2}{4\beta}\right) - \left(p\Delta + (1-p) - \frac{p(1-p)(1-\Delta)^2}{4\beta}\right) \\
&= p\Delta + \frac{p(1-p)}{4\beta}(2\Delta - 3\Delta^2) \\
&= p\Delta + \frac{p(1-p)}{4\beta}\Delta(2 - 3\Delta) \\
&\geq \frac{1}{4}p\Delta + \frac{p(1-p)}{8\beta}\Delta \tag{F.30} \\
&\geq \frac{1}{4}\widetilde{p}\Delta, \tag{F.31}
\end{aligned}
$$

where (F.30) uses the selection that $\Delta \leq \frac{1}{2}$ and (F.31) plugs in (F.24).

Summing (F.31) over all $K$ episodes, following the same analysis, we obtain

$$
\begin{aligned}
\mathbb{E}\big[\mathrm{Regret}_{\mathcal{M}_2}(\xi, K)\big] &= \sum_{k=1}^{K} \mathbb{E}_2^o\big[V^{\pi^*,\beta}(s_1) - V^{\pi^k,\beta}(s_1)\big] \\
&\geq \sum_{k=1}^{K} \mathbb{E}_2^o\left[\mathbb{1}\{\pi^k(s_2) = a_1\} \cdot \frac{\widetilde{p}}{4}\Delta\right] \\
&= \frac{\widetilde{p}\Delta}{4} \cdot \mathbb{E}_2^o\left[\sum_{k=1}^{K}\mathbb{1}\{\pi^k(s_2) = a_1\}\right] \\
&= \frac{\widetilde{p}\Delta}{4} \cdot \mathbb{E}_2^o[S_1(K)] \\
&\geq \frac{\widetilde{p}K\Delta}{8}\mathbb{P}_2^o\Big(S_1(K) > \frac{K}{2}\Big). \tag{F.32}
\end{aligned}
$$

By combining (F.29) and (F.32) with Lemma G.6, we obtain

$$
\begin{aligned}
\mathbb{E}\big[\mathrm{Regret}_{\mathcal{M}_1}(\xi, K) + \mathrm{Regret}_{\mathcal{M}_2}(\xi, K)\big] &\geq \frac{\widetilde{p}K\Delta}{8}\left(\mathbb{P}_1^o\Big(S_1(K) \leq \frac{K}{2}\Big) + \mathbb{P}_2^o\Big(S_1(K) > \frac{K}{2}\Big)\right) \\
&\geq \frac{\widetilde{p}K\Delta}{16}\exp\big(-\mathrm{KL}\big(\mathbb{P}_1^o, \mathbb{P}_2^o\big)\big).
\end{aligned}
$$

We complete the proof by setting $\zeta = \frac{1}{16}$.

### F.2. Proof of Lemma E.3

In order to prove this, we borrow the same technology as Lattimore & Szepesvári (2020, Lemma 15.1). First, by the definition of KL-divergence, we have that

$$\mathrm{KL}\big(\mathbb{P}_1^o, \mathbb{P}_2^o\big) = \mathbb{E}_1^o\left[\log\left(\frac{\mathrm{d}\mathbb{P}_1^o}{\mathrm{d}\mathbb{P}_2^o}\right)\right]. \tag{F.33}$$

Following the notation defined in Theorem E.1, we calculate the Radon-Nikodym derivative of $\mathbb{P}_1^o$ as follows

$$p_1^o(o^1, a^1, r^1, \cdots, o^K, a^K, r^K) = \prod_{k=1}^{K} \Pr(o^k|s_1) \cdot \pi^k(a^k|o^1, a^1, r^1, \cdots, o^{k-1}, a^{k-1}, r^{k-1}, o^k) \cdot \Pr\big(r_{\mathcal{M}_1}(o^k, a^k) = r^k\big),$$

The density of $\mathbb{P}_2$ is exactly identical except that $r_{\mathcal{M}_1}$ is replaced by $r_{\mathcal{M}_2}$, which gives rise to

$$
\log\left(\frac{d\mathbb{P}_1^o}{d\mathbb{P}_2^o}(o^1, a^1, r^1, \cdots, o^K, a^K, r^K)\right) = \sum_{k=1}^{K} \log \frac{\Pr\left(r_{\mathcal{M}_1}(o^k, a^k) = r^k\right)}{\Pr\left(r_{\mathcal{M}_2}(o^k, a^k) = r^k\right)}
$$

$$
= \sum_{k=1}^{K} \mathbb{1}\{o^k = s_2\} \cdot \log \frac{\Pr\left(r_{\mathcal{M}_1}(s_2, a^k) = r^k\right)}{\Pr\left(r_{\mathcal{M}_2}(s_2, a^k) = r^k\right)}, \tag{F.34}
$$

where (F.34) is because the agent receives a fixed reward $r = 1$ when $o^k = s_3$.

Taking expectations on both sides of (F.34), we obtain

$$
\mathbb{E}_1^o\left[\log\left(\frac{d\mathbb{P}_1^o}{d\mathbb{P}_2^o}(O^1, A^1, R^1, \cdots, O^K, A^K, R^K)\right)\right] \tag{F.35}
$$

$$
= \sum_{k=1}^{K} \mathbb{E}_1^o\left[\mathbb{1}\{O^k = s_2\} \cdot \log\left(\frac{P_{r_{\mathcal{M}_1}(s_2, A^k)}(R^k)}{P_{r_{\mathcal{M}_2}(s_2, A^k)}(R^k)}\right)\right]
$$

$$
= \sum_{k=1}^{K} \mathbb{E}_1^o\left[\mathbb{E}_1^o\left[\mathbb{1}\{O^k = s_2\} \cdot \log\left(\frac{P_{r_{\mathcal{M}_1}(s_2, A^k)}(R^k)}{P_{r_{\mathcal{M}_2}(s_2, A^k)}(R^k)}\right)\,\bigg|\, O^k, A^k\right]\right]
$$

$$
= \sum_{k=1}^{K} \mathbb{E}_1^o\left[\mathbb{1}\{O^k = s_2\} \cdot \mathbb{E}_1^o\left[\log\left(\frac{P_{r_{\mathcal{M}_1}(s_2, A^k)}(R^k)}{P_{r_{\mathcal{M}_2}(s_2, A^k)}(R^k)}\right)\,\bigg|\, O^k, A^k\right]\right] \tag{F.36}
$$

$$
= \sum_{k=1}^{K} \mathbb{E}_1^o\left[\mathbb{1}\{O^k = s_2\} \cdot \mathrm{KL}\left(P_{r_{\mathcal{M}_1}(s_2, A^k)}, P_{r_{\mathcal{M}_2}(s_2, A^k)}\right)\right] \tag{F.37}
$$

$$
= \sum_{j=1}^{|\mathcal{A}|} \mathbb{E}_1^o\left[\sum_{k=1}^{K} \mathbb{1}\{O^k = s_2, A^k = a_j\} \cdot \mathrm{KL}\left(P_{r_{\mathcal{M}_1}(s_2, a_j)}, P_{r_{\mathcal{M}_2}(s_2, a_j)}\right)\right]
$$

$$
= \sum_{j=1}^{|\mathcal{A}|} \mathbb{E}_1^o[T_j(K)] \cdot \mathrm{KL}\left(P_{r_{\mathcal{M}_1}(s_2, a_j)}, P_{r_{\mathcal{M}_2}(s_2, a_j)}\right), \tag{F.38}
$$

where (F.36) is because $\mathbb{1}\{O^k = s_2\}$ is measurable with respect to the $\sigma$-field generated by $O^k$ and $A^k$, (F.37) follows from the definition of KL divergence, (F.38) follows from the definition of $T_j$ in (E.2). Combining (F.38) with (F.33), we conclude the proof.

## G. Auxiliary Lemmas

Here, we present some auxiliary lemmas which are useful in the proof.

**Lemma G.1** (Hoeffding's inequality). (Vershynin, 2018, Theorem 2.2.6) Let $X_1, \cdots, X_T$ be independent random variables. Assume that $X_t \in [0, M]$ for every $t$ with $M > 0$. Let $S_T = \frac{1}{T}\sum_{t=1}^{T} X_t$, then for any $\epsilon > 0$, we have

$$
\mathbb{P}\left(\left|S_T - \mathbb{E}[S_T]\right| \geq \epsilon\right) \leq 2\exp\left(-\frac{2T\epsilon^2}{M^2}\right).
$$

**Lemma G.2** (Self-bounding variance inequality). (Maurer & Pontil, 2009, Theorem 10) Let $X_1, \cdots, X_T$ be independent and identically distributed random variables with finite variance. Assume that $X_t \in [0, M]$ for every $t$ with $M > 0$. Let $S_T = \frac{1}{T}\sum_{t=1}^{T} X_t^2 - (\frac{1}{T}\sum_{t=1}^{T} X_t)^2$, then for any $\epsilon > 0$, we have

$$
\mathbb{P}\left(\left|S_T - \sqrt{\mathrm{Var}(X_1)}\right| \geq \epsilon\right) \leq 2\exp\left(-\frac{T\epsilon^2}{2M^2}\right).
$$

**Lemma G.3.** (Weissman et al., 2003, Theorem 2.1) Let $P$ be a probability distribution over $\mathcal{S} = \{s_1, \cdots, s_S\}$, $X_1, \cdots, X_T$ be independent and identically distributed random variables distributed according to $P$. Let $\widehat{P}(s) = \frac{1}{T} \sum_{t=1}^{T} \mathbb{1}\{X_t = s\}$, then for any $\epsilon > 0$, we have

$$\mathbb{P}\big(\big\|P - \widehat{P}\big\|_1 \geq \epsilon\big) \leq 2^S \exp\left(-\frac{T\epsilon^2}{2}\right).$$

**Lemma G.4.** (Panaganti & Kalathil, 2022, Lemma 7) We define $\mathcal{V} = \big\{V \in \mathbb{R}^S : \|V\|_\infty \leq V_{\max}\big\}$. Let $\mathcal{N}_\mathcal{V}(\epsilon)$ be a minimal $\epsilon$-cover of $\mathcal{V}$ with respect to the distance metric $d(V, V') = \|V - V'\|_\infty$ for some fixed $\epsilon \in (0, 1)$. Then we have

$$\log |\mathcal{N}_\mathcal{V}(\epsilon)| \leq |\mathcal{S}| \cdot \log\left(\frac{3V_{\max}}{\epsilon}\right).$$

**Lemma G.5.** (Dann et al., 2017, Lemma F.4) Let $\mathcal{F}_i$ for $i = 1, 2, \cdots$ be a filtration and $X_1, \cdots, X_n$ be a sequence of Bernoulli random variables with $\mathbb{P}(X_i = 1|\mathcal{F}_{i-1}) = P_i$ being $\mathcal{F}_{i-1}$-measurable and $X_i$ being $\mathcal{F}_i$-measurable. It holds that

$$\mathbb{P}\left(\exists n : \sum_{t=1}^{n} X_t \leq \sum_{t=1}^{n} P_t/2 - W\right) \leq e^{-W}.$$

**Lemma G.6** (Bretagnolle-Huber inequality)**.** (Lattimore & Szepesvári, 2020, Theorem 14.2) Let $P$ and $Q$ be probability measures on the same measurable space $(\Omega, \mathcal{F})$, and let $A \in \mathcal{F}$ be an arbitrary event. Then

$$P(A) + Q(A^c) \geq \frac{1}{2}\exp(-\mathrm{KL}(P, Q)).$$

**Lemma G.7.** (Lattimore & Szepesvári, 2020, Section 4.2) The KL-divergence between two Gaussian distributions with means $\mu_1, \mu_2$ and common variance $\sigma^2$ is

$$\mathrm{KL}\big(\mathcal{N}(\mu_1, \sigma^2), \mathcal{N}(\mu_2, \sigma^2)\big) = \frac{(\mu_1 - \mu_2)^2}{2\sigma^2}.$$

**Lemma G.8.** (Sayyareh, 2011, Theorem 3.1) Let $P$ and $Q$ be two probability distributions, then

$$\mathrm{KL}(P\|Q) \leq \chi^2(P\|Q).$$

