# OpenReview forum: "Sample Complexity of Distributionally Robust Off-Dynamics Reinforcement Learning with Online Interaction"
_ICML.cc/2025/Conference — ICML 2025 poster_

### Official Review · Reviewer_pPuk · 2025-02-23

**Overall Recommendation:** 4

**Summary:**

Other reviews checked. I keep my score. Thanks for the rebuttal.

This paper investigates two types of robust Markov decision processes (RMDPs) in tabular reinforcement learning: CRMDP and RRMDP.

It introduces hard instances and visitation ratios and establishes both regret lower and upper bounds.

The authors also present empirical results to support their findings.

**Claims And Evidence:**

Yes

**Essential References Not Discussed:**

na

**Experimental Designs Or Analyses:**

Yes.

The authors assess the algorithms in a simulated RMDP and the Frozen Lake environment, demonstrating their effectiveness under significant distribution shifts.

**Methods And Evaluation Criteria:**

Yes

**Other Comments Or Suggestions:**

The proposed algorithms follow the robust value iteration template.

Assumption 4.5 is interesting. My understanding is that the intuition behind it is that collecting data in the nominal MDP is somewhat analogous to an offline RL problem in the true MDP, requiring adequate data coverage in the true MDP.

I’m curious whether this necessity assumption could be relaxed to require coverage of only d^{pi*} instead of covering d^pi for all pi.

The KL bound in Theorem 4.19 appears to have an exponential dependence on H. Could you provide some intuition for this bound?

This paper presents many upper bound results. Do you have any comments or comparisons with other existing upper bound results?

**Other Strengths And Weaknesses:**

In other parts

**Questions For Authors:**

Some bonus terms include the 1/K component, while others do not. Could you provide some intuition behind the presence of the 1/K term in the bonus functions?

Additionally, why do some P notations include w while others do not, such as in line 274?

**Relation To Broader Scientific Literature:**

Off-dynamics reinforcement learning (RL) has attracted much attention due to scenarios where the transition dynamics of the deployment environment differ from those during the training process. This approach requires exploration strategies that proactively address distributional shifts, ensuring robustness to dynamic changes.

**Theoretical Claims:**

Yes.

This paper presents hard instances to illustrate that a well-defined exploration hardness condition is essential for general online learning in robust Markov decision processes (RMDPs). When state visits are exponentially rare, online learning becomes intractable.

The authors introduce the supremal visitation ratio as a metric for measuring exploration difficulty in RMDPs. They propose computationally efficient algorithms for CRMDPs and RRMDPs with various uncertainty sets and provide theoretical guarantees for these methods.

Plus, the paper establishes regret lower bounds, showing that the supremal visitation ratio is an unavoidable factor in the sample complexity of online RMDP learning.

---

> ### Author Rebuttal · Authors · 2025-04-01
>
> We thank the reviewer for the positive feedback on our work. We hope our response will fully address all of the reviewer's questions.
>
> ---
>
> ### 1. Discussion on possible relaxation of Assumption 4.5
>
> We appreciate the reviewer's insightful question and agree that Assumption 4.5 might be relaxed to require coverage only of $d^{\pi^\*}$ instead of $d^\pi$ for all $\pi$, similar to the off-policy problem. While this modification is theoretically feasible, it is not straightforward due to the dynamics shift in the deployment environment. It would likely involve revising the way we decompose the regret, leading to non-trivial changes in our current proof structure.
>
>
> ---
>
> ### 2. Explanation about KL bound in Theorem 4.19
>
> As stated by Remark 4.2 in Blanchet et al. (2023), the $e^H$ dependency for KL bounds is commonly observed in the literature. This is due to the logarithmic function used in the definition of KL divergence.
>
>
> ---
>
> ### 3. Do you have any comments or comparisons with other existing upper bound results?
>
> Thank you for the question. We would like to point out that no previous work has explored the exact same setting we are studying. The research Lu et al. (2024), which most closely resembles ours in the constrained TV setting, relies on a different assumption that cannot be generalized to other settings considered in our paper. Unlike their approach, we do not concentrate on one specific scenario of uncertainty sets (constrained TV setting) but instead explore the general online learning of robust MDP.
>
> ---
>
> ### 4. Intuition behind the presence of the 1/K term in the bonus functions
>
> The $1/K$ terms in the bonus functions are not intrinsic. In the proof, we constructed $\epsilon$-nets to bound one part of the estimation, which leads to an additional term of $\mathcal{O}(\epsilon)$ or $\mathcal{O}(\sqrt{\epsilon})$, where the value of $\epsilon$ can be arbitrary. We set $\epsilon=\mathcal{O}(1/K)$ or $\epsilon=\mathcal{O}(1/K^2)$, which results in the extra $1/K$ term in the bonus. For a detailed explanation, please refer to Lemmas E.2, E.12, E.16, F.2, F.6, and F.11 respectively for the specific setting.
>
> ---
>
> ### 5. Clarification about notations such as on line 274
>
> As defined in Definition 4.3, $P^o$ represents the nominal transition, $P^w$ represents the worst-case transition, and $P^\pi$ is the corresponding visitation measure. These are distinctly different concepts.
>
> ---
> We hope we have addressed all of your questions/concerns. If you have any further questions, we would be more than happy to answer them.
>
>
> ### References:
>
> [1] Blanchet, J., Lu, M., Zhang, T., & Zhong, H. (2023). Double pessimism is provably efficient for distributionally robust offline reinforcement learning: Generic algorithm and robust partial coverage. Advances in Neural Information Processing Systems, 36, 66845-66859.
>
> [2] Lu, M., Zhong, H., Zhang, T., & Blanchet, J. (2024). Distributionally robust reinforcement learning with interactive data collection: Fundamental hardness and near-optimal algorithm. arXiv preprint arXiv:2404.03578.

---

### Official Review · Reviewer_9Uk6 · 2025-03-04

**Overall Recommendation:** 3

**Summary:**

The paper considers two types of robust MDP formulations - one with constraints and one with regularization.

A new value update is proposed for both which can be shown to guarantee for a tabular case some regret bounds.

Lower and upper regret bounds are given as a function of the supremal visitation ratio which is introduced in the paper.

The authors also show case when the regret is especially high due the supremal visitation ratio.

Finally, the authors provide some simple experiments to support their claims.

**Claims And Evidence:**

The paper is mainly theoretical and its claims are well supported.

**Essential References Not Discussed:**

No to my best knowledge.

**Experimental Designs Or Analyses:**

No.

**Methods And Evaluation Criteria:**

The authors provide experiments that show general behavior as a function of the supremal visitation ratio, but these experiments do not seem to reflect the behavior of the regret they found ($\sqrt{C_{vr}}$).  Also if they could show the behavior reflects the other parameters in their bound it would've been better (e.g. $S$, $A$).

**Other Comments Or Suggestions:**

None.

**Other Strengths And Weaknesses:**

The paper is written very clearly, it has a solid theoretical result that can be significant to the community.

The weaknesses of the paper in my opinion are:

1. The bounds in 3.2 and 3.3. are not explained enough. What is the difference between all of them? do the terms we see there makes sense?

2. It's nice that the proofs are in the appendix, but it would be good if you share the general intuition beyond them - what was the main idea in the proof. This is in my view much more important than showing the bounds for another case.

3. In the third page the text looks cramped. I hope it's an artifact in my viewer since changing the spaces between lines is prohibited as far as I know.

4.  The experiments are not very important because the result is mainly theoretical - but still. The first environment is too simple (H=3 is barely RL). As I wrote before it would make sense to show that the regret bounds behave like you would expect it for all of its parameters (S, A, K, Crv...). I didn't see a discussion on why in Figure 1 ORBIT works less good when there is small or no perturbation?

**Questions For Authors:**

Can you provide a proof sketch for the main result in the paper (Thm. 4.14 and also 4.17)?

**Relation To Broader Scientific Literature:**

The key contributions seem to be well related to relevant literature.

**Theoretical Claims:**

No.

---

> ### Author Rebuttal · Authors · 2025-04-01
>
> Thank you for your valuable time and effort in providing detailed feedback on our work. We hope our response will fully address all of your questions.
>
> ---
>
> ### 1. Explanation about the bounds in 3.2 and 3.3
>
> Sections 3.2 and 3.3 present update formulas for different types of robust sets (constrained vs. regularized) and different $f$-divergences (TV, KL, and $\chi^2$). Although these updates all follow a similar pattern--applying an optimism-based principle and a robust Bellman operator--their specific forms differ because the dual formulations vary across robust set types and divergence measures. Hence, the resulting $Q$-function estimates and bonus terms change accordingly.
>
> We have outlined the final equations for the updates in Sections 3.2 and 3.3 due to space constraints, but we agree that more in-depth explanation would help clarify how each term arises from the respective dual formulation. We plan to include further details in the final version of the paper to make the derivations and resulting bounds more transparent.
>
> ---
>
> ### 2. Proof sketch of this paper
>
> We summarize the core proof idea as follows.
>
> For the upper bounds, we decompose the regret as
> $\text{Regret}=\sum\limits_{k=1}^K\big(V_1^{\*,\rho}-V_1^{{\pi^k},\rho}\big)=\sum\limits_{k=1}^K\big(V_1^{\*,\rho}-V_1^{k,\rho}\big)+\sum\limits_{k=1}^K\big(V_1^{k,\rho}-V_1^{{\pi^k},\rho}\big)$.
>
> We use the dual formulation to estimate $V$ and add an extra bonus term to maintain an optimistic estimate, ensuring that the first term is less than zero. From the definition of $V^k$, it follows that the second term can be controlled by $\sum\limits_{k=1}^K\sum\limits_{h=1}^H\mathbb{E}\_{\\{P_h^{w,k}\\}_{h=1}^H,\pi^k}\big[2\text{bonus}_h^k\big]$. The cumulative bonus is of an order not larger than $\sqrt{K}$, as derived from the concentration inequalities.
>
> For the lower bounds, we construct a key state that is difficult to explore in the nominal environment but easier to explore in the perturbed environment. Consequently, insufficient knowledge about this key state can lead to significant regret.
>
> We will add the proof sketch in the final version.
>
> ---
>
>
> ### 3. The text display on the third page
>
> We believe this might be due to the floating environment in LaTeX.
>
> ---
>
> ### 4. Further clarifications on experiment setup and observed performance
>
>
> In the case of $C_{vr}$, we constructed a well-designed MDP environment where the hyperparameter $\beta$ allows for adjustments to the value of $C_{vr}$ to observe performance changes. However, the $\frac{1}{2}$ order dependency on $C_{vr}$ represents merely a worst-case bound. Furthermore, the regret formulations incorporate additional terms, so we do not anticipate results to be strictly proportional to $\sqrt{C_{vr}}$. Here, we merely demonstrate a general positive correlation to validate our intuition, akin to an ablation study.
>
> Regarding $S$, $A$, and other variables, they are not the primary focus within this topic. The crucial parameters here are $K$ (indicating efficient learning of the problem) and $C_{vr}$ (demonstrating that $C_{vr}$ is a precise metric for the difficulty of the environment). Unlike $C_{vr}$, which is relatively easier to manipulate, altering the values of $S$ and $A$ would lead to a complete structural change in the environment. Given that our environment is specifically designed, it is challenging to introduce $S$ and $A$ as variables into the existing problem.
>
> To answer the reviewer's question, we have conducted a new experiment on the Gambler's Problem (inspired by Panaganti et al., 2022, Section 6). We set the heads-up probability to $p_h=0.6$ for the nominal environment and $p_h=0.4$ for the perturbed environment. We applied the constrained TV setting with $H=30$ and varied the size of $S$ from 15 to 55, reporting the results across 10 training runs as shown in the following table. This demonstrates that performance deteriorates as the sizes of $S$ and $A$ increase.
>
> |size of S|15|23|31|39|47|55|
> |-|-|-|-|-|-|-|
> |mean of reward|0.1294|0.1254|0.1192|0.1240|0.1140|0.0968|
> |std of reward|0.0088|0.0134|0.0164|0.0163|0.0219|0.0118|
>
>
> ---
>
> ### 5. Explanation about performance under minimal perturbation in Figure 1
>
> As is well-known in robust reinforcement learning, algorithms designed to hedge against large uncertainties naturally trade off some performance in near-nominal settings. Therefore, while ORBIT typically outperforms non-robust approaches under significant perturbations, it may appear less effective when perturbations are small or absent because it prioritizes worst-case scenarios over short-term gains.
>
>
> ---
> We hope we have addressed all of your questions/concerns. If you have any further questions, we would be more than happy to answer them.
>
>
> ### References:
>
> [1] Panaganti, K., & Kalathil, D. (2022, May). Sample complexity of robust reinforcement learning with a generative model. In International Conference on Artificial Intelligence and Statistics (pp. 9582-9602). PMLR.

---

### Official Review · Reviewer_WS6K · 2025-03-11

**Overall Recommendation:** 4

**Summary:**

This paper explores robust Markov decision processes (RMDPs) in the context of off-dynamics reinforcement learning, where distribution shifts occur between training and deployment environments. The authors investigate two variants: constrained RMDPs (CRMDPs) and regularized RMDPs (RRMDPs). They propose computationally efficient algorithms for both settings and establish sublinear regret guarantees, demonstrating the effectiveness of their approach in mitigating performance degradation due to environmental shifts.

**Claims And Evidence:**

The paper is well-structured, with clear results. Section 3 introduces the ORBIT algorithm (Online Robust Bellman Iteration), which serves as a foundation for addressing both RMDPs and CRMDPs. Section 4 establishes the corresponding regret bounds, demonstrating the theoretical performance of the proposed methods. Finally, Section 5 presents numerical experiments that validate the effectiveness of the approach in practical scenarios.

**Essential References Not Discussed:**

-

**Experimental Designs Or Analyses:**

See points raised above

**Methods And Evaluation Criteria:**

I am not entirely clear on the key insights from Figure 2. If I understand correctly, in the CRMDP setting, the TV-ambiguity set appears to yield the best performance, while in the RRMDP setting, the $\chi^2$-ambiguity set performs better. Can this be theoretically justified, or is it purely an empirical observation? If it is the latter, what makes this result interesting, and how can we be sure it is not merely an artifact of the Frozen Lake problem setting?

This also raises a broader question: How should one decide which ambiguity set to use? Can you provide theoretical insights to guide this choice, and do the numerical results align with such theoretical expectations?

**Other Comments Or Suggestions:**

-

**Other Strengths And Weaknesses:**

MDPs under environment shifts are a crucial and challenging topic in both the ML and OR communities. This paper explores an interesting direction in addressing these challenges, presenting a well-written and relatively accessible exposition.

**Questions For Authors:**

1) Can you motivate better the specific choice of ambiguity sets?
2) All uncertainty sets are defined a rectangular ambiguity sets. Uncertainty sets defined from maximum likelihood principles, however turn out to be non-rectangular. Can you also handle those?

**Relation To Broader Scientific Literature:**

The related literature is adequately summarized. I have no compaints here.

**Theoretical Claims:**

I am wondering why the authors decided to choose exactly these 3 ambiguity sets: (1) TV, (2) KL, (3) $\chi^2$?

Could you also include the popular Wasserstein or MMD ambiguity set?

---

> ### Author Rebuttal · Authors · 2025-04-01
>
> We thank the reviewer for the positive feedback on our work. We hope our response will fully address all of the reviewer's questions.
>
> ---
>
> ### 1. The key insights from Figure 2
>
> The primary purpose of Figure 2 is to show that our algorithm converges stably by the end of training in a fixed environment, rather than to compare performance under different ambiguity sets. In the Frozen Lake experiment, our main goal is to highlight the algorithm’s robustness, specifically its ability to handle worst-case scenarios effectively.
>
> ---
>
> ### 2. How should one decide which ambiguity set to use?
>
> As depicted in Figure 3(a), the computational costs vary with different ambiguity settings, where the regularized setting generally proves to be more computationally efficient than the constrained setting. Selecting an appropriate ambiguity set (TV v.s. KL v.s. $\chi^2$) may be specific to the application and the structure of the problem. For hyperparameters like the radius or the regularization parameters, we can select them via hyperparameter tunning based on empirical performance.
>
> ---
>
> ### 3. Can you better motivate the specific choice of ambiguity sets?
>
> Our primary goal is to study the general online robust problem. We focus on these three ambiguity sets because they are widely used and extensively investigated in the reinforcement learning literature (Panaganti & Kalathil, 2022; Yang et al., 2022; Xu et al., 2023; Shi et al., 2024), enabling more direct comparisons with existing studies.
>
> ---
>
> ### 4. Could you also include the popular Wasserstein or MMD ambiguity set?
>
> Thanks for the great suggestion! Our algorithm has the potential to be adapted to these two ambiguity sets since we address the general online robust problem. However, this adaptation might require additional analysis, as these ambiguity sets lack a closed-form dual formulation, posing challenges for direct application of our current algorithm. This will be a great future direction on online robust MDPs.
>
> ---
>
> ### 5. Can you also handle those non-rectangular ambiguity sets?
>
> Non-rectangular ambiguity sets differ fundamentally from those discussed in this paper, and our algorithm is not equipped to handle these due to their inherent complexity. To our knowledge, nearly all studies in the robust MDP literature use rectangular ambiguity sets. Moreover, Wiesemann et al. (2013) showed that solving DRMDPs with general uncertainty sets can be NP-hard. We therefore identify this as an open problem for future research.
>
>
> ---
> We hope we have addressed all of your questions/concerns. If you have any further questions, we would be more than happy to answer them.
>
>
> ### References:
>
> [1] Panaganti, K., & Kalathil, D. (2022, May). Sample complexity of robust reinforcement learning with a generative model. In International Conference on Artificial Intelligence and Statistics (pp. 9582-9602). PMLR.
>
> [2] Yang, W., Zhang, L., & Zhang, Z. (2022). Toward theoretical understandings of robust markov decision processes: Sample complexity and asymptotics. The Annals of Statistics, 50(6), 3223-3248.
>
> [3] Xu, Z., Panaganti, K., & Kalathil, D. (2023, April). Improved sample complexity bounds for distributionally robust reinforcement learning. In International Conference on Artificial Intelligence and Statistics (pp. 9728-9754). PMLR.
>
> [4] Shi, L., Li, G., Wei, Y., Chen, Y., Geist, M., & Chi, Y. (2023). The curious price of distributional robustness in reinforcement learning with a generative model. Advances in Neural Information Processing Systems, 36, 79903-79917.
>
> [5] Wiesemann, W., Kuhn, D., & Rustem, B. (2013). Robust Markov decision processes. Mathematics of Operations Research, 38(1), 153-183.

---

### Official Review · Reviewer_uMDK · 2025-03-13

**Overall Recommendation:** 3

**Summary:**

This paper considers distributionally robust RL with online access to the nominal model, including the constrained MDP (CMDP) and the regularized robust MDP (RRMDP) frameworks. They propose *supremal visitation ratio* $C_{vr}$ as a hardness measure and show that this measure is unavoidable in the regret lower bounds. They also provide numerical results to validate how $C_{vr}$ would affect learning efficiency.

**Claims And Evidence:**

See Question 1.

**Essential References Not Discussed:**

No.

**Experimental Designs Or Analyses:**

The experimental setups in Section 5 are clearly described.

**Methods And Evaluation Criteria:**

Yes.

**Other Comments Or Suggestions:**

1. What are the definitions of $P_h^\pi$ and $d_h^\pi$ in Assumption 4.5?

**Other Strengths And Weaknesses:**

**Strengths:**

1. This paper introduces supremal visitation ratio as a hardness measure and shows it is unavoidable in the lower bounds.
2. They present numerical results showing how the harness measure would affect learning performance.

**Weaknesses:** See Questions.

**Questions For Authors:**

1. Can you compare ORBIT and the algorithms in (Liu & Xu, 2024a) and (Lu et al., 2024)?
2. In Figure 3(a), why the average training time for $\chi^2$ cases is significantly longer?
3. Is it correct that the fail-state conditions 4.1 and 4.2 are special cases of bounded visitation measure ratio?

**Relation To Broader Scientific Literature:**

N/A

**Theoretical Claims:**

See Questions.

---

> ### Author Rebuttal · Authors · 2025-04-01
>
> Thank you for your valuable time and effort in providing detailed feedback on our work. We hope our response will fully address all of your questions.
>
> ---
>
> ### 1. The definitions of $P_h^\pi$ and $d_h^\pi$ in Assumption 4.5
>
> The definitions are provided in Definition 4.3, where $P_h^\pi$ represents the visitation measure under the worst-case scenario, and $d_h^\pi$ represents the visitation measure in the nominal environment, both are defined to be induced by the policy $\pi$.
>
> ---
>
> ### 2. Comparison of ORBIT with algorithms in Liu and Xu (2024a) and Lu et al. (2024)
>
> All three works investigate the online robust MDP setting using value-iteration-based methods. In particular, Liu and Xu (2024a) focuses on the linear function approximation regime, whereas Lu et al. (2024) and our work consider the tabular setting. However, Liu and Xu (2024a) and Lu et al. (2024) address only the constrained TV-divergence scenario. In contrast, our study addresses a broader range of online robust problems, including robust sets defined by three different $f$-divergences, as well as constrained RMDP and regularized RMDP, the latter of which was not considered in their work. Consequently, our algorithm provides distinct update formulations and bonus terms across six settings.
>
> ---
>
> ### 3. Discussion on the average training time for $\chi^2$ cases in Figure 3(a)
>
> Lemmas E.15 and F.10 reveal that deriving the dual formulation for the $\chi^2$ update requires solving a multi-dimensional optimization problem, which leads to longer training times. In contrast, the constrained KL setting (Lemma E.11) relies only on one-dimensional optimization. Furthermore, for constrained TV (as illustrated in Algorithm 2), regularized TV (Lemma F.1), and KL settings (Lemma F.5), the optimization problems have closed-form solutions. This is why the average training time for $\chi^2$ cases is significantly longer.
>
> ---
>
> ### 4. Is it correct that the fail-state conditions 4.1 and 4.2 are special cases of bounded visitation measure ratio?
>
> No, our assumptions do not imply theirs, nor do theirs imply ours. The fail-states condition was specifically designed for the constrained TV robust sets, as explained in Proposition 4.4. In contrast, our assumption addresses the general online robust problem and is applicable to various other $f$-divergence robust sets.
>
>
> ---
> We hope we have addressed all of your questions/concerns. If you have any further questions, we would be more than happy to answer them.
>
>
> ### References:
>
> [1] Liu, Z., & Xu, P. (2024, April). Distributionally robust off-dynamics reinforcement learning: Provable efficiency with linear function approximation. In International Conference on Artificial Intelligence and Statistics (pp. 2719-2727). PMLR.
>
> [2] Lu, M., Zhong, H., Zhang, T., & Blanchet, J. (2024). Distributionally robust reinforcement learning with interactive data collection: Fundamental hardness and near-optimal algorithm. arXiv preprint arXiv:2404.03578.

---

### Decision · Program_Chairs · 2025-05-01

**Decision:**

Accept (poster)

**Comment:**

Sample Complexity of Distributionally Robust Off-Dynamics Reinforcement Learning with Online Interaction

Paper-Summary: This paper addresses the problem of robust policy learning under dynamics uncertainty in online reinforcement learning. Unlike prior works that rely on offline data or assume strong coverage conditions, this paper focuses on online settings where the agent can only interact with the nominal training environment but must learn policies that generalize to distributionally shifted deployment dynamics. The authors study two robust MDP formulations—Constrained Robust MDPs (CRMDPs) and Regularized Robust MDPs (RRMDPs). A key contribution is the introduction of the supremal visitation ratio as a metric to capture the exploration difficulty under distributional shift. The paper shows that if the ratio is unbounded, sample-efficient learning is provably impossible. It proposes a computationally efficient algorithm called ORBIT and proves regret bounds that optimally depend on this ratio and the number of episodes. The theoretical findings are validated with numerical experiments on simulated environments and the Frozen Lake benchmark.


Review and feedback: We received 4 expert reviews, with scores, 3, 3, 4, 4, and the average score is 3.5.

The reviewers are generally positive about this paper. The introduction of supremal visitation ratio (C_vr) as a general measure of robustness-induced exploration difficulty is very interesting and theoretically justified, and is appreciated by all reviewers. The paper provides comprehensive regret analyses for both CRMDPs and RRMDPs, achieving tight sample complexity bounds that depend optimally on key parameters such as horizon length, state-action space size, and C_vr. The proposed algorithm, ORBIT, is computationally tractable and generalizable across different uncertainty sets, including Total Variation, KL, and χ². Hard instance constructions and impossibility results are particularly well studied. The paper is well-written and accessible.

Reviewers have also made suggestions for improvement. One concern is the lack of experimental results beyond a simple tabular setting.  Reviewers have also asked to include more qualitative/intuitive explanations for the proofs and justification for the assumptions.
Please address these comments/suggestions while preparing the final submission.